# A transcriptomic taxonomy of mouse brain-wide spinal projecting neurons

Carla C. Winter[1,2,3,4,5,13], Anne Jacobi[1,2,3,12,13✉], Junfeng Su[1,2,3,13], Leeyup Chung[1,2,3], Cindy T. J. van Velthoven[6], Zizhen Yao[6], Changkyu Lee[6], Zicong Zhang[1,2,3], Shuguang Yu[1,2,3], Kun Gao[7,8], Geraldine Duque Salazar[1,2,3], Evgenii Kegeles[1,2,3,4], Yu Zhang[1,2,3], Makenzie C. Tomihiro[1,2,3], Yiming Zhang[1,2,3], Zhiyun Yang[1,2,3], Junjie Zhu[1,2,3], Jing Tang[1,2,3], Xuan Song[1,2,3], Ryan J. Donahue[1,2,3], Qing Wang[7,8], Delissa McMillen[6], Michael Kunst[6], Ning Wang[6], Kimberly A. Smith[6], Gabriel E. Romero[9], Michelle M. Frank[9], Alexandra Krol[10], Riki Kawaguchi[7,8], Daniel H. Geschwind[7,8], Guoping Feng[10], Lisa V. Goodrich[9], Yuanyuan Liu[11], Bosiljka Tasic[6], Hongkui Zeng[6✉] & Zhigang He[1,2,3✉]

The brain controls nearly all bodily functions via spinal projecting neurons (SPNs) that carry command signals from the brain to the spinal cord. However, a comprehensive molecular characterization of brain-wide SPNs is still lacking. Here we transcriptionally profiled a total of 65,002 SPNs, identified 76 region-specific SPN types, and mapped these types into a companion atlas of the whole mouse brain[1]. This taxonomy reveals a three-component organization of SPNs: (1) molecularly homogeneous excitatory SPNs from the cortex, red nucleus and cerebellum with somatotopic spinal terminations suitable for point-to-point communication; (2) heterogeneous populations in the reticular formation with broad spinal termination patterns, suitable for relaying commands related to the activities of the entire spinal cord; and (3) modulatory neurons expressing slow-acting neurotransmitters and/or neuropeptides in the hypothalamus, midbrain and reticular formation for 'gain setting' of brain–spinal signals. In addition, this atlas revealed a LIM homeobox transcription factor code that parcellates the reticulospinal neurons into five molecularly distinct and spatially segregated populations. Finally, we found transcriptional signatures of a subset of SPNs with large soma size and correlated these with fast-firing electrophysiological properties. Together, this study establishes a comprehensive taxonomy of brain-wide SPNs and provides insight into the functional organization of SPNs in mediating brain control of bodily functions.

Descending pathways, consisting of SPNs that project directly from various brain regions to the spinal cord, transform brain commands ('thoughts') into the bodily behavioural repertoire ('actions'). These SPNs are essential not only for voluntary and involuntary movements but also for sensory modulation and autonomic functions such as blood pressure, heart rate and the fear response[2–9]. The importance of SPNs is illustrated by sensory, motor and autonomic dysfunction following injury to their soma or axons in spinal cord injury, stroke and amyotrophic lateral sclerosis. As a result, comprehensive characterization of SPNs is essential for decoding brain–body interactions and developing neural repair strategies after injury.

Pioneered by Kuypers[8], early efforts aimed to characterize SPNs and their projections with a variety of anatomical tracing methods and classified SPNs into different descending tracts[4]. For example, corticospinal neurons (CSNs) in the motor and somatosensory cortices and rubrospinal neurons (RuSNs) in the midbrain (MB) red nucleus elaborate the corticospinal tract and rubrospinal tract, respectively. By contrast, the reticulospinal neurons (ReSNs) in the MB and pontomedullary reticular formation project their axons diffusely in different parts of the spinal cord, forming the anatomically, molecularly and functionally poorly defined reticulospinal tract. However, these early studies were limited by inefficient targeting and relied on fragmented descriptions of SPNs arising from various supraspinal regions across studies and species. To overcome these challenges, more recent efforts have sought to create brain-wide anatomical atlases of SPNs and/or used virus-mediated retrograde labelling[10–12]. In light of increasingly

[1]F. M. Kirby Neurobiology Center, Boston Children's Hospital, Boston, MA, USA. [2]Department of Neurology, Harvard Medical School, Boston, MA, USA. [3]Department of Ophthalmology, Harvard Medical School, Boston, MA, USA. [4]PhD Program in Biological and Biomedical Sciences, Harvard Medical School, Boston, MA, USA. [5]Harvard-MIT MD-PhD Program, Harvard Medical School, Boston, MA, USA. [6]Allen Institute for Brain Science, Seattle, WA, USA. [7]Semel Institute for Neuroscience and Human Behavior, David Geffen School of Medicine, University of California Los Angeles, Los Angeles, CA, USA. [8]Program in Neurogenetics, Department of Neurology, David Geffen School of Medicine, University of California Los Angeles, Los Angeles, CA, USA. [9]Department of Neurobiology, Harvard Medical School, Boston, MA, USA. [10]McGovern Institute for Brain Research, Department of Brain and Cognitive Sciences, Massachusetts Institute of Technology, Cambridge, MA, USA. [11]Somatosensation and Pain Unit, National Institute of Dental and Craniofacial Research, National Center for Complementary and Integrative Health, National Institutes of Health, Bethesda, MD, USA. [12]Present address: F. Hoffman-La Roche, pRED, Basel, Switzerland. [13]These authors contributed equally: Carla C. Winter, Anne Jacobi, Junfeng Su. ✉e-mail: Anne.Jacobi@roche.com; HongkuiZ@alleninstitute.org; zhigang.he@childrens.harvard.edu

recognized neuronal heterogeneity in individual brain regions[13,14], it has been difficult to infer properties of SPNs without cell-type-specific data. Recent studies have considered the cellular heterogeneity of SPN tracts using single-cell transcriptomics, but these studies relied on retrograde labelling from restricted spinal levels with limited numbers of SPNs[15] and/or only characterized CSNs[16]. As a consequence, for many SPN groups there are no molecular markers available, and the overall cellular organization of the spinal projecting system is still unknown. In addition, as SPNs constitute only a small portion of the resident neurons in individual brain regions, and their extraordinary size probably underlies selective vulnerability during high-throughput cell isolation procedures, transcriptomic profiling efforts across the whole brain (WB) must specifically enrich and optimize for these unique cells.

Here we used efficient virus-mediated retrograde labelling, WB imaging and high-throughput single-nucleus transcriptomics to build a unified anatomic and transcriptomic atlas of SPNs across the entire mouse brain at single-cell resolution. The present study reveals several important organizational and functional principles that underlie the brain control of spinal cord circuits.

## Generation of a transcriptomic atlas of SPNs

Owing to their complex anatomical distribution and fragility, brain-wide SPNs have been difficult to characterize at single-cell resolution. To tackle this, we developed a pipeline to efficiently label, image and transcriptionally profile SPN nuclei across the entire adult mouse brain (Fig. 1a). First, we retrogradely labelled SPNs via injection of recombinant retrograde adeno-associated virus (rAAV2/retro-Syn-H2B-fluorescent protein)[17] into multiple segments centring on the cervical (green fluorescent protein (GFP)) and lumbar (mScarlet) spinal cord of postnatal day 42 (P42) C57BL/6J mice (Extended Data Fig. 1). At postnatal day 56 (P56), we assessed the anatomical distribution of SPNs across the whole mouse brain with serial two-photon tomography (STPT) (Fig. 1b, Extended Data Fig. 2a and Supplementary Video 1) and traditional serial sectioning and slide-scanner imaging. As expected, SPNs are widely distributed across the mouse brain[10–12], but with concentrations in the cortex (CTX; 41.77% of all SPNs), hypothalamus (HY; 2.90%), MB (12.88%), cerebellum (CB; 2.18%), rostral pons (PONS; 16.01%) and caudal pons and medulla (MED; 24.26%) (Extended Data Fig. 2b), consistent with findings from non-viral labelling methods[10].

To generate the transcriptomic taxonomy, we relied on high-throughput fluorescence activated nucleus sorting (FANS) and single-nucleus RNA sequencing (snRNA-seq). Adult C57BL/6J mice received retrograde labelling, were euthanized at P56, and the major regions of interest (ROIs) were microdissected (Extended Data Fig. 3a–c). Equal numbers of male and female mice were pooled per ROI to input sufficient nuclei for FANS (Extended Data Fig. 3b). Each dissection ROI was kept as a separate sample, and the nuclei of SPNs were isolated and sorted by FANS and verified by microscopic inspection (Extended Data Fig. 3d,e). snRNA-seq profiles of all ROIs were generated with the 10x v3.1 (10x) platform and profiles of targeted regions were generated with SMART-Seq v4 (SSv4) (Supplementary Table 1).

In addition to directly retrogradely labelling SPNs ('first-order'), adeno-associated virus (AAV) vectors may also transduce 'second-order' cells owing to transsynaptic labelling[18] or release of AAV after neuronal degeneration. In addition to tracing expected SPNs in CSN-harbouring cortical layer 5 (L5CTX), the retrograde targeting protocol also labels a small proportion of nuclei in other cortical layers (Extended Data Fig. 4a,b). As a result, we attempted to distinguish the directly labelled SPNs (first-order) from these indirectly labelled (second-order) nuclei in the snRNA-seq dataset in silico. We reasoned that first-order neurons should have higher expression of fluorescent proteins (XFP) than second-order nuclei because of longer expression duration and higher

viral load. As expected, the fluorescent protein level, on average, was higher in nuclei within layer 5 than in nuclei outside of layer 5 (Extended Data Fig. 4c). In addition, we found that the proportion of nuclei within a cluster that expressed mRNA for XFP was below 10% in clusters corresponding to known non-SPNs: glia, non-layer 5 cortical neurons and cerebellar granule cells (Extended Data Fig. 4d). Following this, we applied this methodology to the whole snRNA-seq dataset to identify the first- and second-order cell types (Extended Data Fig. 4e–g) and removed these putative second-order nuclei for downstream analyses. It should be noted that there is one exception: type MB-EW-Cck-Ucn was below the 10% threshold but was not removed because of literature support of *Ucn*+ Edinger–Westphal (EW) neurons projecting to the spinal cord[19] and confirmatory anterograde tracing results (Extended Data Fig. 5).

After applying standard quality control metrics and removing second-order labelled nuclei in silico, we obtained 65,002 high-quality snRNA-seq profiles of SPNs ($n = 61,484$ 10x; $n = 3,518$ SSv4) (Extended Data Fig. 6a). A multilevel iterative clustering approach was used to generate a transcriptomic taxonomy of SPNs (Fig. 1c and Extended Data Fig. 6b). In the first level of the global dendrogram (Fig. 1c), SPNs were classified into three divisions: (1) glutamatergic neurons in the CTX, MB and CB ('CTX MB CB glutamatergic', 34,565 neurons, 53.18% of all SPNs); (2) glutamatergic and GABA/glycinergic neurons in the MB and hindbrain (HB) ('MB HB glutamatergic or GABAergic/glycinergic', 23,174, 35.65%); and (3) subcortical modulatory neurons ('Modulatory', 7,263, 11.17%). The second round of clustering split these broad divisions into 13 subclasses, and a third round parcellated the subclasses into 76 types defined by differentially expressed marker genes (Supplementary Table 2). Unsupervised clustering and visualization in uniform manifold approximation and projection (UMAP) space reveals nuclei from different ROI-enriching dissections are largely segregated yet have some overlapping types, probably reflecting anatomical continuity and/or transcriptomic similarities across multiple brain regions (Fig. 1d). Divisions and subclasses span multiple brain regions, suggesting that segregation of cell type is a combination of regional and molecular identities (Fig. 1e,f). The global relationships between types are represented by a constellation plot (Extended Data Fig. 7). Types are conserved across replicates and have a median transcript capture of 20,224 unique molecular identifiers per 10x profile (Extended Data Fig. 6c). This approach effectively assigns SPN types into traditionally defined spinal projecting populations, such as CTX-derived CSNs, CB-derived cerebellospinal neurons (CbSNs), red nucleus-derived RuSNs, brainstem-derived ReSNs and modulatory populations such as hypothalamospinal neurons and raphespinal neurons.

## Integration of SPN and WB taxonomies

To begin validating the taxonomy, we leveraged WB single-cell RNA sequencing (scRNA-seq) and multiplexed error-robust fluorescence in situ hybridization (MERFISH) datasets generated by the Allen Institute for Brain Science (AIBS) in a companion study[1]. The MERFISH data were generated using a 500-gene panel design based on AIBS' WB transcriptomic taxonomy by selecting the set of marker genes with the greatest distinguishing power among all the transcriptomic clusters. First, we mapped (Supplementary Tables 3–5) and assessed the correspondence of (Extended Data Fig. 8) types defined in the SPN taxonomy to the AIBS WB scRNA-seq taxonomy. While great cell-type correspondence for the majority of SPN types in divisions 1 and 3 is observed, SPNs from the red nucleus in division 1 and some division 2 SPN types from the PONS and MED mapped to a single AIBS WB cluster, 4347 PGRN-PARN-MDRN Hoxb5 Glut_2 (Extended Data Fig. 8). Neurons in these areas are highly heterogeneous but distinctions between cell types are subtle. In addition, these neurons are heavily myelinated, making them very difficult to isolate for scRNA-seq without

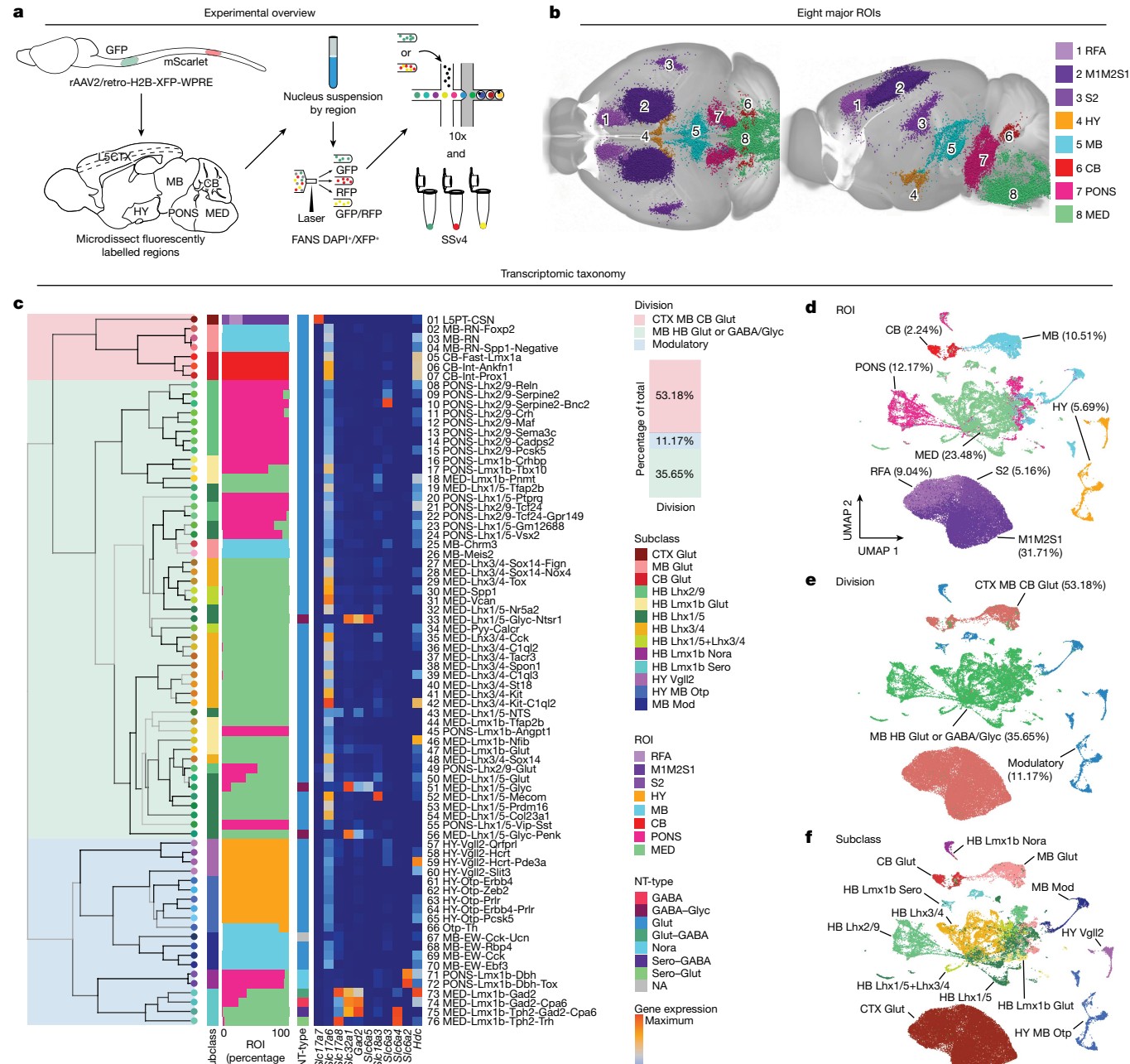

**Fig. 1 | An anatomically informed transcriptomic atlas of brain-wide spinal projecting neurons. a**, SPNs were labelled via spinal cord injections of a retrograde AAV construct that localizes fluorescent protein expression to the nucleus for parallel anatomical and transcriptomic profiling. Test tubes were drawn using templates from Servier Medical Art (Creative Commons Attribution 3.0 Unported Licence https://creativecommons.org/licences/by/3.0/). The sagittal atlas outline was adapted from the Allen Mouse Brain Atlas (https://atlas.brain-map.org/)[53]; ©2017 Allen Institute for Brain Science. **b**, WB reconstructions of segmented and registered SPN nuclei imaged with STPT revealed that SPNs are present across the CTX (RFA, M1M2S1 and S2), HY, MB, CB, PONS and MED. **c**, Multilevel iterative clustering revealed 76 SPN transcriptomic 'types' organized into a taxonomy across 13 'subclasses' and 3 'divisions' (*n* = 61,484 10x; *n* = 3,518 SSv4). The colour blocks shading the taxonomy tree indicate division. The nodes at the end of the dendrogram indicate type, with type numbers and names on the far right. From left to right, the bar plots indicate subclass, fractions of nuclei profiled from brain region-enriching dissections and NT-type across each type. The heatmap shows expression of neurotransmitter marker genes. The bar plot below the division legend indicates percentage of total nuclei belonging to each division. **d–f**, Clustering of SPNs and visualization in UMAP space coloured by brain ROI-enriching dissections (**d**), division (**e**) and subclass (**f**). ROI, division, and subclass colours in **c** apply to **d,e** and **f**, respectively. Percentage labels in **d** and **e** indicate percentage of total nuclei belonging to each ROI-enriching dissection or division. NT-type, neurotransmitter type.

substantial cell damage. These results show that the SPN taxonomy is likely to provide better distinction between SPN types than the WB taxonomy.

To map SPNs to their anatomical locations, we integrated the SPN taxonomy with the AIBS WB MERFISH dataset (see below). With identified marker genes, we further verified the spatial distribution of different SPN types by performing retrograde labelling with AAVs expressing Cre-dependent H2B–GFP in 14 transgenic lines (Extended Data Fig. 9). In general, we found high correspondence between the Cre-dependent retrograde tracing and the MERFISH anatomical results.

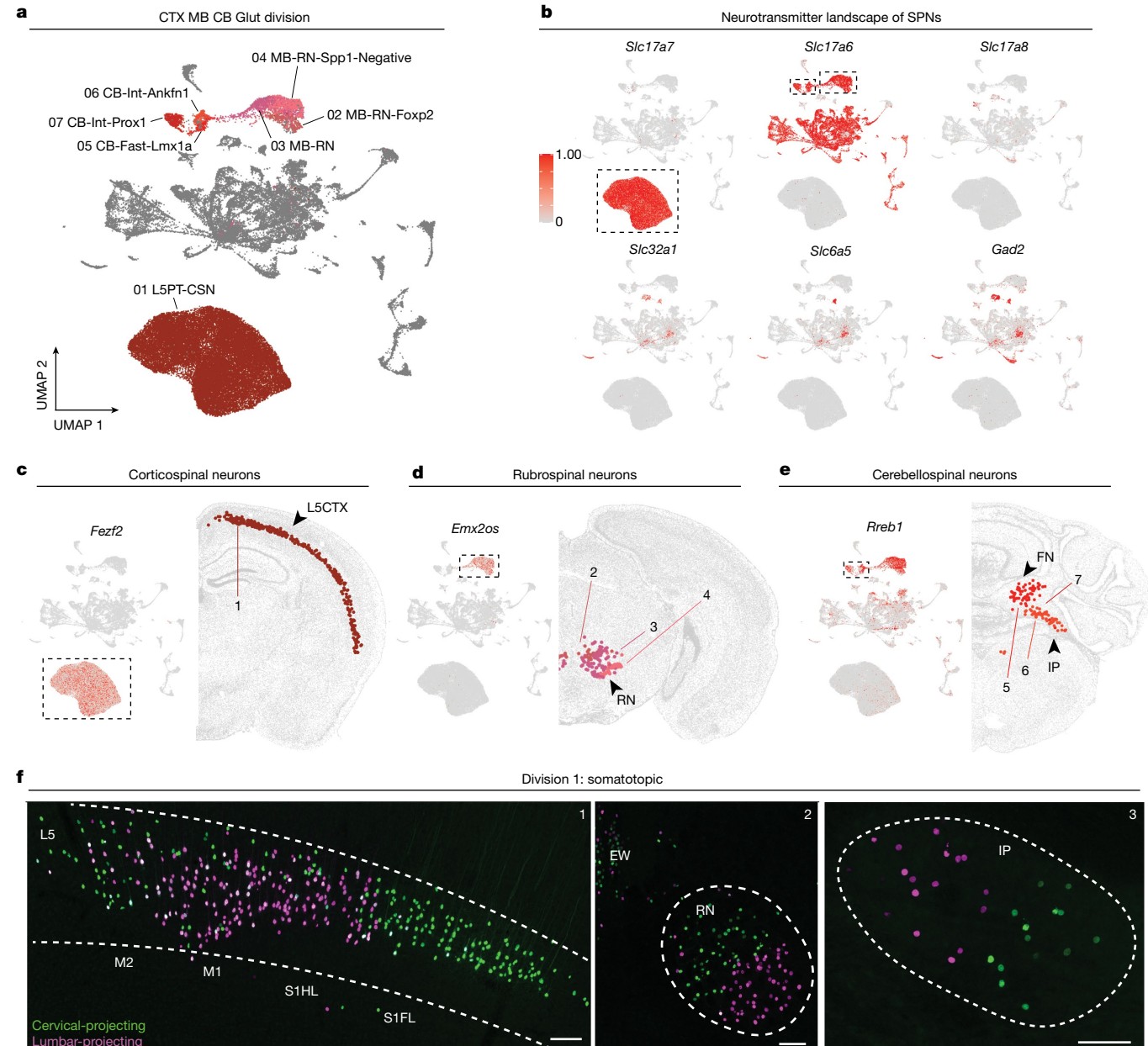

**Fig. 2 | SPNs from cortical layer 5, red nucleus and cerebellum are transcriptionally and anatomically discrete. a**, UMAP of all SPNs, coloured by types within taxonomy division 1. **b**, Expression of select genes for neurotransmission machinery. **c**–**e**, Expression of select marker genes (top) and representative MERFISH sections (bottom) of corticospinal neurons (**c**), rubrospinal neurons (**d**) and cerebellospinal neurons (**e**). Number labels depicted on the MERFISH panels correspond to SPN types in **a** and in the global dendrogram in Fig. 1c. **f**, Confocal microscopy images of retrogradely labelled cervical- (green) and lumbar- (magenta) projecting corticospinal neurons (1), rubrospinal neurons (2) and cerebellospinal neurons (3). Scale, 100 μm. Representative images shown here from *n* = 3 injected mice. Expression scale in **b** applies to all expression plots. FN, fastigial nucleus; IP, interposed nucleus; RN, red nucleus; S1FL, primary somatosensory cortex forelimb region; S1HL, primary somatosensory cortex hindlimb region.

## Transcriptomic landscape of SPNs

With both transcriptomic and spatial annotations, all SPNs could be classified into three major divisions (Fig. 1c), representing three critical components of spinal projecting pathways.

### CTX MB CB glutamatergic division

This division (division 1, types 1–7) comprises exclusively glutamatergic SPNs with concentrated anatomical locations (that is, L5CTX, MB red nucleus and deep cerebellar nuclei) and aggregated projections to the spinal cord (that is, corticospinal tract, rubrospinal tract and cerebellospinal tract). This division has three subclasses and seven types (Figs. 1c and 2a). The 'CTX glut' subclass corresponds to CSNs, which express *Slc17a7* (also known as *Vglut1*) and, as expected, other canonical layer 5 markers (for example, *Fezf2*, *Bcl11b* and *Crym*) (Fig. 2b,c). These CSNs arise in the rostral forelimb area (RFA), primary and secondary motor CTX/primary somatosensory CTX (M1M2S1), and secondary somatosensory CTX (S2) (Fig. 1b). As expected, mapping CSN snRNA-seq profiles onto AIBS WB MERFISH data localized profiles to L5CTX (Fig. 2c). In addition, CSNs within the L5PT-CSN type map to several clusters in the AIBS WB scRNA-seq taxonomy that correspond to different cortical regions, indicating that CSNs arising from different cortical regions are transcriptionally distinct (Extended Data Fig. 10 and Supplementary Tables 3–5). Differential expression analysis among CSNs from the

three CTX region-enriching dissections (M1M2S1, RFA and S2) reveal differentially expressed genes (Supplementary Table 6). The 'MB glut' subclass within this division includes three types corresponding to RuSNs. These types express *Slc17a6* (*Vglut2*) and markers *Rreb1* and *Emx2os* (Fig. 2b,d,e). Mapping their snRNA-seq profiles onto AIBS WB MERFISH data localizes these three types to the MB red nucleus (Fig. 2d). Finally, the 'CB glut' subclass with three types corresponds to CbSNs. Like RuSNs, they express *Slc17a6* (*Vglut2*) and *Rreb1* (Fig. 2b,e). CbSNs are known to arise from the fastigial and interposed deep cerebellar nuclei[20]; accordingly, we find that the SPN types CB-Int-Prox1 and CB-Int-Ankfn1 map to the interposed nucleus and CB-Fast-Lmx1a maps to the fastigial nucleus (Fig. 2e).

Another shared feature of division 1 neurons is their somatotopic locations and spinal projections. Among all brain regions, the populations in division 1 exhibit clear segregation of GFP and mScarlet cells (Fig. 2f and Extended Data Fig. 11). CSNs in RFA and S2 are entirely cervical-projecting, whereas those in M1M2S1 display a distinctive somatotopic distribution with dual- and lumbar-projecting CSNs medial to cervical-projecting CSNs. Similarly, RuSNs also display a somatotopic distribution with cervical- and lumbar-projecting SPNs originating in the dorsomedial and ventrolateral red nucleus, respectively. CbSNs from the anterior interposed nucleus also display somatotopic distribution of cervical- and lumbar-projecting populations[20]. For the small number of dual-projecting CSNs, we cannot eliminate the possibility that these neurons project only to the lumbar cord, and the AAVs injected to the cervical spinal cord label their passing axons. However, this is unlikely because AAVs similarly injected to the cervical spinal cord did not label the lumbar-projecting RuSNs and CbSNs, consistent with previous reports[11,12,20,21]. Together, common anatomical features of all division 1 neurons are concentrated regions of origin with somatotopic distribution of cervical- and lumbar-projecting types. These features render these neurons well-suited for transmitting point-to-point command signals.

## MB HB glutamatergic or GABAergic/glycinergic division

This division (division 2, types 8–56), defined by the global dendrogram with 6 subclasses and 49 types (Fig. 1c), is the most transcriptionally complex. Many types are highly related, as depicted by the global constellation plot (Extended Data Fig. 7), but can be defined by combinations of several marker genes. Most (46 of 49) types are glutamatergic (*Slc17a6*+), and a minority (3 of 49) of types are GABAergic and/or glycinergic (*Slc32a1*+ and/or *Slc6a5*+). Integration with the AIBS WB scRNA-seq and MERFISH datasets localized this widespread, transcriptionally complex division to the MB and pontomedullary reticular formation (Extended Data Fig. 12). Most types within this division were mapped throughout the pontomedullary reticular formation (Extended Data Fig. 12b–f, see 'LIM' section below). In addition, two types (MB-Meis2 and MB-Chrm3) originated from MB-enriching dissections and localized to disperse areas in the MB reticular nucleus (Extended Data Fig. 12a), a known origin of the reticulospinal tract[22]. As a result, this division corresponds to the excitatory and inhibitory ReSNs. As this is the most complex population of SPNs, the sections below provide a more in-depth analysis of ReSNs.

## Modulatory division

This division (division 3, types 57–76) contains 5 subclasses and 20 types distributed across the HY, MB and HB, but not the brain regions harbouring neurons of division 1 (Fig. 3). In contrast to the SPNs in divisions 1 and 2, which are largely defined by markers of fast-acting neurotransmission, the 'modulatory' branch is associated with neuropeptides and/or slow neurotransmitters (Extended Data Fig. 13). These modulatory inputs may amplify and/or prolong the fast commands and have been considered an 'emotional motor pathway'[23], acting in different physiological and pathological conditions[4,24,25].

Hypothalamospinal neurons primarily originate from the paraventricular HY and lateral hypothalamic area and comprise the 'HY MB Otp' and 'HY Vgll2' subclasses, respectively (Fig. 3b,c). All hypothalamospinal neurons defined here are excitatory (*Slc17a6*+). However, they are unique in their high expression of various neuropeptides such as arginine vasopressin (*Avp*), orexin (*Hcrt*), growth hormone releasing hormone (*Ghrh*) and oxytocin (*Oxt*) (Extended Data Fig. 13). In addition, type 'Otp-Th' within the 'HY MB Otp' subclass is defined by tyrosine hydroxylase (*Th*) expression (Fig. 3b). This population is primarily present in the MB-enriching dissections but also a small proportion from HY-enriching dissections, indicating that this population is present at the HY–MB border. This type is likely to be the A11 group of neurons[26] based on spatial mapping via MERFISH (Fig. 3b) and Cre-dependent retrograde labelling in a Th-Cre mouse line (Extended Data Fig. 9a). In addition to *Slc17a6*, this type expresses *Th*, dopa decarboxylase (*Ddc*) and vesicular monoamine transporter 2 (*Slc18a2*), suggesting monoaminergic activity; interestingly, however, the 'Otp-Th' type lacks expression of other monoamine transporters such as the dopamine transporter (*Slc6a3*). These results might be relevant to the notion that A11 neurons contain sufficient enzymatic machinery to generate dopamine despite not expressing *Slc6a3* (ref. 27).

The remaining four MB types in this division comprise the 'MB Mod' subclass and originate in the EW nucleus (Fig. 3d). The 'MB-EW-Cck-Ucn' type is molecularly rather atypical: these neurons are depleted for markers of excitatory or inhibitory neurotransmission and are instead enriched for a set of genes encoding neuropeptides including *Cck*, *Cartpt* and *Ucn* (Figs. 1c and 3d). By contrast, the MB-EW-Cck type expresses neuropeptides *Cck* and *Cartpt* (but not *Ucn*), and glutamatergic markers (*Slc17a6*). It is intriguing that the MB-EW-Cck-Ucn type expresses modulatory neuropeptides without detectable expression of fast-acting neurotransmitter machinery. However, the functional significance of these 'obligate neuropeptidergic'[28,29] neurons is unknown.

The PONS and MED also host modulatory SPNs. These types are in the 'HB Lmx1b Nora' and 'HB Lmx1b Sero' subclasses, corresponding to the monoaminergic locus coeruleus and medullary raphe nuclei, respectively (Fig. 3e). It should be noted that the 'HB Lmx1b Sero' subclass comprises heterogeneous neurotransmitter types, including Sero-Glut (*Slc6a4*+/*Slc17a8*+), Gaba (*Slc32a1*+) and Glut-Gaba (*Slc17a8*+/*Slc32a1*+). In addition, types in the 'HB Lmx1b Sero' subclass mapped to multiple spatial locations (Extended Data Fig. 12b), highlighting the transcriptomic and spatial heterogeneity of descending raphespinal systems. The 'HB Lmx1b Sero' types have sparse expression of *Slc17a8* (*Vglut3*), a third vesicular glutamate transporter shown to be present in other serotonergic neuronal populations[30–32]. Sparse *Slc17a8* expression is also present in other subcortical types, such as SPNs from the nucleus of the solitary tract (NTS; MED-Lhx1/5-NTS) (Fig. 3f). Though not within the 'Modulatory' division, MED-Lhx1/5-NTS highly expresses the neuropeptide glucagon (*Gcg*). As the NTS is a major parasympathetic sensory nucleus, its SPN components suggest direct supraspinal communication between visceral and spinal circuits.

Evidence of machinery for dual or partial neurotransmitter types is not unique to the 'modulatory' division. For example, the GABA/Glyc populations in the PONS and MED express *Slc6a5* (*GlyT2*) and the common vesicular transporter for GABA and glycine, *Slc32a1* (*Vgat*). In addition, the glutamatergic *Slc17a6*+ neurons in the MED-Lhx1/5-NTS type also express *Gad2* but with low or absent expression of *Slc32a1*, suggesting that they may not use GABA for synaptic transmission (Fig. 3f). This type maps to cluster 4361 in the AIBS WB atlas, which is in the NTS and intermediate reticular nucleus and similarly shows co-expression of *Slc17a6* and *Gad2* with low expression of *Slc32a1* (Extended Data Fig. 14a, b). Immunohistochemistry (IHC) for GABA confirms the presence of cytosolic GABA in *Gcg*+ cells labelled with Cre-dependent H2B-GFP in a Gcg-Cre transgenic line (Extended Data Fig. 14c,d). These results support the presence of functional *Gad2* protein (GAD65) in *Gcg*+ SPNs. The function of *Gad2* with no or low *Slc32a1* expression remains to be elucidated. In addition, it is worth noting that the data

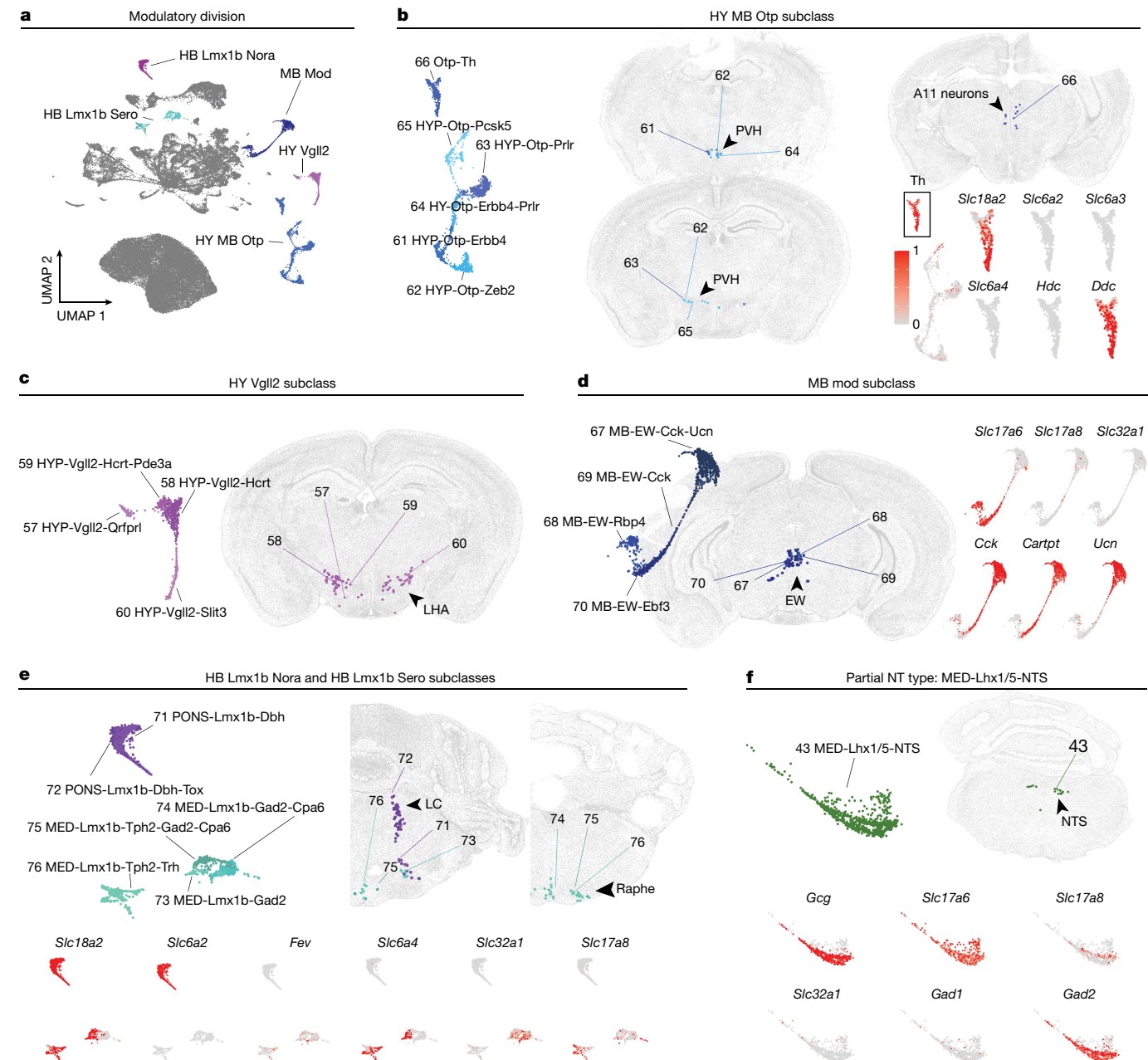

**Fig. 3 | Diverse neurotransmitter identities of SPNs. a**, UMAP of all SPNs, coloured by subclasses within taxonomy division 3. **b**, Subset UMAP (left) and MERFISH representation (right) of types within HY MB Otp subclass. Bottom right, expression of *Th* and additional monoaminergic markers. **c**, Subset UMAP (left) and MERFISH representation (right) of types within HY Vgll2 subclass.

**d**, Subset UMAP (left) and MERFISH representation (middle) of types within the MB Mod subclass. Expression of select marker genes (right). **e,f**, Same as **d** but for types in HB Lmx1b Nora and Sero subclasses (**e**) and type MED-Lhx1/5-NTS (**f**). Expression scale in **b** applies to all expression plots. Types coloured and assigned number labels as in Fig. 1c.

fail to identify adrenergic C1 neurons in the MED, relatively rare but well-documented SPNs for regulating autonomic functions[7], which might be relevant to a caveat of AAV2/retro for its low targeting efficiencies towards certain types of modulatory neurons[33].

## A LIM transcription factor code for adult ReSNs

The most anatomically and transcriptionally complex group of SPNs are ReSNs originating from the pontomedullary reticular formation, which include the majority of SPNs in division 2 and the noradrenergic and serotonergic SPNs from division 3. As the reticular formation's descending outputs towards the spinal cord, ReSNs are essential for regulating diverse functions such as voluntary movement, involuntary

postural and gait control, and autonomic functions[34–40]. Studies to date have primarily relied on *Chx10* (*Vsx2*) to characterize and target pontomedullary ReSNs[41–44], but it is unknown whether this marker covers some, or all, ReSNs. In addition, early studies have shown that ReSNs can be partitioned into unique molecularly defined subpopulations during development[45]. However, the cellular composition and organizational principles of adult ReSNs remain undetermined.

To define the 53 pontomedullary reticular formation types in the final SPN taxonomy (Fig. 1c), we subset, re-embedded and analysed nuclei from the PONS- and MED-enriching dissections together (Extended Data Fig. 15a,b). On the basis of a notable pattern of LIM homeobox gene expression across ReSNs (Supplementary Table 7 and Extended Data Fig. 15c), we classified all pontomedullary ReSNs into one of five 'LIM

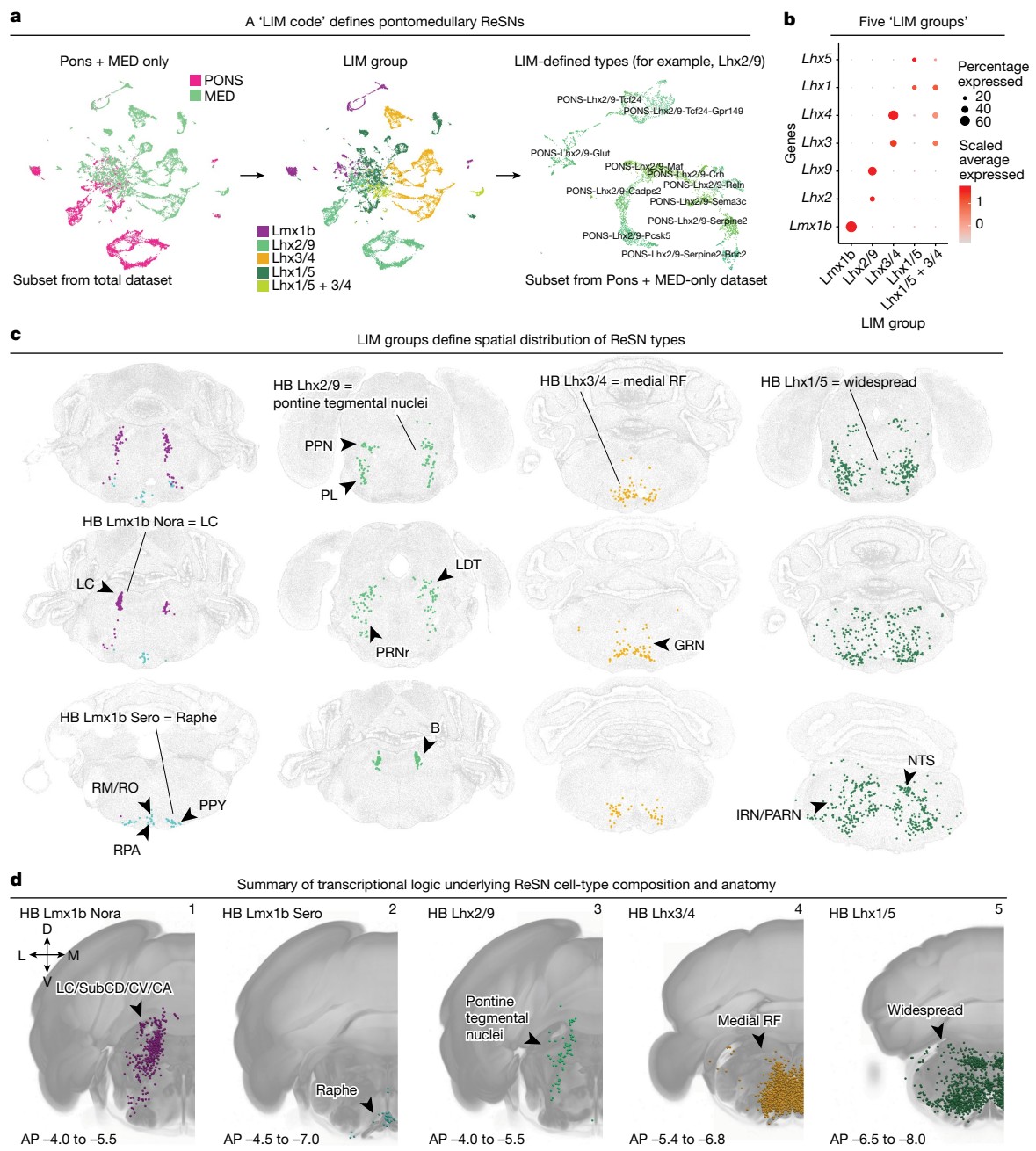

**Fig. 4 | LIM homeobox genes parcellate reticulospinal neurons into spatially and transcriptionally distinct subsets. a**, Nuclei from PONS- and MED-enriching dissections were subset and re-embedded (10x data, $n = 22,100$ nuclei). Left/middle, UMAP of nuclei from PONS- and MED-enriching dissections coloured by ROI and LIM groups, respectively. Right, each LIM group was subset and re-embedded to identify the final types depicted in Fig. 1c. Shown here is the re-embedding of the Lhx2/9 group. **b**, Dot plot showing expression of *Lmx1b*, *Lhx2*, *Lhx9*, *Lhx3*, *Lhx4*, *Lhx1* and *Lhx5* grouped by the five LIM groups. **c**, Representative MERFISH sections of ReSN types, each coloured by their respective LIM-defined subclass. **d**, Anatomical distributions of LIM-defined subclasses summarized using representative coronal reconstructions of Cre-dependent retrogradely labelled nuclei in transgenic lines corresponding to LIM-defined subclasses (from Extended Data Fig. 9). Panel 1: Dbh-Cre,

TH-Cre and Phox2b-Cre, coloured as HB Lmx1b Nora subclass. Panel 2: ePet-Cre, coloured as HB Lmx1b Sero subclass. Panel 3: Lhx2-2A-CreER, coloured as HB Lhx2/9 subclass. Panel 4: Chx10-Cre, coloured as HB Lhx3/4 subclass. Panel 5: ChAT-Cre, GlyT2-Cre, and Gcg-iCre, coloured as HB Lhx1/5 subclass. AP, anterior–posterior position (relative to Bregma); B, Barrington's nucleus; LC, locus coeruleus; LDT, laterodorsal tegmental nucleus; IRN, intermediate reticular nucleus; PARN, parvicellular reticular nucleus; PL, paralemniscal nucleus (Paxinos nomenclature[54]); PPN, pedunculopontine nucleus; PPY, parapyramidal nucleus; PRNr, pontine reticular nucleus rostral part; RF, reticular formation; RM, nucleus raphe magnus; RO, nucleus raphe obscurus; RPA, nucleus raphe pallidus; SubCD/CV/CA, subcoeruleus nucleus dorsal/ventral/alpha parts (Paxinos).

groups' (Fig. 4a and Extended Data Fig. 15d,e), which each specifically express *Lmx1b* or one pair of paralogous LIM homeobox genes (*Lhx2* and *Lhx9*, *Lhx3* and *Lhx4*, *Lhx1* and *Lhx5*) (Fig. 4b and Extended Data Fig. 15f). The *Lmx1b* group split into three subclasses based on major

neurotransmitter (HB Lmx1b Nora, HB Lmx1b Sero and HB Lmx1b Glut) in the final taxonomy (Fig. 1c). Only three types (MED-Pyy, MED-Spp1 and MED-Vcan) are not exclusively defined by a single pair of LIM homeobox genes and are classified into the Lhx1/5 + Lhx3/4 group. It should

be noted that *Chx10* (*Vsx2*), a commonly used marker for ReSNs[41–44], exhibits nearly complete expression correspondence with the Lhx3/4 group (Extended Data Fig. 15g). As a result, a Chx10-Cre mouse is likely to label only a subset of ReSNs. These results suggest the presence of a 'LIM code' for classifying adult ReSNs, perhaps reflecting the role of LIM genes in not only establishing but also maintaining neuronal identities[45]. Single-cell regulatory network inference and clustering (SCENIC) applied to pontomedullary ReSNs also identified these LIM genes among the top type-specific transcription factors (Extended Data Fig. 16).

We find that ReSN types have varying levels of transcriptomic complexity. We have demonstrated the relatedness of pontomedullary ReSN types with a constellation plot and have found that types within the Lhx1/5 group are most highly connected with other types, with type MED-Lhx1/5-Glut forming the central node (Extended Data Fig. 17a). A distinctive feature of ReSNs is the central portion of the UMAP, which resembles a 'splatter of clusters'. We have found nuclei within this central region of the UMAP to be ill-defined by specific marker genes and to contain the highest degree of heterogeneity at a single-nucleus level (Extended Data Fig. 17b). These features (that is, highly heterogeneous nuclei with subtle distinctions between cell types) are similarly present in HB neurons profiled in the AIBS scRNA-seq companion study[1]. The LIM code identified here presents a valuable framework for deciphering the transcriptional logic of ReSNs and, possibly, the reticular formation as a whole.

Integrating the ReSN types with the AIBS WB MERFISH dataset identified a striking anatomical distribution of LIM-defined types (Fig. 4c and Extended Data Fig. 12). Specifically, types within the HB Lmx1b Nora and HB Lmx1b Sero subclasses map to the locus coeruleus and medullary raphe nuclei, respectively. Types in the HB Lhx2/9 subclass correspond to SPNs arising in the pontine tegmental nuclei, such as the rostral pontine reticular nucleus, Barrington's nucleus, pedunculopontine nucleus and laterodorsal tegmental nucleus. Types in the HB Lhx3/4 subclass correspond to nuclei in the medial reticular formation such as the caudal pontine reticular nucleus and gigantocellular reticular nucleus. In line with this, MERFISH data show that *Lhx9* expression is highest in the pontine tegmentum, *Lhx1* expression is widespread throughout the HB, and *Vsx2* is expressed in the medial pontomedullary reticular formation (Extended Data Fig. 15h). The broad expression of *Lhx1* and *Lhx5* suggest that types in the HB Lhx1/5 subclass may arise in widespread anatomical domains and therefore may or may not coincide with the anatomical domains of the other LIM-defined regions. For example, the widespread pontomedullary glycinergic types are within the HB Lhx1/5 subclass. Unifying these findings with the Cre-dependent retrograde labelling results assigns the LIM groups to broad anatomical regions (Fig. 4d). Taken together, the LIM-defined types presented here comprehensively define the cellular composition of adult ReSNs and suggest a transcriptional logic underlying their heterogeneity and anatomical origin in the adult HB.

## Molecular specification of spinal targets

SPNs differ in their projection patterns at different spinal levels according to their specific functions. To inform the transcriptomic taxonomy with spinal cord projection properties, we analysed differences among SPNs projecting to the cervical and/or lumbar spinal cord. First, we analysed the projection-type composition of the 76 transcriptionally defined SPN types (Fig. 5a) by examining the distribution in UMAP space (Fig. 5b) and determining the proportion of cervical- and dual-/lumbar-projecting neurons in each SPN type (Fig. 5c). Most types are composed of SPNs with mixed projection targets, but a few types project to only the cervical spinal cord (for example, HB Lhx1/5 Glyc Penk). Types from the pontomedullary reticular formation contain high proportions of dual- and/or lumbar-projecting SPNs. Next, we investigated the transcriptional

signatures of SPNs with different spinal cord targets by differential expression analysis (Supplementary Table 8). It should be noted that *Epha4*, *Epha6*, *Epha7* and *Efna5* are among the genes that are differentially expressed between cervical- and dual-/lumbar-projecting SPNs in all ROIs (Fig. 5d,g). These guidance molecules have been suggested to regulate topographic mapping of the corticospinal tract[46], and their continued expression in adult SPNs may suggest a potential role in maintaining their respective spinal terminations. Finally, by analysing the higher sequencing depth SSv4 data, we identified a set of differentially expressed genes between cervical- and lumbar-/dual-projecting neurons in CSNs and RuSNs (Fig. 5g, and Supplementary Tabled 9 and 10) and dual- versus lumbar-projecting RuSNs (Supplementary Table 11). Select differentially expressed genes were verified by single-molecule fluorescence in situ hybridization (ISH) and the Allen ISH Atlas (Extended Data Fig. 18a–e). Gene Ontology enrichment analysis identified several relevant ontologies, including terms related to axon guidance, axon length and neurotransmission (Fig. 5h and Extended Data Fig. 18f). As a result, some of the molecular differences between cervical- and lumbar-projecting CSNs and RuSNs may be relevant to their functional tuning and structural maintenance.

## Unique properties of fast-firing SPNs

SPNs relay command signals to the spinal cord to instruct a diverse set of functions. Despite being composed of a majority glutamatergic output, it is unknown if glutamatergic SPNs differ in their electrophysiological properties. By analysing the expression of top marker and activity-related genes across SPN types, we identified a notable pattern of gene expression: *Pvalb* and *Kcng4* were expressed in most RuSNs and CbSNs and subsets of ReSNs, and these neurons strongly co-express *Spp1* (Extended Data Fig. 19a,b). *Pvalb*, *Kcng4* and *Spp1* are all associated with fast-conducting projection neurons such as alpha retinal ganglion cells[47] and fast motor neurons[48], suggesting a similar electrophysiological profile in *Pvalb*/*Kcng4*/*Spp1* positive SPNs. By focused on RuSNs, we found that *Pvalb*/*Kcng4*/*Spp1* expression is different among RuSN types: MB-RN and MB-RN-Foxp2 types have high expression, whereas MB-RN-Spp1-Negative has low expression (Fig. 6a). As expected, differential expression analysis between *Spp1* positive (*Spp1*[+]) and negative (*Spp1*[−]) RuSNs revealed that the top differentially expressed genes support electrophysiological (for example, *Pvalb*, *Kcng4*, *Kcnip4*, *Hpca*, *Kcnn3*, *Kcnc4* and *Gabrb1*) and cell-size (for example, *Spp1*, *Nefh*, *Nefm*, *Nefl* and *S100b*) differences (Fig. 6a, Extended Data Fig. 19c and Supplementary Table 12). IHC for SPP1 in retrogradely labelled tissue revealed that SPP1[−] RuSNs are primarily present in the rostral–lateral red nucleus (probably corresponding to the parvocellular red nucleus) and SPP1[+] RuSNs are abundant in the medial red nucleus (probably the magnocellular red nucleus)[11] (Fig. 6b–d).

To assess their electrophysiological features, we performed loose cell-attached and whole-cell recordings of retrogradely labelled SPP1[+] and SPP1[−] RuSNs in brain slices. Postrecording IHC (Fig. 6e) and analysis of SSv4 data reveal that somata of recorded RuSNs and number of genes per nucleus are larger in the SPP1[+] population (Fig. 6f,g). In cell-attached recordings, the negative phase of the action potential spike waveform was significantly shorter in SPP1[+] RuSNs, suggesting they have a membrane physiology that supports fast action potentials (Fig. 6h). In whole-cell recordings, we similarly found that SPP1[+] RuSNs had narrower action potentials (Fig. 6i), and a higher average spontaneous firing rate (Fig. 6j,k). In addition, SPP1[+] RuSNs had smaller input resistance and were less excitable than SPP1[−] RuSNs (Fig. 6l,m). We found no significant difference in action potential threshold, amplitude, peak or fast afterhyperpolarization (Extended Data Fig. 20a–d; representative traces and action potentials of SPP1[+] and SPP1[−] RuSNs are shown in Extended Data Fig. 20g–h). The number of cells and statistical tests are summarized in Supplementary Table 13.

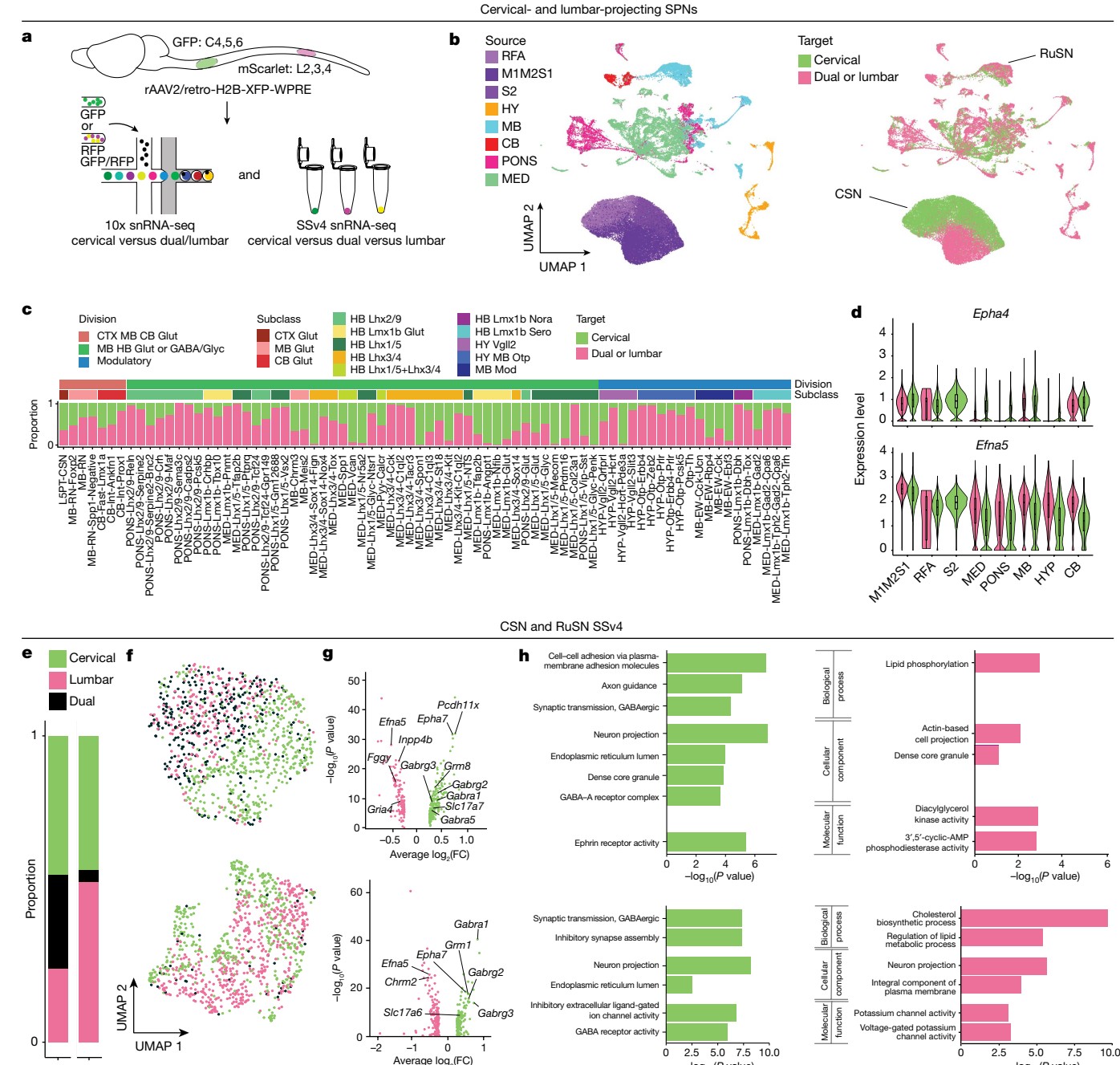

**Fig. 5 | Transcriptomic differences among SPNs terminating at different spinal targets. a**, The 10x data were generated by separately collecting cervical-projecting (GFP⁺) from dual- (GFP⁺/mScarlet⁺) and lumbar- (mScarlet⁺) projecting SPNs nuclei across the WB, and the SSv4 dataset used indexed plate-based sorting to separate cervical- (GFP⁺), lumbar- (mScarlet⁺) and dual- (GFP⁺/mScarlet⁺) projecting SPNs. **b**, UMAP visualization of all SPNs coloured by source (ROI, as in Fig. 1d) and target (spinal cord level). **c**, Proportion of cervical (green) versus dual or lumbar (pink) across SPN transcriptomic types. Bars above proportion plot indicate subclass and division. Types ordered as in Fig. 1c. **d**, Violin plot of the differentially expressed genes *Epha4* and *Efna5* across all SPNs, grouped by ROI (10x data, n = 61,484 nuclei). The centre line of the overlayed box and whisker plots depicts the median value (50th percentile)

while the box contains the 25th to 75th percentiles; the whiskers correspond to the 5th and 95th percentiles. **e**, Proportion of cervical-, dual- and lumbar-projecting neurons in the SSv4 CSN and RuSN datasets. **f**, SSv4 UMAPs of CSNs (top) and RuSNs (bottom). **g**, Volcano plots of differentially expressed genes for cervical- (green) versus dual-/lumbar- (magenta) projecting CSNs (top) and RuSNs (bottom). Genes of interest are labelled. Differential expression analysis was performed with Seurat using the MAST test; significant genes were defined as those with a false discovery rate adjusted *P* value of less than 0.05. **h**, enrichR was used to determine enriched Gene Ontology terms of differentially expressed genes in **g** in CSNs (top) and RuSNs (bottom). enrichR uses adjusted *P* values computed using the Benjamini–Hochberg method for correction for multiple hypotheses testing.

It is conceivable that fast-firing properties might be a shared feature of SPP1⁺ RuSNs and SPP1⁺ retinal ganglion cells or motor neurons. Given these larger cell sizes and fast-firing properties, these neurons may also have thicker axons for fast action potential conduction⁴⁹.

In addition, such expression and cell-size correlation also extends to CbSNs and subsets of ReSNs (Extended Data Fig. 21a–d). SPP1⁺ ReSNs primarily correspond to 'MED-Vcan' and 'MED-Spp1' types, which are likely to include vestibulospinal neurons, as indicated by MERFISH

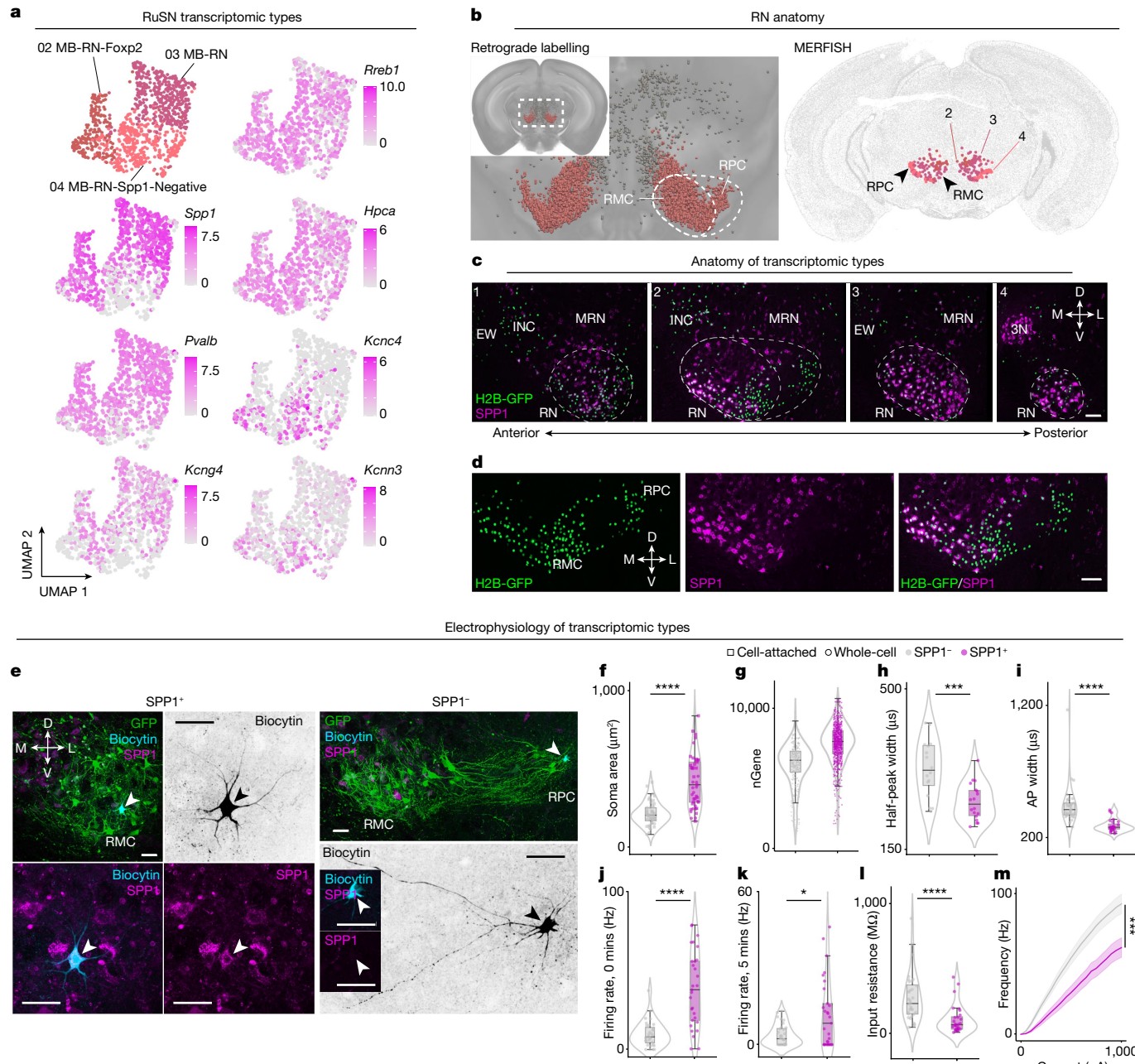

**Fig. 6 | Anatomy, morphology and electrophysiology of rubrospinal types.**
**a**, UMAP of RuSNs (SSv4, *n* = 1,031 nuclei) coloured by type (top left) and expression of general RuSN marker gene (*Rreb1*, top right). Expression of differentially expressed genes for SPP1⁺ (*Spp1*, *Pvalb*, *Kcng4*; left, bottom three) and SPP1⁻ (*Hpca*, *Kcnc4*, *Kcnn3*; right, bottom three) RuSNs. **b**, Left, coronal reconstruction of RuSNs imaged with STPT. Right, MERFISH mapping depicting types 02–04 (same as Fig. 2). **c**, Representative IHC results for SPP1 on retrogradely labelled RuSNs. **d**, Expanded view of panel 2 in **c**. Scale, 100 μm. *n* = 3 brains stained to yield representative results. **e**, Confocal images of representative SPP1+ and SPP1− RuSNs in slices from cell-attached recordings. Biocytin was injected into recorded cells and labelled with fluorophore-conjugated streptavidin. Scale, 50 μm. **f**, Quantification of soma size from cell-attached and whole-cell recordings. **g**, Number of genes detected in RuSNs

(SSv4, *n* = 1,031 nuclei). **h**, Half-peak width of spike waveforms from cell-attached recordings. **i–m**, Action potential width (**i**), spontaneous firing rate at 0 min (**j**) and 5 min (**k**), input resistance (**l**), and frequency–current injection (**m**) from whole-cell recordings. ****$P < 0.0001$, ***$P < 0.001$, **$P < 0.01$, *$P < 0.05$, Mann–Whitney test (two-sided) (**f**,**h**–**l**); generalized linear mixed effect model (two-sided) (**m**). In **f**–**l**, the centre line of the box plot depicts the median value while the box contains the 25th–75th percentiles; whiskers correspond to the 5th and 95th percentiles. In **m**, the solid line shows mean firing frequency and shaded area shows s.e.m. Number of cells and statistical tests are summarized in Supplementary Table 13. 3N, cranial nerve 3; INC, interstitial nucleus of Cajal; MRN, MB reticular nucleus; RMC, magnocellular red nucleus; RPC, parvocellular red nucleus.

mapping and the Allen ISH Atlas (Extended Data Figs. 12f and 21e). Together, these results suggest transcriptional signatures of SPNs with large soma size, which correlate with fast-firing electrophysiological properties, pointing to different physiological channels in relaying brain-derived commands.

## Discussion

In this study, we created a comprehensive transcriptomic atlas of mouse SPNs by profiling 65,002 SPN nuclei defined by AAV-assisted retrograde labelling from different spinal cord levels. Multilevel iterative

clustering, in combination with spatial characterization by MERFISH and other methods, classified SPNs into molecularly and functionally distinct groups with different spatial distributions (summarized in Extended Data Fig. 22). From this spatially resolved transcriptomic atlas, we identified a LIM homeobox transcription factor code that partitions the transcriptionally heterogeneous ReSNs into five molecularly distinct and spatially segregated populations. In addition, molecular signature-guided electrophysiological studies suggested functionally distinct SPN populations differing in their firing and possibly conduction properties. Taken together, these results begin to reveal the organizational principles for the brain–spinal cord highways that turn 'thoughts' into 'actions'.

How brain commands are organized to control spinal cord function is a fundamental question in neuroscience. Here our unbiased computational analyses suggest that the entire spinal projecting system consists of three distinct yet complementary components (or 'divisions'). The first division includes the neurons in three discrete brain regions: CTX, red nucleus and CB. They are exclusively excitatory and relatively transcriptionally homogeneous, suggesting that their functional specifications might be mainly determined by their synaptic partners. In light of their somatotopic spinal terminations, these SPNs might be important for transmitting specialized point-to-point command signals to the spinal cord, consistent with their known function in controlling independent movement of the extremities[20,50,51]. By contrast, SPNs in the second division include both excitatory and inhibitory neurons and are highly heterogeneous. These neurons are widely distributed in the reticular formation and represent a major population of ReSNs. Together with their innervation throughout the spinal cord, these neurons are well suited for relaying commands related to different (sensory/motor/autonomic and excitatory/inhibitory) activities and patterns of the entire spinal cord and might be relevant to the role of the reticulospinal tract in coordinating different aspects of gait and posture control[34–40]. Complementary to the fast-acting neurons in divisions 1 and 2, division 3 corresponds to modulatory SPNs which express slow-acting neurotransmitters and/or neuropeptides. These neurons are distributed in the HY, MB and HB, but not in the CTX, red nucleus and CB, where division 1 SPNs arise. Their broad spinal terminations and neuromodulatory capacity render them as a potentially powerful 'gain setting' system. Remarkably, the three divisions defined by our unbiased computational analyses correspond to Kuypers'[8] and Holstege's[23] theory of motor control, namely the lateral pathway for finer appendicular motor control (division 1), the medial pathway organizing postural movements (division 2) and the emotional motor system (division 3). As a result, our analyses suggest a molecular basis for the organization of descending pathways predicted from classic anatomical and functional observations across different species[4,8].

Previous efforts on SPNs have primarily focused on CSNs[4–6,16,52], leaving ReSNs and their hosting pontomedullary reticular formation poorly characterized. The reticular formation is well known for its anatomical and functional complexity[34–40]. We show here that all pontomedullary ReSNs can be defined by two complementary components: ReSNs in division 2 are defined by fast-acting (excitatory or inhibitory) neurotransmitters and those in division 3 are modulatory, suggesting that the reticular formation elaborates both fast and slow command signals to the spinal cord. In addition, several LIM transcription factors assign most ReSNs into five groups with complementary spatial distribution. It should be noted, the Lhx3/4 and Lhx1/5 groups are associated with two major forms of structures (that is, concentrated nuclei versus a distributed network) of the reticular formation. Specifically, SPNs in the Lhx3/4 group are mainly concentrated in several distinct nuclei in the medial pontomedullary reticular formation, whereas SPNs in the Lhx1/5 group are sparsely and/or widely distributed throughout the brainstem. Like the 'LIM code' presented here for ReSNs, results from the companion AIBS WB paper suggest that transcription factors form a combinatorial code that defines cell types across the WB[1]. Taken together, this atlas provides a framework for how to probe into this transcriptionally complex population of ReSNs.

Despite the apparent organization of brain–spinal commands into three divisions, it remains unclear how signal transmission is organized into individual components. For example, are command signals coded by the same types of axon (that is, volume-based) and/or different types of axon (that is, labelled lines)? Our results demonstrate an example of different functional pathways. Among excitatory SPNs, most RuSNs and CbSNs and subsets of ReSNs express a set of genes, such as *Pvalb*, *Kcng4* and *Spp1*, characteristic of fast-spiking and fast-conducting neurons in subsets of retinal ganglion cells[47] and spinal motor neurons[48]. The electrophysiological studies here provide further support of this prediction. In addition to the diversity of neurotransmitters and neuropeptides of SPNs, at least two different pathways (namely, *Pvalb*/*Kcng4*/*Spp1* positive and negative) with different electrophysiological properties might underlie the transmission of brain signals to the spinal cord.

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

## Methods

### Animals
All experimental procedures were performed in compliance with animal protocols approved by the Institutional Animal Care and Use Committee at Boston Children's Hospital (Protocol no. 20-05-4165 R). Mice were provided with food and water ad libitum, housed on a 12-hour light/dark schedule (7 a.m.–7 p.m. light period) with no more than five mice of the same sex per cage, and allowed to acclimate for 1 week after arrival. C57BL/6J wild-type mice (Jackson Labs, Strain no 000664) were used for all snRNA-seq and electrophysiological experiments, and to generate the wild-type retrogradely labelled histological datasets. Transgenic driver lines (Supplementary Table 14) were used for the histological Cre-dependent retrograde labelling datasets (that is, those generated via retrograde labelling with AAVs expressing Cre-dependent GFP). Nucleus suspensions were generated from 15 adult mice (8 female and 7 male) for 10x and 15 mice (6 female and 9 male) for SSv4 (Supplementary Table 1). An equal number of male and female mice were used for histological and electrophysiological studies.

### Retrograde labelling and imaging of adult SPNs
**Retrograde labelling.** To retrogradely label SPNs, AAV2/retro-Syn-H2B-GFP and AAV2/retro-Syn-H2B-mScarlet were injected (produced by Boston Children's Hospital Viral Core). Adult mice (P42) were anaesthetized with intraperitoneal injection of ketamine-xylazine (KX, 100–120 mg kg$^{-1}$ ketamine, 10 mg kg$^{-1}$ xylazine) and placed on a stereotactic frame. For cervical level (C4–6) injection, a dorsal incision was made along the vertebral column at cervical level. The overlying muscle was dissected to expose the cervical vertebral column and a laminectomy was performed at C4–6 to expose the spinal cord. Then, 2.4 µl of AAV2/retro-Syn-H2B-GFP ($5 \times 10^{12}$ genome copies per ml (gc ml$^{-1}$)) were injected at 0.40 mm lateral to midline and 0.60 mm and 1.00 mm depth using a glass micropipette (tipped at 25 µm in diameter). The dorsal muscle layers were sutured, and skin closed with sterile wound clips. For lumbar level (L2–4) injections, this process was repeated with the AAV2/retro-Syn-H2B-mScarlet ($5 \times 10^{12}$ gc ml$^{-1}$) virus at 0.35 mm lateral to midline and 0.60 mm and 1.00 mm depth. AAV2/2-H2B, which does not have retrograde labelling capacity, was injected using the same viral titre, volume and coordinates to assess spread of virus at injection sites.

**Histology and imaging.** At P56, retrogradely labelled mice were anaesthetized with KX and euthanized by transcardiac perfusion with phosphate buffered saline (PBS) followed by 4% paraformaldehyde (PFA) in PBS. The skull and spinal column were dissected and postfixed in 4% PFA in PBS for 24 h. The brain and spinal cord tissue were then microdissected and cryoprotected in 30% sucrose, embedded and frozen in optimal cutting temperature medium and processed using a cryostat (brain: 50 µm coronal sections, olfactory bulb through caudal MED, four series; spinal cord: 40 µm coronal or dorsal–ventral horizontal sections, four series). Sections were washed with PBS and then treated with a blocking solution (5% normal donkey serum, 0.5% Triton-X) for 1 h at room temperature. Floating sections were incubated in primary antibodies (AVES chicken-anti-GFP, Rockland rabbit-anti-red fluorescent protein (RFP); 1:500 dilution in blocking solution) at 4 °C overnight, washed three times for 10 min with PBS, and subsequently incubated with secondary antibodies (Alexa Fluor 488 donkey-anti-chicken, Alexa Fluor 594 donkey-anti-rabbit; 1:500 dilution in PBS) for 2 h at room temperature. Finally, sections were washed with PBS, mounted with DAPI Fluoromount-G mounting medium, sealed with clear nail polish and stored at 4 °C until imaging. For GABA staining, 30 µm coronal sections were used and incubated with rabbit-anti-GABA (Sigma A2052, 1:1000) for 20 h at room temperature. Whole sections were imaged on an Olympus VS120 Virtual Slide Microscope at ×10 magnification. Higher magnification images were acquired as Z-stacks with a ×20 or ×63 objective on a Zeiss LSM 900 Confocal Laser Scanning Microscope (Zen 3.3 software).

### Anterograde labelling and imaging of adult SPNs
**Labelling of Ucn$^{+}$ EW neurons.** Three 9-week-old Ucn-Cre mice (two males and one female) were anaesthetized with 5% isoflurane in 100% O$_2$ using a precision vaporizer (VetEquip) and placed on a stereotactic frame. A surgical plane of anaesthesia was maintained with 1.5–2% isoflurane in 100% O$_2$ throughout the duration of the procedure. A 1.5 cm transverse incision was made to reveal the skull, and the skin and underlying tissue were retracted with sterile cotton swabs. A ×40 centring scope (Kopf Instruments) was used to level the skull laterally and on the rostro–caudal axis so that lambda and bregma were in the same plane. To reveal the brains surface, a 2-mm-diameter hole was drilled into the skull using a carbon steel burr (Fine Science Tools, 0.5 mm tip diameter) attached to a high-speed rotary micromotor kit (Foredom) and centred 3.5 mm caudal to bregma (antero–posterior (AP) −3.5 mm, medial–lateral (ML) 0 mm). A beveled (30°) borosilicate glass pipette attached to an injection pump (World Precision Instruments, Nanoliter 2000) was positioned to the same coordinate using a three-axis motorized manipulator (Scientifica, IVM Triple). The pipette descended at 10 µm per sec, and 100 nl of AAV2/9-CAG-Flex-ChR2-TdTomato ($5 \times 10^{-12}$ gc ml$^{-1}$) was injected (50 nl min$^{-1}$) at two depths, dorsal–ventral (DV) −3.5 mm and −3.1 mm. The pipette remained at each DV depth for 5 min after injection before being retracted.

**Labelling of Spp1+ RuSNs.** Spp1-Cre mice at 4–6 weeks old received cervical and lumbar spinal cord injections of AAV2/Retro-Flex-Flpo as mentioned above (Addgene no. 387306, virus produced by Boston Children's Hospital Viral Core, $5 \times 10^{12}$ gc ml$^{-1}$). One week later, these mice were head-fixed on a stereotactic frame for brain injection. Craniotomy was performed to expose brain surface centring at AP −3.8 mm, ML −0.8 mm. Then, 100 µl of AAV2/8-Syn-Flpx-rc[ChrimsonR-tdTomato] (Addgene np. 128589, virus produced by Boston Children's Hospital Viral Core, $5 \times 10^{12}$ gc ml$^{-1}$) was slowly injected into DV −4.0 mm below dura.

**Histology, imaging and generation of axonal contour plots.** Four weeks after local injection, mice were anaesthetized, euthanized by transcardiac perfusion with PBS and 4% PFA, and the brain and spinal cord tissue were prepared for cryosectioning as described above (brain: 50 µm coronal sections; spinal cord: 40 µm coronal sections, four series). IHC was performed as described above, with rabbit-anti-RFP (Abcam ab34771; 1:1,000) and goat-anti-ChAT (Millipore AP144P; 1:100) primary antibodies, and corresponding secondary antibodies (Alexa Fluor 594 donkey-anti-rabbit, Alexa Fluor 647 donkey-anti-goat 1:1,000). Sections were acquired as Z-stacks with a ×20 objective on a Zeiss LSM 900 Confocal Laser Scanning Microscope. To create contour plots to depict axonal density, representative spinal cord axonal projection images were first converted to 8-bit black–white TIF format and then loaded into MATLAB. After background subtraction with 'strel' function, the images were binarized with a threshold of 120 arbitrary units and only signals with more than two connected pixels were kept. The images were divided into 20-by-20 bins and the number of above-threshold pixels within each grid were counted. The count pixel density was then smoothed with 100-by-100 bins. Contour plots were generated from the smoothed results and then overlaid on the inverted spinal cord image.

### Quantification of SPNs across major brain regions
SPNs were quantified across the eight major brain regions (RFA, M1M2S1, S2, HY, MB, PONS and MED) using QuPath (v.0.4.1). To summarize, VS120 slide-scanner images of one (of four) IHC-processed coronal series were loaded into QuPath as a project. The polygon tool was used to manually annotate ROIs. Classifiers were then created using 'Positive Cell Detection' (for GFP and mScarlet nuclei) and 'Composite

Classifier' (for GFP⁺/mScarlet⁺ nuclei). 'Annotation results' were saved per ROI, and the number of nuclei quantified per ROI were multiplied by four (as only one of four coronal series was quantified) to get total brain counts. $n = 5$ retrogradely labelled brains were quantified to yield total SPN counts across each of the eight ROIs ($n = 3$ labelled with GFP into cervical cord and RFP into lumbar cord; $n = 2$ with GFP into both cervical and lumbar cord).

### Measurement of virus-mediated fluorescent protein expression

QuPath was used to measure the fluorescence intensity of GFP and mScarlet labelled nuclei in cortical layer 5 versus non-layer 5 of retrogradely labelled tissue (without IHC to maintain unamplified fluorescence levels). As described above, the polygon tool was used to manually annotate cortical layer 5, upper CTX (layers 1–4), and lower CTX (layer 6), and 'Positive Cell Detection' and 'Composite Classifiers' were run to detect GFP⁺, mScarlet⁺ and GFP⁺/mScarlet⁺ nuclei. 'Detection results' were saved per ROI and fluorescence intensity was extracted from the 'Nucleus.GFP.mean' and 'Nucleus.mScarlet.mean' results. Statistical significance was assessed with the Wilcoxon test (ggpubr).

### STPT, registration and nucleus segmentation

**Image acquisition.** Whole retrogradely labelled brains were imaged using the TissueCyte STPT platform. At P56, retrogradely labelled mice were anaesthetized with KX and euthanized by transcardiac perfusion with PBS and 4% PFA/PBS. The skull and spinal column were dissected and postfixed in 4% PFA in PBS for 24 h. The brain was then microdissected, rinsed with PBS and embedded in covalent agarose. A 4.5% agarose solution in 10 mM NaIO₄ was made by gently stirring agarose and NaIO₄ with phosphate buffer (PB; 0.42 g l⁻¹ monobasic sodium phosphate, 0.92 g l⁻¹ dibasic sodium phosphate) for 2–3 h at room temperature in a hood protected from light and filtered with a 0.2 µm filter. A microwave was then used to bring agarose to a boil, then cooled down to 60–65 °C to embed the brain. The agarose-embedded brain was mounted in the TissueCyte using Quick Bond adhesive. The sample was immersed in PB during image acquisition. Images were captured at $XY$ resolution of 1.38 µm per pixel with 10 µm stacks and 50 µm vibratome planes. Images were stitched with TissueVision's proprietary stitching algorithm 'Stitcher'.

**Registration and segmentation.** Labelled nuclei were segmented and reconstructed using NeuroInfo (MBF Bioscience, v.2023-1-1). Stitched images were first stacked with MicroFile+. Stacked images were then registered to the Allen Mouse Common Coordinate Framework (Allen CFF) with the volume registration function in NeuroInfo. H2B signals in red (mScarlet) or green (GFP) channels were detected and labelled separately. The detected nuclei were manually annotated with different colours based on their registered regions. Videos and horizontal, coronal or sagittal projections of the three-dimensional (3D) reconstructions were output via Neuroinfo's 3D visualization window.

### snRNA-seq

**Brain preparation.** Brains were prepared by adapting a protocol from the AIBS available at protocols.io (https://doi.org/10.17504/protocols.io.bq6wmzfe). At P56, retrogradely labelled mice were anaesthetized with isoflurane and transcardially perfused with 25 ml of ice-cold oxygenated artificial cerebrospinal fluid (aCSF-O2; 0.5 mM calcium chloride dihydrate, 25 mM D-glucose 20 mM HEPES, 10 mM magnesium sulfate heptahydrate, 1.25 mM sodium phosphate monobasic monohydrate, 2.5 mM potassium chloride, 3 mM sodium pyruvate, 5 mM sodium L-ascorbate, 25 mM sodium bicarbonate, 2 mM thiourea, 96 mM HCl, 96 mM N-methyl-D-glucamine, 3 mM myo-inositol, 12 mM N-acetyl-L-cysteine, 0.01 mM taurine). The brain was then rapidly dissected and placed in an acrylic brain matrix in a bath of aCSF-O₂. For 10x samples, brains were sliced with a 1 mm coronal matrix until

−3 mm (relative to Bregma) to capture forebrain ROIs. Then the brain was transferred to a 1 mm sagittal matrix for MB and HB to allow for comprehensive dissection of these ROIs. For SSv4, a 1 mm coronal matrix was used to section the entire brain to allow for finer dissections of ROIs. Slices were then transferred into a Sylgard-coated dissection dish containing ice-cold aCSF-O₂ with 0.0132 M trehalose. Brightfield and fluorescent images were taken of the intact tissue to document ROIs with a Zeiss SteREO Discovery Dissecting Microscope and subsequently microdissected with a needle blade microknife. ROI dissection was guided by visualization of fluorescently labelled regions in the eyepiece. Brightfield and fluorescent images were acquired of the dissected ROI for verification. For 10x, target ROIs were RFA, M1M2S1, S2, HY (coronal slices), MB, CB, PONs and MED (sagittal slices). Tissue was frozen between collection and nucleus isolation to allow pooling of ROIs from multiple animals to collect sufficient numbers of nuclei for the 10x platform. ROIs were placed on a razor blade and Whatman filter paper was used to absorb excess CSF. The razor blade was then held in liquid nitrogen vapour until the tissue froze. Frozen tissue was transferred into a microcentrifuge tube with a frozen moisture reservoir of optimal cutting temperature medium to prevent frost damage and stored at −80 °C until use. For SSv4, targeted dissections were made of cortical layer 5, red nucleus, pontine reticular nucleus and GRN. The dissected regions were not frozen and instead transferred into aCSF-trehalose in a microcentrifuge tube and stored on ice until nucleus isolation upon completion of dissections.

**Nucleus isolation.** Single nuclei were isolated by adapting a protocol from the AIBS available at protocols.io (https://doi.org/10.17504/protocols.io.bq7emzje). Microdissected regions were transferred into homogenization buffer (10 mM Tris pH 8.0, 250 mM sucrose, 25 mM KCl, 5 mM MgCl₂, 0.1% Triton-X 100, 0.5% RNAsin Plus, 1× protease inhibitor, and 0.1 mM DTT) and placed into a 1 ml dounce homogenizer on ice. For 10x samples, frozen tissue from multiple animals was pooled per ROI for each replicate (Extended Data Fig. 3b). For SSv4 samples, each animal and ROI was maintained as a separate sample. Tissue was homogenized with 15–20 strokes of the loose pestle followed by 15–20 strokes of the tight pestle. Subcortical regions with more white matter tracts require a larger number of strokes to homogenize tissue, though unhomogenized pieces of white matter remain following douncing. An additional 1 ml of homogenization buffer was added to the dounce and 1 ml of homogenization buffer was used to wet a 30 µm cell strainer. Half of the nucleus suspension was passed through the cell strainer into a 15 ml conical tube to remove large debris. Another 1 ml of homogenization buffer was added to the dounce and then all the nucleus suspension was passed through the cell strainer into a 15 ml conical tube. A final 2 ml of homogenization buffer was added to the dounce to rinse any remaining nuclei into suspension and then passed through the strainer for a total of 6 ml homogenization buffer–nucleus suspension. The homogenate was spun at 900g for 10 min in a 4 °C swinging bucket centrifuge. After centrifugation, supernatant was removed without disturbing the visible pellet at the bottom of the conical tube and the pellet was resuspended in 0.5–1.0 ml of sort buffer (PBS with 0.8% BSA and 0.5% RNAsin Plus). Finally, 4'-6-diamidino-2-phenylindole (DAPI) was added to the nucleus suspensions at a final concentration of 0.1 µg ml⁻¹ and incubated on ice until FANS.

**Spinal projecting neuron enrichment via FANS.** SPNs were enriched from the nucleus suspensions using FANS. For 10x, single SPN nuclei were sorted using a BD FACSARIA II with a 70 µm custom pressure nozzle (50 psi). Single nuclei were captured by sorting on 'four-way-purity mode' and gating on DAPI-positive while excluding debris and aggregates, then gating on GFP and/or mScarlet signal (Extended Data Fig. 3d). Nuclei were sorted (to a maximum number of 16,000 nuclei) using two-way sorting (GFP-positive separate from mScarlet-positive and double-positive) into polymerase chain reaction (PCR) tubes

(precoated with 5% BSA overnight) containing 20 μl of sort buffer. After sorting, PCR tubes were briefly centrifuged and then placed on ice until proceeding with the 10x platform. For SSv4, single nuclei were sorted using a Sony SH800 Cell Sorter or MA900 Multi-Application Cell Sorter using a 100 μm chip. Single nuclei were sorted on 'single-cell' mode and captured by gating on singlet DAPI-positive, then gating on GFP and/or mScarlet signal to specifically sort cervical and/or lumbar project SPNs, respectively. Indexed plate-based sorting was used to sort GFP-positive, mScarlet-positive and double-positive nuclei into strip tubes containing 11.5 μl of SMART-Seq v4 lysis buffer. Lysed FANS-sorted nuclei were then briefly centrifuged, frozen on dry ice and stored at −80 °C until proceeding with the SSv4 platform.

**cDNA amplification and library construction.** The processing of the 10x dataset was performed using the Chromium Next GEM Single-Cell 3' Kit v3.1 (1000268, 10x Genomics). To optimize yield, sorted nucleus suspensions were loaded directly into the 10x Chromium Controller without post-FANS spin down and re-suspension. Reverse transcription, cDNA amplification and library construction were performed according to the manufacturer's protocol. Libraries were sequenced on an Illumina NovaSeq 6000, targeted at a sequencing depth of 120,000 reads per nucleus. Sequencing reads were aligned to the mouse reference transcriptome (mm10, version 2020-A) using CellRanger (v.6.1.2).

SSv4 processing was performed according to previously established procedures[55] available at protocols.io (https://doi.org/10.17504/protocols.io.8epv517xdl1b/v2). The SSv4 Ultra Low Input RNA Kit for Sequencing (634894, Takara) was used to reverse transcribe poly(A) RNA and amplify full-length cDNA. Samples were amplified for 16–21 cycles in eight-well strips. Library preparation was then performed using Nextera XT DNA Library Preparation (FC-131-1096, Illumina) with a custom index set (Integrated DNA Technologies) according to the manufacturer's instructions with modifications to reduce the volumes of all reagents and cDNA input to 0.2× of the original protocol. Libraries were sequenced on either an Illumina NovaSeqSP-XP or an Illumina NextSeq2000, targeting 500,000 reads per nucleus. Samples were aligned to Mouse reference, mm10/genecode.vM23, using STAR (v.2.7.1a).

**snRNA-seq preprocessing, quality control and removal of second-order nuclei.** Ambient RNA contamination was removed from each sample using CellBender[56] (v.0.2.1, 'remove-background', default parameters). We then used Seurat v.4.3.0 for downstream analysis[57]. First, each replicate from the same ROI-enriching dissection was merged using Seurat's 'merge' function (merge.data = TRUE) and then the percentage of counts originating from mitochondrial RNA per nucleus was calculated. Nuclei were filtered to retain only those with less than 5% mitochondrial counts and more than 2,000 genes detected. Standard processing for each sample was performed, which consisted of normalization of feature expression (NormalizeData), identifying the most variable genes with FindVariableFeatures (selection.method = 'vst', nfeatures = 2,000) and scaling expression values (ScaleData). We then performed principal component analysis (RunPCA) with 30 components (FindNeighbors) and clustered the nuclei using the Louvain algorithm with resolution set at 0.5, 1, 2 and 3 to obtain a spectrum of coarse to fine clusters (FindClusters). Finally, UMAP embedding was performed (RunUMAP).

Nuclei were then assessed for second-order labelling across each ROI. GFP or mScarlet sequences were added to the mouse reference genome mm10-2020-A by mkref function from CellRanger (v.5.0.1, on UCLA's server). The reference genomes with GFP or mScarlet sequence were uploaded to the 10x cloud analysis server, and the transcriptome counts including GFP or mScarlet were detected by CellRanger (v.6.1.2). As the GFP and mScarlet sequences are 93.60% similar, to increase the detection sensitivity and specificity, we used reference genome with GFP sequence to detect samples sorted on GFP signal and used

reference genome with mScarlet sequence to detect samples sorted on mScarlet signal. The nuclei with at least one GFP or mScarlet count from the raw count matrix were considered as XFP+ nuclei, and the percentage of XFP+ nuclei was calculated within each cluster. In general, a 10% cut off threshold was used to assess first- and second-order labelling. The PONS and MED were pooled as 'hindbrain' for this analysis because they are clustered together in downstream analyses.

**Clustering and type assignment.** SPN types were defined iteratively through multiple rounds of clustering of the 10x data. After clusters containing less than 10% of nuclei expressing XFP were removed, each Seurat object consisting of replicates from the same ROI-enriching dissection underwent the standard Seurat v.4 workflow (normalization, variable feature identification, principal component analysis, Louvain clustering and UMAP embedding, as described above). Again, nuclei were clustered with resolution set at 0.5, 1, 2 and 3 to obtain a spectrum of coarse to fine clusters. Differential expression analysis was performed with model-based analysis of single-cell transcriptomes (MAST)[58] using Seurat's FindAllMarkers function (adjusted $P < 0.05$) at the various resolutions. At these various resolutions, we removed putative doublets by identifying clusters expressing genes consistent with more than one cell type (for example, clusters co-expressing neuron-glial markers, or layer 5 and non-layer 5 markers; estimated multiplet rate of each 10x sample shown in Supplementary Table 15). We reviewed the resulting clusters and their markers and assigned type identity. In dissections from cortical layer 5, we merged several clusters based on shared expression of canonical marker genes (for example, *Fezf2*, *Bcl11b*, *Crym*) and identified and removed artefactual clusters with the top differentially expressed genes that were mitochondrial and long non-coding RNAs. We then merged Seurat objects from bordering regions (that is, HY and MB; MB and PONS; PONS and MED) and re-ran the Seurat v.4 workflow to identify and merge duplicate types for those that are present across ROI-enriching dissections. At this point, clusters in RFA, M1M2S1, S2, HY, MB and CB were well-resolved by marker genes but several clusters in PONS- and MED-enriching dissections were ill-defined. To define the final PONS and MED types in the SPN taxonomy (Fig. 1c), we re-ran the Seurat v.4 workflow on nuclei from PONS and MED together (Extended Data Fig. 15). Based on differential expression analysis of these clusters at the various resolutions, we identified a pattern of LIM homeobox gene expression across nuclei from PONS and MED. We again subset and re-ran the Seurat v.4 workflow on each of these five LIM-defined groups (that is, Lmx1b, Lhx2/9, Lhx3/4, Lhx1/5 and Lhx1/5+Lhx3/4 (Extended Data Fig. 15d–f)).

This iterative annotation process identified 76 types across all SPNs. Seurat's FindAllMarkers (MAST test) was used to find the marker genes of the 76 SPN types. We tested only genes that were detected in a minimum of 25% of the nuclei (min.pct = 0.25) and that showed at least a 0.25-fold log-scale difference (logfc.threshold = 0.25) between the nuclei in type and all other nuclei. Genes are annotated as 'significant' if the adjusted $P$ value is less than 0.05 (Supplementary Table 2).

**Joint clustering of 10x and SSv4 datasets.** Types identified in the 10x dataset with the above iterative annotation process were then identified in the SSv4 data. SSv4 data was processed with the Seurat v.4 workflow and differential expression of clusters was performed. Clusters were assigned to one of the 76 SPN types based on the top differentially expressed genes. Subsequently, all 10x and SSv4 data were integrated together. To do this, all 10x Seurat objects from ROI-enriching dissections were merged using Seurat's 'merge' function (merge.data = TRUE) and the Seurat v.4 workflow was run with SCTransform using 25 principal components. In parallel, all SSv4 Seurat objects from ROI-enriching dissections were similarly merged and the Seurat v.4 workflow with SCTransform was run using 30 principal components. The 10x and SSv4 data were then integrated using Seurat's integration pipeline.

Integration features were selected using Seurat's 'SelectIntegration-Features function (nfeatures = 3000). SCT-based integration was then run (PrepSCTIntegration, FindIntegrationAnchors, IntegrateData). Finally, the Seurat v.4 workflow was run for a final time as above (RunPCA, FindNeighbors, FindClusters, RunUMAP) with 30 components and 0.5 resolution.

**Constructing the cell-type taxonomy tree.** The SPN-type taxonomy tree of all SPNs in the 10x and SSv4 datasets was built with the build_dend function in the scratch.bigcat package, as described in the companion study[1]. We calculated the average expression of 1,871 marker genes at the 'type' level and used this information to build the tree. To gain a comprehensive understanding of the cell-type landscape and relationships between cell types, we considered the taxonomy tree, the UMAPs and the constellation plot.

**Assigning division, subclass and type names.** All SPNs were organized into a hierarchy with three levels (divisions, subclasses and types) based on the cell-type taxonomy tree (Fig. 1c). Divisions were assigned based on the splits at the first level of the taxonomy tree (and fine-tuned based on neurotransmitter identity for the 'modulatory' division), resulting in three divisions. Subclasses for divisions 1 and 3 were determined based on splits at the subsequent taxonomy tree levels, resulting in three subclasses in division 1 and five subclasses in division 3. Two types within division 2 were glutamatergic and arose from MB-enriching dissections, so were assigned to the 'MB glut' subclass. As clustering of the types within division 2 was based on the abovementioned LIM homeobox genes, subclasses for division 2 were similarly defined by LIM homeobox gene expression along with the major expressed neurotransmitter to yield five additional subclasses in division 2. Subclasses and types were named using a combination of representative region names, major neurotransmitters and, in some cases, marker genes. Type ID numbers were assigned sequentially based on the taxonomy tree order of types.

**Constellation plot.** A constellation plot was used to visualize the global relatedness between SPN types, where each transcriptomic type is represented by a node (circle) whose surface area reflects the number of nuclei within the type in log scale. The position of each node corresponds to the centroid position of the corresponding type in UMAP coordinates. The relationships between nodes are shown as edges that were calculated as follows. For each nucleus, 15 nearest neighbours were determined in reduced dimension space and summarized by type. For each type, the fraction of nearest neighbours that were assigned to other types was calculated. The edges connected two nodes in which at least one of the nodes had more than 5% of nearest neighbours in the connecting node, and the width of the edge at the node reflects the fraction of nearest neighbours that were assigned to the connecting node and was scaled to node size. For all nodes, the maximum fraction of 'outside' neighbours was determined and set as edge width = 100% of the node width. These plots were created using the plot_constellation function included in scratch.bigcat package.

**Module scores for feature expression programmes.** Seurat's 'AddModuleScore' function was used to calculate the average expression levels of modules of multiple genes (that is, paralogous LIM homeobox genes in Extended Data Fig. 15).

**Differential expression and Gene Ontology analysis.** Differential expression analysis for projection target differences and SPP1⁺ versus SPP1⁻ neurons was performed with MAST[58] using Seurat's FindAllMarkers or FindMarkers function (adjusted $P < 0.05$). enrichR[59] and the GO_Biological_Process_2021, GO_Cellular_Component_2021 and GO_Molecular_Function_2021 libraries were used to identify enriched biological processes or molecular functions.

**SCENIC.** To evaluate the activity of transcription factor regulons in each cluster of SPNs from PONS- and MED-enriching dissections, we employed SCENIC[60,61]. First, we extracted the raw count matrix of the merged Seurat object and used it as input to identify co-expression modules by 'grn'. Next, we performed *cis*-regulatory motif enrichment analysis using 'ctx' with a nes_threshold of 2.5 and a min_genes of 10. We then used 'aucell' to calculate the regulon activity in each individual cell. To visualize the regulon activity in each cluster, we averaged and scaled the regulon activity by cluster. Specifically, we visualized the regulon activity of *Lhx1*, *Lhx2*, *Lhx3*, *Lhx4*, *Lhx5*, *Lhx9* and *Lmx1b* using 'pheatmap' in R. We downloaded the list of transcription factors from mouse from https://github.com/aertslab/pySCENIC/tree/master/resources and the transcription factor annotation motifs v.9 collection from mouse from https://resources.aertslab.org/cistarget/.

**Assessing heterogeneity of nuclei.** To produce a measure of heterogeneity within the snRNA-seq dataset (Extended Data Fig. 17b), we computed for each nucleus the average distance to the $K$ nearest neighbours ($K = 15$) in the reduced principle dimensional space (dimension = 100) and normalized it as $Z$-score by subtracting the means and dividing by the standard deviation. Nuclei in the highly homogenous clusters have much shorter distance to their $K$ nearest neighbours compared to nuclei in highly heterogenous clusters. This metric can be used to measure the local heterogeneity of each nucleus, regardless of their cell-type identities.

**Mapping to AIBS WB scRNA-seq.** To assess the correspondence between the SPN types identified in this study with those in the companion AIBS WB taxonomy[1], we mapped 10x SPN nuclei data onto the mouse WB taxonomy using Hierarchical Approximate Nearest Neighbour mapping available in scratch-mapping package and calculated the confusion matrix at cluster, supertype and subclass levels of WB taxonomy (cluster-level correspondence is shown in Extended Data Fig. 8). The correspondence was summarized by each SPN cell type with its members' average correlation of available marker genes to the mapped WB taxonomy clusters (Supplementary Tables 3–5), which showed that more fine-grained types were identified in the MB and HB region in this study.

**Spatial localization of SPN types**

**Integration with AIBS WB MERFISH.** To map SPN types to their anatomical locations, we leveraged data available in the companion AIBS WB taxonomy[1]. We performed integrative clustering using WB 10x cells, WB MERFISH cells and the 10x nuclei SPN dataset using i_harmonize function from the scratch.bigcat package. While the integrated clustering show great cell-type correspondence for the majority of cell types, it remains ambiguous for cell types present in the ventral medial region of the MB and HB. Some of these types are likely to be better represented in the SPN taxonomy due to the more targeted sampling strategy than in the WB taxonomy. To further elucidate the cell-type correspondence, we identified a subset of integrative clusters that contained at least five nuclei from SPN taxonomy, mapped the WB MERFISH and WB 10x cells from these clusters to the SPN taxonomy and mapped the 10x nuclei from SPN taxonomy to the subset of WB taxonomy covered by these integrated clusters. Clusters in SPN taxonomy with fewer than three mapped cells in WB taxonomy and vice versa were removed from the correspondence comparison. We compared the identities of the nuclei from SPN taxonomy and their mapped clusters from the WB taxonomy, and generated the confusion matrix at cluster, supertype and subclass level of WB taxonomy. MERFISH cells from the integrated clusters shared by SPN clusters were mapped to SPN taxonomy directly. All the mapping was performed by assigning cells/nuclei to the nearest cluster centroid for the corresponding taxonomy using the top differentially expressed genes between all pairs of clusters, except for

the MERFISH dataset, in which case, all the genes on the MERFISH gene panels were used.

**MERFISH spot plotting.** After decoding of raw MERFISH data we obtained a file containing the $x$, $y$ and $z$ location of all detected mRNA molecules (detected_transcripts.csv) for each section. The data from each section was rotated to have the dorsal surface point upwards and ventral downwards using the rotate_2d function from the rearrr package. The locations of mRNA molecules were plotted as a scatterplot using the ggplot function from the ggplot2 package with the $Z$-planes collapsed into one. Genes of interest were highlighted in red while the remaining molecules were coloured in grey to provide anatomical context of the section. Files were exported as .png with a resolution of 1,200 dpi (Extended Data Fig. 15h).

**Cre-dependent retrograde labelling.** To validate anatomical locations of neurons expressing certain marker genes, retrograde labelling was performed in various recombinase driver lines (Supplementary Table 14). AAV2/retro-CAG-FLEX-H2B-GFP (produced by Boston Children's Hospital Viral core, $5 \times 10^{12}$ gc ml$^{-1}$) was injected in the cervical and lumbar spinal cord as described above, animals were perfused 2 weeks postinjection, and tissue was processed as described above to locate GFP-expressing nuclei. Mouse lines were selected based on clustering results. For lines requiring tamoxifen induction, tamoxifen was administered via oral gavage (100 μl per 25 g body weight) consecutively for 7 days, starting from 7 days post spinal injection. Mice were euthanized 7 days after completion of tamoxifen dosing. Tamoxifen (VWR IC15673883) was prepared by dissolving in corn oil (20 mg ml$^{-1}$) followed by overnight rotation. Tissue sections were cryosectioned (50 μm, coronal) and one fourth of the serial sections were IHC-processed (AVES chicken-anti-GFP primary 1:500 dilution, donkey-anti-chicken 488 secondary 1:1000 dilution) and imaged on a VS120 slide-scanner as described above. Labelled nuclei were segmented and reconstructed using NeuroInfo. Images containing individual sections were aligned and registered to the Allen CFF with NeuroInfo. H2B-positive nuclei were then detected and mapped to the reference atlas. Cre-specific detections were merged into the same file for comparison and visualization purposes. Horizontal, coronal or sagittal projections of the 3D reconstructions were output via Neuroinfo's 3D visualization window.

### Hybridization chain reaction on sorted nuclei
To validate select differentially expressed genes between cervical- and lumbar-projecting SPNs, single-molecule fluorescence ISH via digital hybridization chain reaction (Molecular Instruments) and quantification was performed on sorted nuclei. GFP$^+$ and/or mScarlet$^+$ nuclei were FANS-sorted as described above (BD FACSAria, 100 μm nozzle) into a PCR tube containing 100 μl of sort buffer. The sorted nucleus suspension was transferred onto a poly-D-lysine coated glass coverslip, incubated for 30 min at 4 °C, fixed with 4% PFA for 15 min and rinsed three times with PBS at room temperature. Digital hybridization chain reaction was then performed by adapting manufacturer's protocols for mammalian cells on a slide. Coverslips were incubated in 70% ethanol overnight at −20 °C. After this, ethanol was aspirated, samples allowed to air dry, washed three times with 2× SSC at room temperature, then prehybridized with probe hybridization buffer for 30 min at 37 °C. The probe hybridization buffer was then removed and a freshly prepared probe solution (16 nM probe in probe hybridization buffer) was added to the samples for overnight incubation at 37 °C. After 12–16 h, the probe solution was removed and samples washed with probe wash buffer four times at 37 °C, followed by two washes with 5× SSCT (SSC + Tween-20) at room temperature. Samples were then pre-amplified in amplification buffer for 30 min at room temperature. Hairpins were freshly prepared by heating to 95 °C for 90 seconds, snap-cooling to room temperature for 30 min, then adding to the amplification buffer at a concentration of

0.06 μM. The hairpin solution was added to the samples for 60 min to ensure single-molecule dots were diffraction limited. The hairpin solution was removed and samples washed five times with 5× SSCT at room temperature. Nuclei were counterstained with DAPI at a final concentration of 1 μg ml$^{-1}$ for 5 min and washed three times with PBS. Nuclei were mounted with ProLong Glass and sealed with clear nail polish. ProLong Glass was allowed to cure for 48–60 h at room temperature and then stored at 4 °C until imaging. Forty probe sets were used for each probe, and probe names and associated accession numbers were as follows: Pcdh11x (NM_001271809.1), Fezf2 (NM_080433.3). Nuclei were imaged on a Zeiss LSM 900 confocal using a ×63 oil immersion objective with ×2 zoom, confocal resolution and optimal $Z$-stacks. For quantification, $Z$-stacks were converted to maximal-intensity orthogonal projections and the number of punctae were manually quantified per nucleus. Statistical significance was assessed with the Wilcoxon test (ggpubr).

### Electrophysiology experiments
**Retrograde labelling.** To retrogradely label SPNs for cell-attached slice recording, AAV2/retro-CAG-GFP (produced by Boston Children's Hospital Viral Core) was injected into the lumbar (L2–4) spinal cord of P42 mice as described above. To retrogradely label SPNs for whole-cell slice recording, AAV2/retro-CAG-tdTomato (produced by Boston Children's Hospital Viral Core) was injected at postnatal day 4. Postnatal day 4 pups were anaesthetized with ice and placed under the ultrasound probe (MS550, Ultrasonic Vevo 3100). A glass pipette was guided to the L3–4 enlargement and 0.8 μl of AAV2/retro-CAG-tdTomato was injected bilaterally at L3 and L4 segments. Pups were then placed on a heat pad to recover.

**Brain slice preparation.** Acute coronal slices were prepared at postnatal day 60–70 for cell-attached recording or postnatal day 20–30 for whole-cell recording. After isoflurane anaesthesia and transcardiac perfusion with ice-cold slicing solution, brains were rapidly extracted. Coronal brain slices (200 μm) were cut with a Leica VT 1200 vibratome in ice-cold slicing solution and subsequently incubated in aCSF solution for 40 min at 36 °C. Slices were then kept at room temperature until transferring to the recording chamber. Slicing solution consisted of (in mM): sucrose 75, NaCl 80, KCl 2.5, NaH$_2$PO$_4$ 1.25, NaHCO$_3$ 26, CaCl$_2$ 0.5, MgCl$_2$ 3.5, Na-ascorbate 1.3, Na-pyruvate 3. aCSF consisted of (in mM): NaCl 125, KCl 2.5, NaH$_2$PO$_4$ 1.25, glucose 25, NaHCO$_3$ 26, CaCl$_2$ 2, MgCl$_2$ 1, Na-ascorbate 1.3, Na-pyruvate 3.

**Cell-attached and whole-cell recordings.** After localizing retrogradely labelled RuSNs using GFP or tdTomato fluorescence, loose-seal cell-attached recordings were made with pipette-solution-filled pipettes of 20–60 MΩ resistance in voltage-clamp mode at 0 mV holding potential. Whole-cell recordings were performed in current-clamp mode with 3–5 MΩ resistance. Pipette solution consisted of (in mM): K-Gluconate 130, KCl 5, HEPES 10, EGTA 0.5, MgATP 4, NaGTP 0.5, biocytin (~0.2 %, Tocris 3349) and Alexa 488 or Alexa 594 fluorescent dye (20 μM, ThermoFisher, A10436, A10438) (mOsm, 280-300; pH adjusted to 7.3–7.4 with KOH). At the end of cell-attached recording, membrane was ruptured to fill the recorded cell with biocytin and dye. Signals sampled at 20 kHz were passed through DigiData 1440 A and amplified via a Multiclamp 700B amplifier, and recorded with pClamp software (v.11, Molecular Devices). All cell-attached and whole-cell recordings were done at room temperature (18–22 °C and 30–32 °C, respectively) in aCSF.

For spike shape analysis in cell-attached recording, single spikes with a negative peak (±5 ms around the peak) were detected for a 10 s period at 10 min from the recording start. Each negative peak was normalized to have 0 baseline level and −1 amplitude at the peak. Baseline was determined based on a 3.75 ms period at the trace beginning. Individual normalized spikes were averaged for each cell and used to determine the half-peak width. Spike width was calculated as the time difference

between the intersections of averaged waveform after linear interpolation and the horizontal line at the amplitude −0.5.

In whole-cell recording, spontaneous firing rate at 0 min was measured from the firing rate during the 10 s immediately after whole-cell configuration. At this time, MultiClamp was in 'I = 0' mode. Spontaneous firing rate at 5 min was calculated 5 min later for 10 s in 'I = C' mode. To measure action potential-related parameters, membrane potential was maintained between −65 to −60 mV with current injection at least 10 min after whole-cell configuration. Action potential-evoking current was determined by 10 or 50 pA steps. Current injection at this value was repeated three times. The first action potential waveform was used for action potential-related variable calculation. Action potential threshold (mV) was the first point right below the d$V$/d$t$ threshold (12 V s$^{-1}$ or 0.6 mV 50 μs$^{-1}$). Action potential peak was the maximum membrane potential above the threshold. Action potential amplitude was the difference between the action potential threshold and action potential peak. Action potential half width (μs) was the time difference at half of action potential amplitude membrane potential with linear interpolation between samples. fAHP was the membrane potential difference between action potential threshold and negative peak within 4 ms past action potential peak. Input resistance was estimated from the slope of linear regression fit (Clampfit 10) between membrane potential changes after current steps from 0 to −50 pA with −10 pA steps or from 0 to −250 pA with −50 pA steps. For frequency–current relationship, action potential number was counted during current steps (50 pA, 1 s) increasing from 0 to 1,000 pA.

**Floating IHC on acute slices.** Acute slices containing Alexa 488 or Alexa 594 and biocytin-filled patched cells were fixed overnight in 4% PFA at 4 °C. Slices were washed with PBS and then treated with a blocking solution (5% normal donkey serum, 0.5% Triton-X) for 2 h at room temperature. Floating slices were incubated in primary antibodies (AVES chicken-anti-GFP or Rockland rabbit-anti-RFP, R&D Systems goat-anti-Osteopontin, Millipore mouse anti-NeuN clone A60; all 1:500 dilution in blocking solution) at 4 °C overnight. After 12–16 h, slices were washed three times for 15 min with PBS at room temperature, and subsequently incubated with streptavidin (conjugated with Alexa Fluor 488 or 594; 1:500 dilution in PBS) and secondary antibodies (Alexa Fluor 488 donkey-anti-chicken or Alexa Fluor 594 donkey-anti-rabbit, Alexa Fluor 647 donkey-anti-goat, Alexa Fluor 405 donkey-anti-mouse; all 1:500 dilution in PBS) for 2 h at room temperature. Finally, sections were washed with PBS three times for 15 min at room temperature, mounted with ProLong Glass mounting medium and sealed with clear nail polish. ProLong Glass was allowed to cure for 48–60 h at room temperature and then stored at 4 °C until imaging. Patched cells were imaged on a Leica TCS SP8 Laser Scanning Confocal with a ×20 objective and optimal Z-stack (LAS X Stellaris software). Soma area was manually measured using the 'Freehand Selection' tool in Fiji. SPP1 status was determined by assessing SPP1 and Streptavidin colocalization; in cells determined to be SPP1$^-$, NeuN signal was confirmed to ensure antigens were retained in cytoplasm despite cell membrane puncturing during electrophysiological recording.

**Statistics.** Statistical analysis was performed using the Mann–Whitney test or mixed-effects model as indicated in Supplementary Table 13.

**IHC for SPP1 on rubrospinal neurons.** To retrogradely label RuSNs, AAV2/retro-Syn-H2B-GFP was injected into the cervical (C4–6) and lumbar (L2–4) spinal cord of P42 mice as described above. Tissue sections and immunocytochemistry were prepared as described above with AVES chicken-anti-GFP and R&D Systems goat-anti-Osteopontin primary antibodies (1:500 dilution in blocking solution), and donkey-anti-chicken 488 and donkey-anti-goat 647 secondary antibodies (1:1000 dilution in PBS). Images were acquired as Z-stacks with a ×20 objective on a Zeiss LSM 900 Confocal Laser Scanning Microscope.

$n$ = 3 retrogradely labelled brains were stained to yield representative results depicted in Fig. 6c,d.

**Soma size measurement and SPP1 status assignment for SPNs.** To measure soma size and SPP1 status of SPNs throughout the entire mouse brain, AAV2/retro-CAG-GFP was injected into the cervical (C4–6) and lumbar (L2–4) spinal cord of P42 mice as described above. At P56, mice were perfused and tissue sections and IHC were prepared as described above with AVES chicken-anti-GFP and R&D Systems goat-anti-osteopontin primary antibodies (1:500 dilution in blocking solution), and donkey-anti-chicken 488 and donkey-anti-goat 647 secondary antibodies (1:1,000 dilution in PBS). Images of all ROIs in one coronal series were acquired as Z-stacks with a ×20 objective on a Zeiss LSM 900 Confocal Laser Scanning Microscope. Automated segmentation of GFP$^+$ cells was performed using a neural network model fine-tuned on manually annotated images using CellPose[62]. A customized quantification, measurement and classification pipeline was implemented using Python. Cell-size measurements were obtained from the segmentation masks. Identification of double-positive GFP$^+$/SPP1$^+$ cells was done by pixel overlap intensity thresholding, counting SPP1$^+$ cells as those with at least 65% of the pixels within the mask having high co-occurrent intensity in both channels, which accounts for the cytoplasmic but not nuclear staining pattern characteristic of SPP1. False colocalization positives were manually corrected. Statistical significance was assessed with the Wilcoxon test (ggpubr).

## Reporting summary

Further information on research design is available in the Nature Portfolio Reporting Summary linked to this article.

## Data availability

The data are accessible through the Neuroscience Multi-omics (NeMO) Archive (https://assets.nemoarchive.org/dat-76h044v) and the Gene Expression Omnibus (GEO; accession number GSE247602). The AIBS WB atlas data are accessible through NeMO (https://assets.nemo-archive.org/dat-qg7n1b0). Source data are provided with this paper.

## Code availability

Code to reproduce analyses here is available at https://github.com/ZhigangHeLab/SPN_atlas. Additional code used in the manuscript is available at https://github.com/Allen Institute/scrattch.bigcat and https://github.com/AllenInstitute/scrattch.mapping.

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

**Acknowledgements** We thank C. Chen, K. H. Wang, J. Sanes and C. Woolf for critical reading of the manuscript; C. Saper, C. Cepko, V. VanderHorst, F. Chen, S. Conti and J. Page for thoughtful discussions; R. Gupta for statistics and R curbsides; the Allen Institute Molecular

Biology team and Imaging team for generating the SMART-seq and MERFISH datasets, respectively; M. Blanchard, M. Ocaña and the Neurobiology Imaging Facility at Harvard Medical School, Boston, MA, for imaging samples at the TissueCyte; M. Scimone and the Boston Children's Hospital (BCH) Cellular Imaging Core, funded by NIH P50 HD105351; H. Meng and the BCH Viral Core, funded by NIH grants HD018655 and P30EY012196; R. Mathieu, T. Berisha, and the BCH Flow Core, funded by HSCI-BCH-89141.05; J. Daley and the Dana Farber Cancer Institute Flow Core; C. Woolf for providing the Penk-Cre mouse line; G. Mentis for providing the Chx10-Cre mouse line; A. Kumar for QuPath advice; P. Ge for guidance on sorted nucleus coverslipping. This work was supported by the National Institutes of Health (grant numbers F30AG074598 to C.C.W., U19MH114830 to H.Z., R01NS109947 and R01AT010779 to Z.H.), Allen Institute for Brain Science (to C.T.J.v.V., Z. Yao, C.L., D.M., M.K., N.W., K.A.S., B.T., and H.Z.), Wings for Life Spinal Cord Research Foundation (to C.C.W. and Z.Z.), Department of Public Health of Massachusetts (to Z.H.), and Dr. Miriam and Sheldon G. Adelson Medical Research Foundation (to Z.H.).

**Author contributions** C.C.W., A.J., Y.L., B.T., H.Z., Z.H. conceptualized the work. C.C.W. was the data analysis lead and responsible for coordination. snRNA-seq data generation was led by C.C.W., with help from A.J., Z.Z., Q.W. and K.A.S. snRNA-seq data processing and analysis was led by C.C.W., with help from A.J., J.S., C.T.J.v.V., Z.Y., C.L., S.Y., K.G., Yu Zhang, X.S., R.K., D.H.G. AIBS WB scRNA-seq and MERFISH mapping were led by C.T.J.v.V. and Z. Yao, with help from C.L., D.M., M.K. and N.W. Retrograde injections were performed by Z.Z., with help from J.Z. and J.T. Anterograde injections were performed by G.E.R. and J.T. Histology and analysis was led by C.C.W., with help from J.S., G.D.S., M.C.T. and R.J.D. Electrophysiological experiments and data analysis were led by L.C., with help from C.C.W. and E.K. Generation of mouse lines was done by M.M.F. and L.V.G. (Ucn-Cre), A.K. and G.F. (Spp1-Cre). Production of viral vectors was led by Yiming Zhang, with help from Yu Zhang, M.C.T., and Z. Yang. Writing of the Article was done by C.C.W. and Z.H. The generation of figures was led by C.C.W., with help from A.J., J.S., L.C., C.T.J.v.V., Z. Yao, C.L., S.Y., K.G., E.K., Yu Zhang and X.S. All authors contributed to manuscript review and editing.

**Competing interests** Z.H. is an advisor of SpineX, Life Biosciences, and Myro Therapeutics. H.Z. is on the scientific advisory board of MapLight Therapeutics, Inc. The remaining authors declare no competing interests.

**Additional information**
**Correspondence and requests for materials** should be addressed to Anne Jacobi, Hongkui Zeng or Zhigang He.

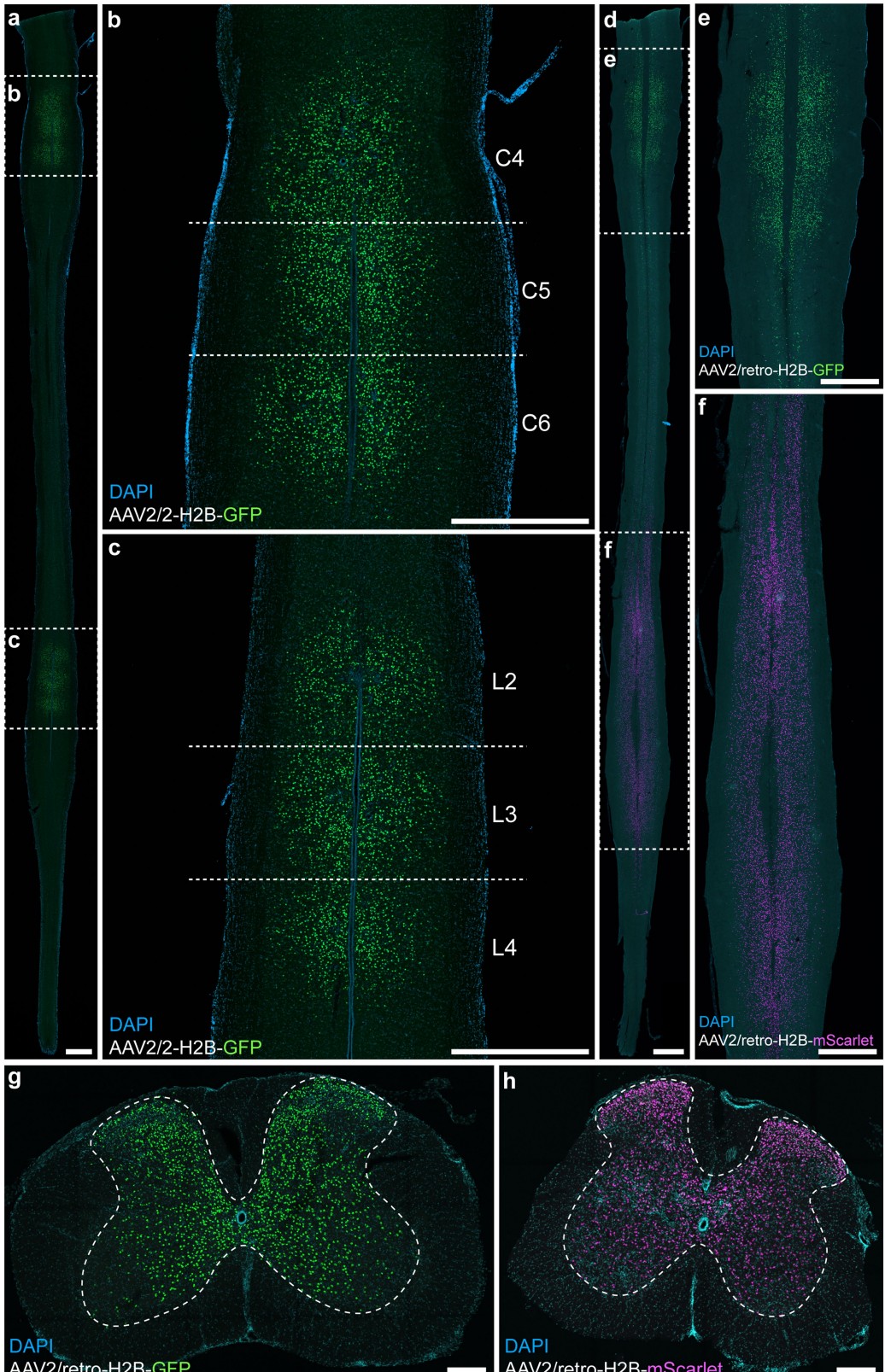

**Extended Data Fig. 1** | See next page for caption.

**Extended Data Fig. 1 | Spinal injection sites of adeno-associated viruses for nuclear labelling.** Recombinant AAVs were injected into multiple segments centring on the cervical and lumbar spinal cord of P42 C57BL/6J mice. Mice were euthanized 2 weeks later at P56. **a-c**, Representative horizontal (i.e., dorsal-ventral longitudinal) sections of spinal cords injected with AAV2/2-H2B into cervical levels 4-6 (C4-6) and lumbar levels 2-4 (L2-4). AAV2/2-H2B does not retrogradely label neurons and therefore labels cells only local to spinal injection sites. **b** and **c** are zoom-in of dashed boxed in **a. d-f**, Representative horizontal sections of spinal cords injected with AAV2/retro-H2B into C4-6 and L2-4. AAV2/retro-H2B, the viral vector used to retrogradely label spinal projecting neurons, has retrograde labeling capability and therefore the spinal injection sites depict both locally transduced cells and retrogradely labelled intraspinal projection neurons (i.e., propriospinal neurons). **e** and **f** are zoom-in of dashed boxed in **d. g-h**, Representative coronal sections of (**g**) cervical and (**h**) lumbar spinal injection sites labelled by AAV2/retro-H2B demonstrating labeling in ventral horn, dorsal horn, and intermediate zone of the spinal cord grey matter (marked by dashed outline). N = 3 spinal cords were labelled with AAV2/2-H2B and N = 3 spinal cords were labelled with AAV2/retro-H2B to yield representative images shown here. Scale **a-f**, 1000 μm; scale **g-h**, 200 μm.

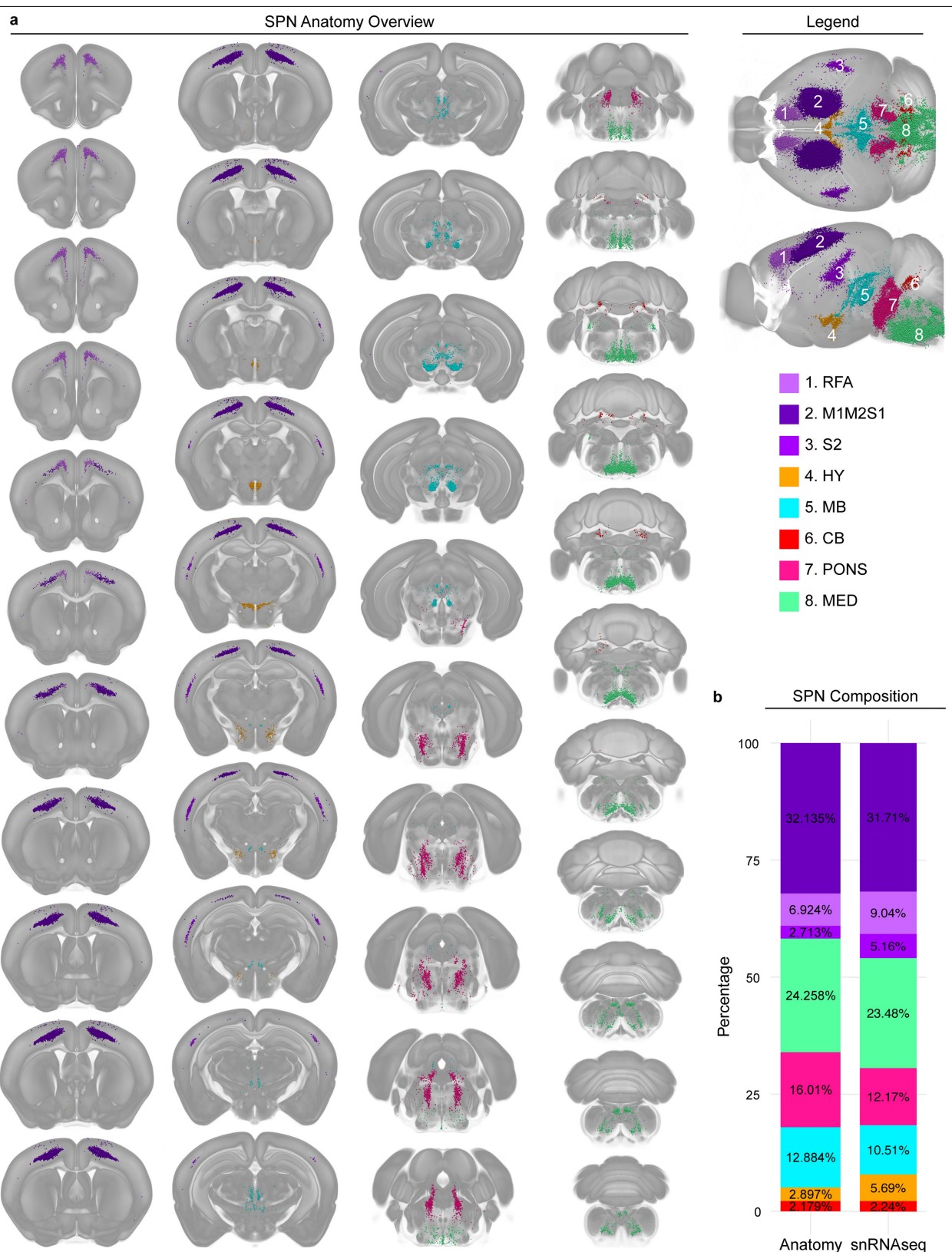

**Extended Data Fig. 2 | Anatomical overview of spinal projecting neurons.** **a**, Coronal reconstructions (250 μm thickness) of segmented and registered SPN nuclei imaged with STPT. **b**, Bar plots represent percentages of SPNs profiled from each SPN-containing brain region in snRNA-seq (10x and SSv4; number of nuclei in each ROI-enriching dissection are as follows, M1M2S1 : 20612, RFA: 5876, S2: 3354, MED: 15264, PONS: 7913, MB: 6834, HY: 3696, CB: 1453)

and histological datasets (slide-scanner imaging; N = 5 mice, mean and standard deviation for number of nuclei in each ROI are as follows, RFA: 5586.40 +/− 425.76, S2: 2188.80 +/− 101.84, M1M2S1 : 25925.60 +/− 857.80, HY: 2337.60 +/− 267.77, MB: 10394.40 +/− 654.72, CB: 1757.60 +/− 265.87, PONS: 12916.80 +/− 1124.68, MED: 19570.40 +/− 465.30). Legend for (**a**) and (**b**) is depicted in upper right corner (same as panel Fig. 1b).

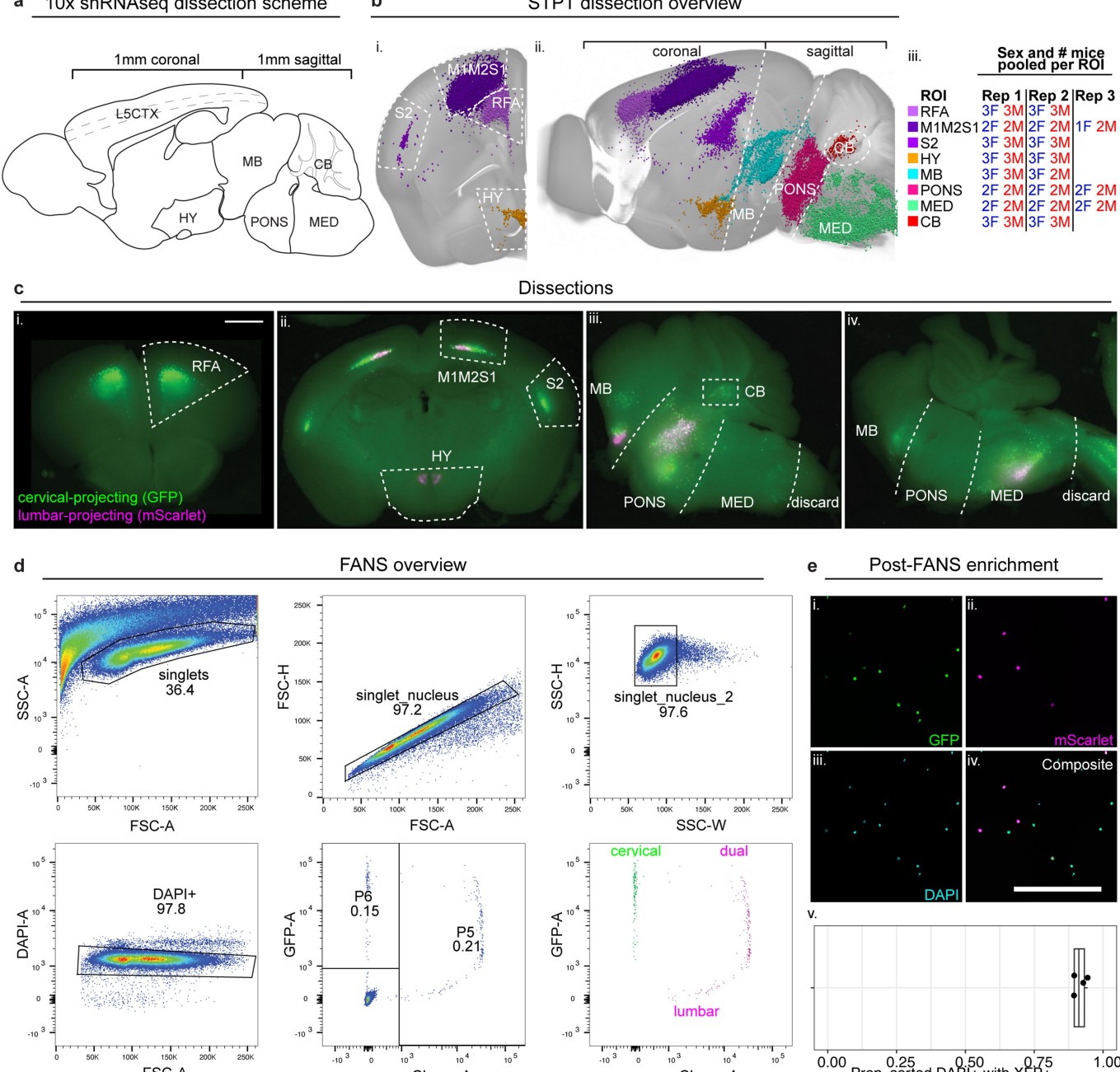

**Extended Data Fig. 3 | High-throughput isolation of retrogradely labelled spinal projecting neurons for snRNA-seq. a**, Regions containing retrogradely labelled SPNs were dissected from P56 mice. A 1 mm coronal-oriented brain matrix was used to section brains from the olfactory bulb to Lambda, and a sagittal-oriented matrix was used to section the remaining subcortex. Test tubes were drawn using templates from Servier Medical Art (Creative Commons Attribution 3.0 Unported License https://creativecommons.org/licences/by/3.0/). Sagittal atlas outline adapted from Allen Institute for Brain Science (https://mouse.brain-map.org/static/atlas)[53]. **b**, Dissection scheme was based on STPT imaging, which shows the cleanest dissection planes to separate the major regions containing SPNs are coronal for forebrain (**i**) and sagittal for midbrain and hindbrain (same as panel Fig. 1b) (**ii**). Equal numbers of male and female mice were pooled per ROI to input sufficient nuclei for FANS (**iii**).

Panel **b** is same as in Fig. 1b. **c**, Example dissecting microscope images showing the dissection planes based on the scheme in **a** and **b**. Estimated scale based on Paxinos adult mouse brain atlas, -1000 μm. This scheme was used to dissect the N = 30 mice from which the snRNA-seq datasets were generated (donor information summarised in Supplementary Table 1). **d**, Following microdissection, nuclei were isolated from each ROI and SPNs enriched using FANS. Representative plots shown are from an M1M2S1 sample. **e**, Enrichment for GFP+ and/or mScarlet+ nuclei was confirmed post-FANS. Scale, 100 μm. Box plot depicts the proportion of sorted nuclei (labelled with DAPI) that had detectable GFP and/or mScarlet via widefield microscopy (N = 4 M1M2S1 samples). The centre line of the box plot depicts the median value (50th percentile) while the box contains the 25th to 75th percentiles; whiskers correspond to the 5th and 95th percentiles. Rep, replicate.

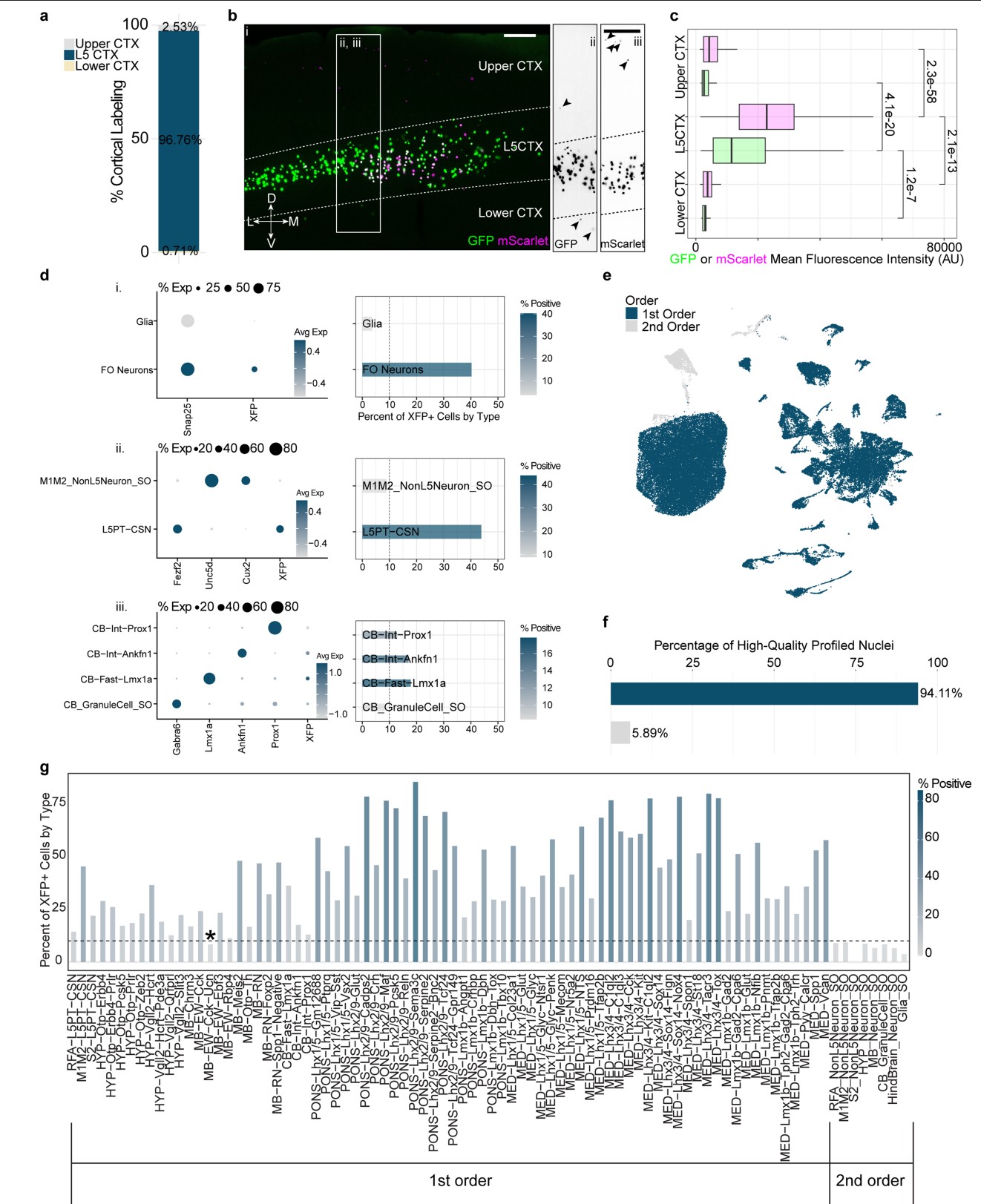

**Extended Data Fig. 4** | See next page for caption.

**Extended Data Fig. 4 | Identification of first- and second-order retrogradely labelled nuclei. a**, The proportion of labelled nuclei were quantified in the cortex, revealing 96.76% and 3.24% of nuclei were labelled within and outside of L5CTX, respectively (N = 5 mice, mean and standard deviation are as follows, L5CTX: 33700.80 +/- 929.89, Upper CTX: 882.40 +/-296.89, Lower CTX: 246.40 +/-147.61). **b**, Qualitative assessment shows lower fluorescence intensity of nuclei in non-L5CTX compared to L5CTX (Cervical: N = 17 lower CTX, 5082 M1M2S1, 42 upper CTX; Lumbar: N = 25 lower CTX, 2606 M1M2S1, 123 upper CTX across 1 retrogradely labelled sample). Arrows indicate faintly labelled nuclei. Scale, 200 μm. **c**, Box plot depicting fluorescence intensity of GFP- and mScarlet- labelled nuclei is significantly higher in L5CTX compared to upper (layers 1–4) and lower (layer 6) cortex. Numbers above boxplots indicate p-values (Wilcoxon test, two-sided). The centre line of box plots depicts the median value, box contains the $25^{th}$–$75^{th}$ percentiles, whiskers correspond to the $5^{th}$ and $95^{th}$ percentiles. **d**, GFP and mScarlet mRNA detection in 10x snRNA-seq data in putative first- and second-order clusters across glial/neuronal (**i**), cortical (**ii**), and cerebellar (**iii**) nuclei. Left: dot plots of marker genes and XFP expression. Right: percent of XFP+ nuclei across types. **e**, UMAP of putative first- and second-order clusters. First- and second-order clusters are defined as those with >10% and <10% of nuclei expressing XFP, respectively. **f**, Percentage of all nuclei that pass quality control thresholds that are classified as first- and second-order. **g**, XFP mRNA detection in snRNA-seq data across all 76 types and the putative second-order clusters that do not pass the 10% threshold, separated by dissection region and neurons vs. glia. Pons and medulla pooled as 'Hindbrain'. *indicates one type that is below the 10% threshold that was designated as first-order because of literature support and confirmatory anterograde labeling (Extended Data Fig. 5). XFP, GFP or mScarlet; AU, arbitrary units; FO, first-order; SO, second-order.

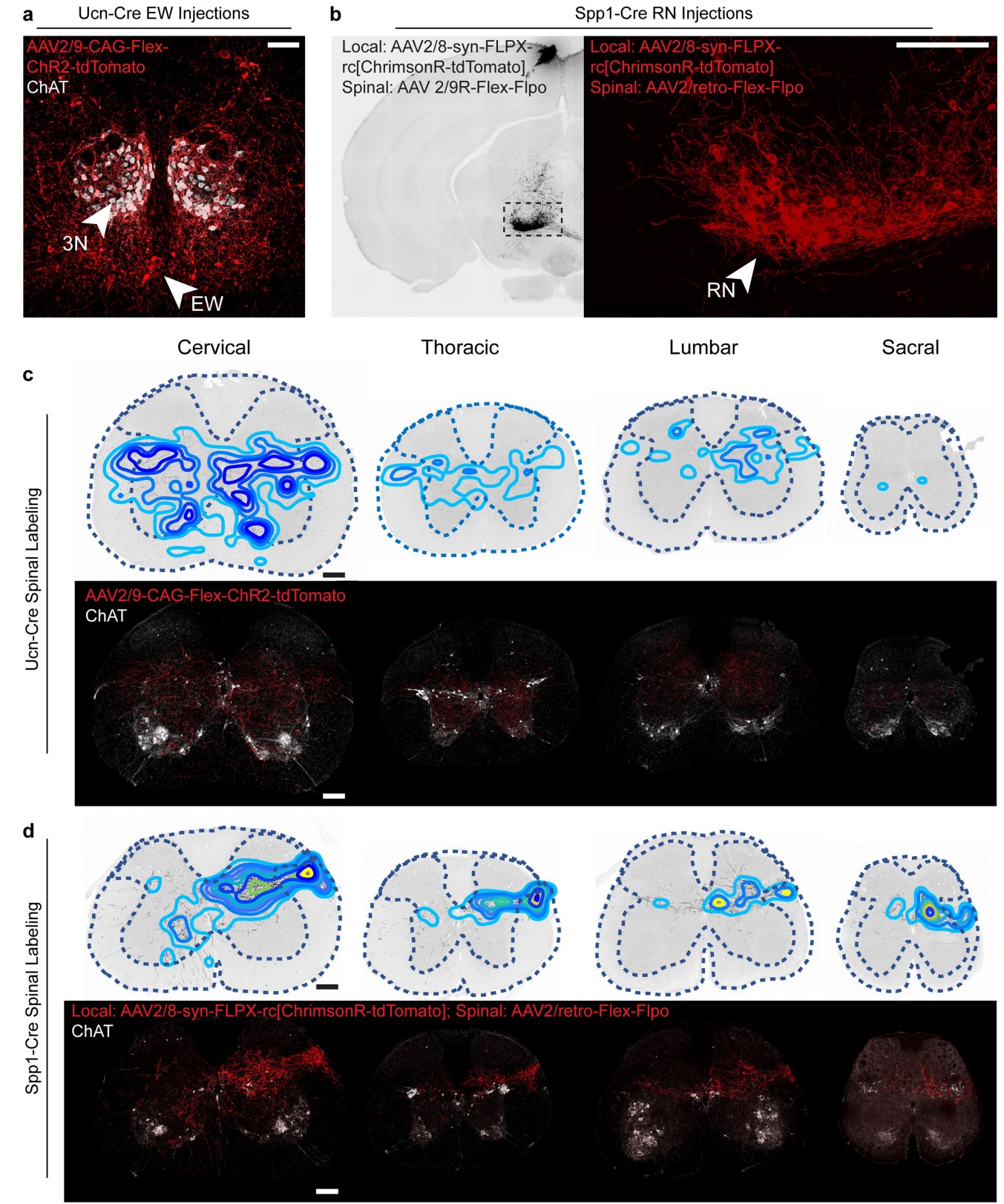

**a** Ucn-Cre EW Injections

AAV2/9-CAG-Flex-ChR2-tdTomato
ChAT

3N

EW

**b** Spp1-Cre RN Injections

Local: AAV2/8-syn-FLPX-rc[ChrimsonR-tdTomato]
Spinal: AAV 2/9R-Flex-Flpo

Local: AAV2/8-syn-FLPX-rc[ChrimsonR-tdTomato]
Spinal: AAV2/retro-Flex-Flpo

RN

Cervical | Thoracic | Lumbar | Sacral

**c**

Ucn-Cre Spinal Labeling

AAV2/9-CAG-Flex-ChR2-tdTomato
ChAT

**d**

Spp1-Cre Spinal Labeling

Local: AAV2/8-syn-FLPX-rc[ChrimsonR-tdTomato]; Spinal: AAV2/retro-Flex-Flpo
ChAT

**Extended Data Fig. 5** | See next page for caption.

**Extended Data Fig. 5 | Local injection and spinal cord projection pattern of Ucn+ EW and Spp1+ RN spinal projecting neurons. a**, Adult Ucn-Cre mice received injection of AAV2/9-CAG-Flex-ChR2-TdTomato into the midline EW nucleus. Panel shows 20x magnification confocal scanning of local infection of EW neurons. Red, infected Ucn+ neurons expressing ChR2-tdTomato; white, ChAT immunopositive 3N neurons. **b**, Adult Spp1-Cre mice received unilateral injection of AAV2/8-syn-FLPX-rc[ChrimsonR-tdTomato] into the RN, and injection of AAV2/retro-Flex-Flpo into C4-6 and L2-4 spinal cord. Left panel, 10x magnification epi-fluorescent scanning showing overview of coronal section at AP -3.8. Dashed area is zoomed in on the right as 20x magnification confocal scanning showing expression of ChrimsonR-tdTomato in Spp1+ SPNs. **c**, Spinal cord projections of Ucn+ EW SPNs. From left to right are representative cervical, thoracic, lumbar, and sacral sections. Top row, axonal signal density contours overlaying axonal signal. Bottom row, 20x confocal scanning of the same section as the top row. Red, ChR2-tdTomato axonal signal; white, ChAT immunostaining signal. **d**, Spinal cord projections of Spp1+ RN SPNs. From left to right are representative cervical, thoracic, lumbar, and sacral sections. Top row, axonal signal density contours overlaying inverted axonal signal. Bottom row, 20x confocal scanning of the same section as the top row. Red, ChrimsonR-tdTomato+ axonal signal; white, ChAT immunostaining signal. Scale, 200 µm. Representative images shown are from N = 3 injected Ucn-Cre mice and N = 2 injected Spp1-Cre mice. RN = red nucleus, AP = antero-posterior position, EW = Edinger-Westphal, 3N = oculomotor nucleus.

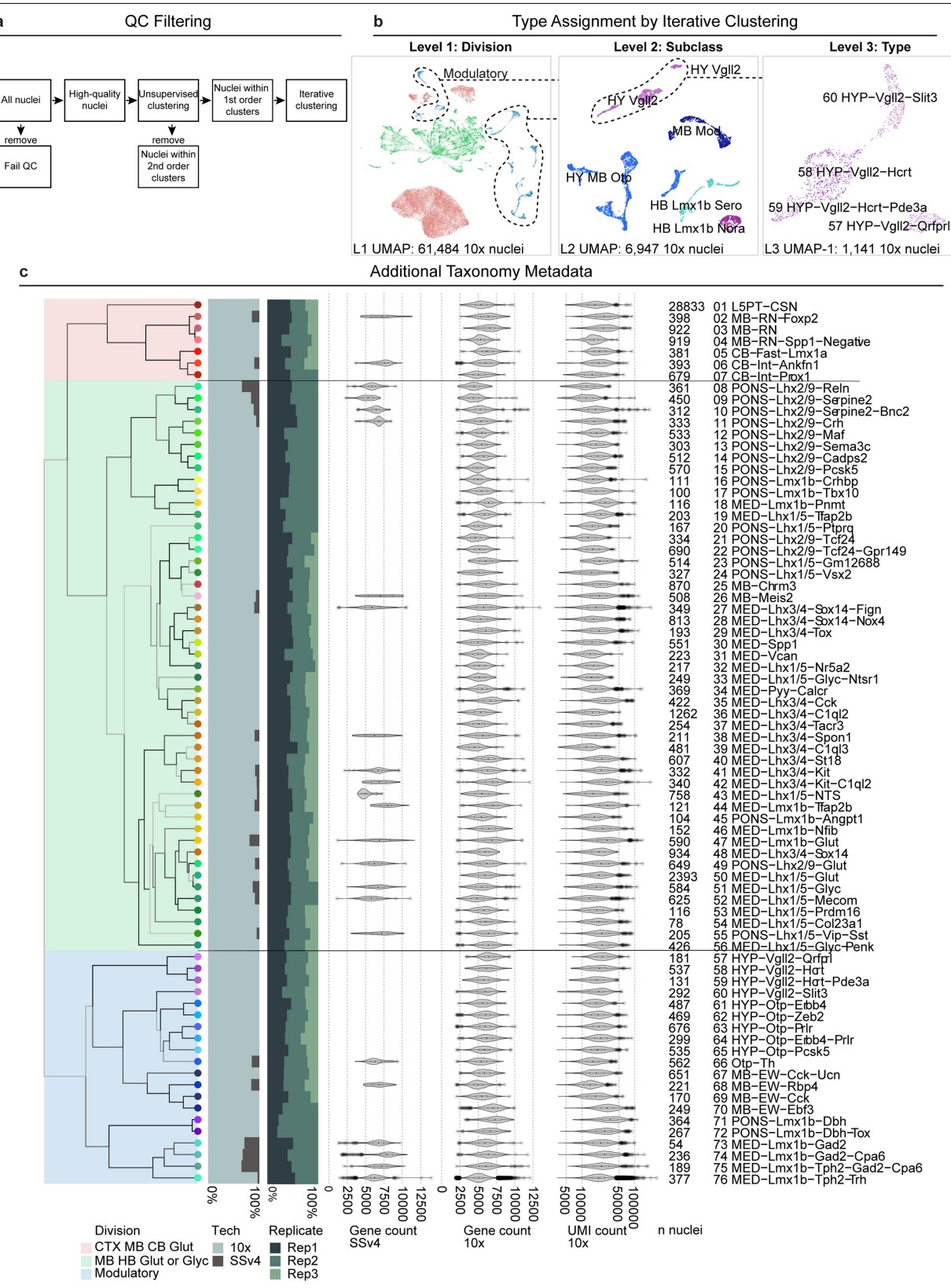

**Extended Data Fig. 6** | See next page for caption.

**Extended Data Fig. 6 | Development of a spinal projecting neuron taxonomy.**
**a**, Workflow diagram of QC filtering of snRNA-seq datasets. Nuclei that passed standard QC metrics were assessed for putative first- and second-order labeling. Putative second-order nuclei were removed, and first-order nuclei underwent multi-level iterative clustering. **b**, UMAPs at three levels of the taxonomy from iterative analysis. Example shows iterative clustering of the Modulatory Division into 5 Subclasses to yield 4 final HY Vgll2 Types. **c**, Dendrogram of SPN taxonomy as in Fig. 1, showing additional metadata. The colour blocks shading the taxonomy tree indicate division. The nodes at the end of the dendrogram indicate 'type', with type number labels and names on the far right. From left to right, the bar plots represent fractions of nuclei profiled with 10x and SSv4 platforms and replicate contribution to each type. Violin plots show gene counts in SSv4 and 10x data, and UMI count in 10x data. The centre line of the box and whisker plot depicts the median value (50th percentile) while the box contains the 25th to 75th percentiles; the whiskers correspond to the 5th and 95th percentiles. The number of nuclei (n nuclei) profiled per type is labelled. QC, quality control; Tech, technology; Rep, replicate; UMI, Unique molecular identifier.

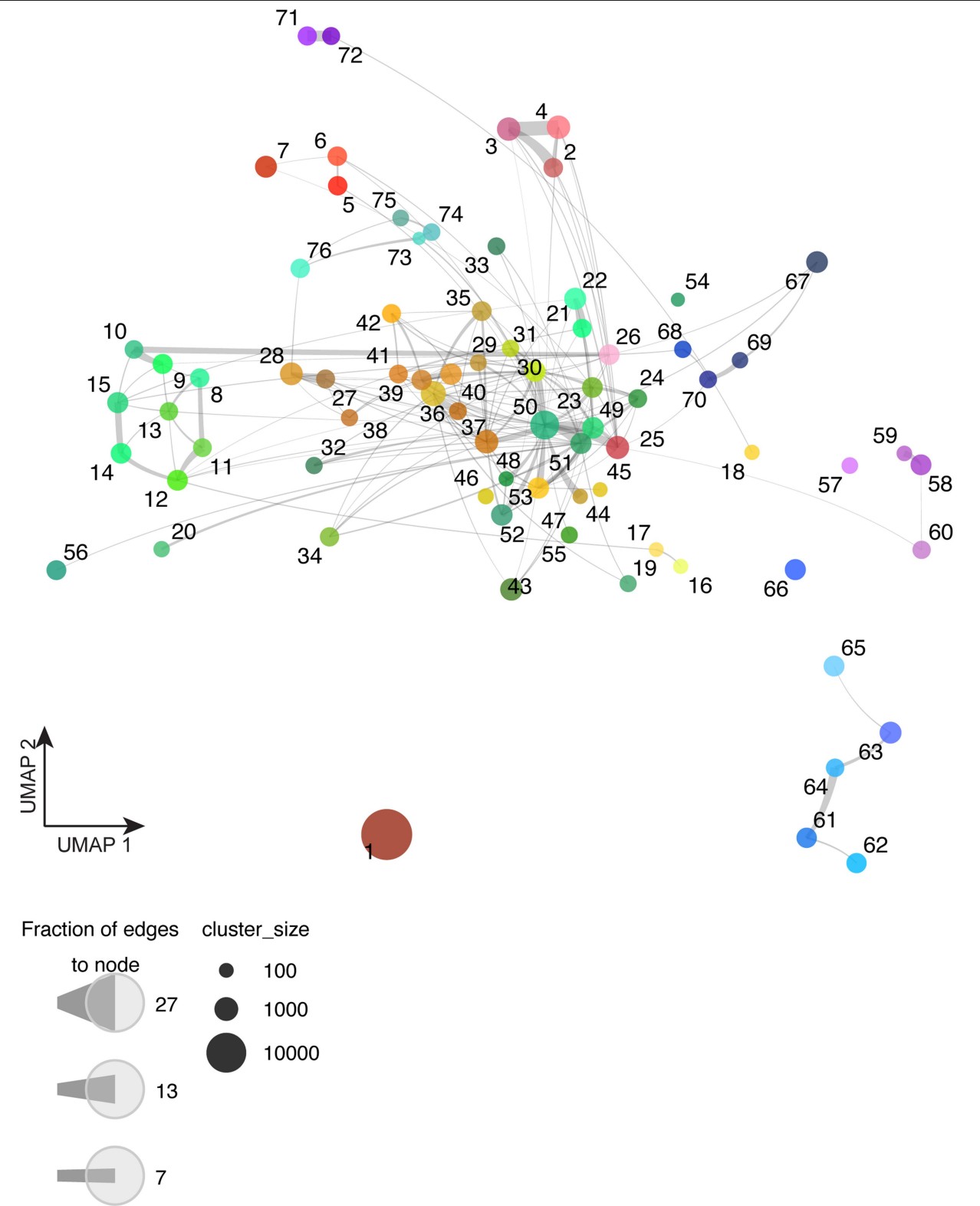

**Extended Data Fig. 7 | Global constellation plot of spinal projecting neurons.** The constellation plot shows the global relatedness across all SPNs. Each transcriptomic type is represented by a node whose area represents the number of nuclei (log-scale). Number labels and colours on nodes correspond to 'type' number from Fig. 1c. Nodes are positioned at the centre of the corresponding type cluster in UMAP space in Fig. 1d,e,f. Relationships between nodes are indicated by edges.

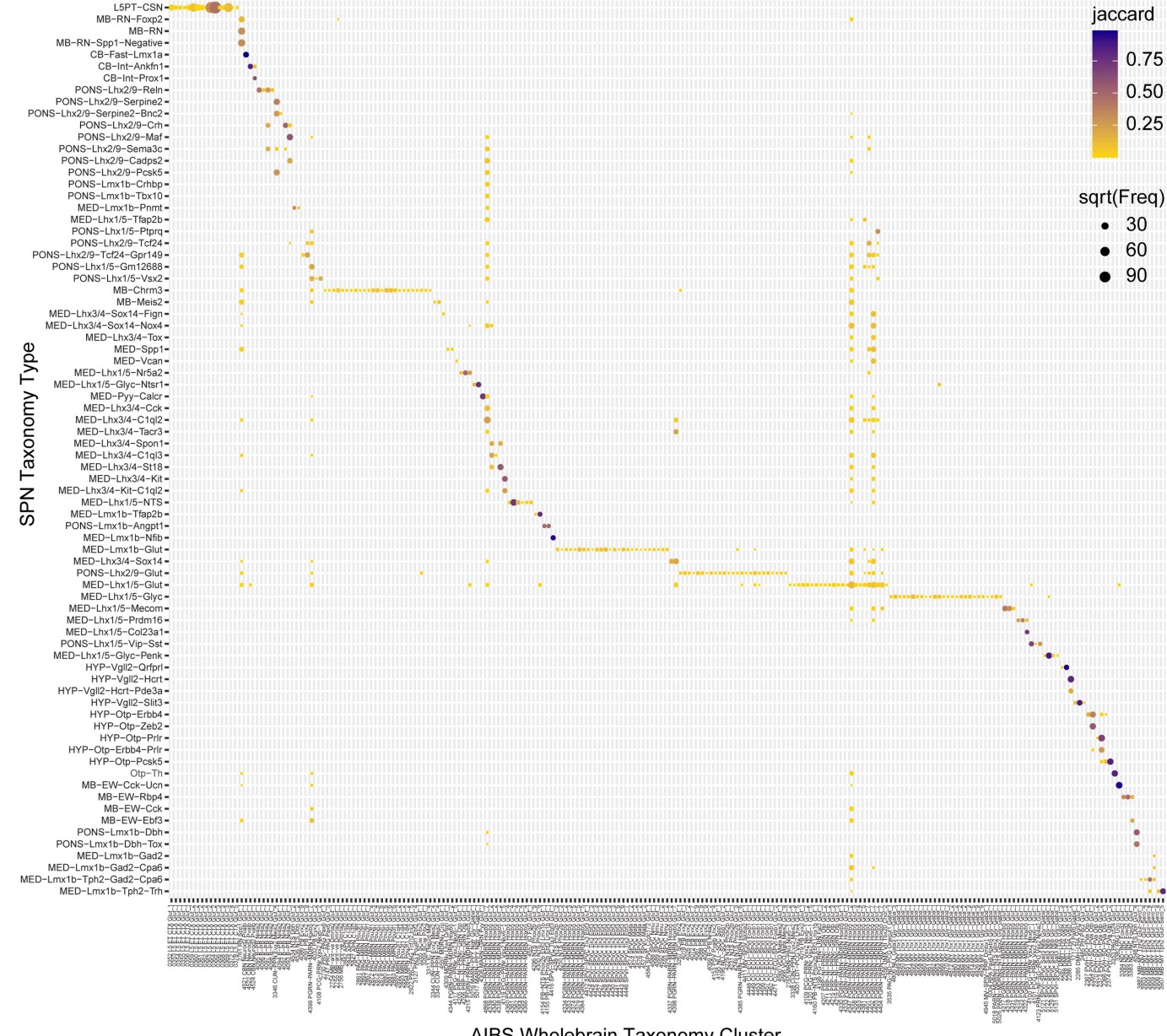

**Extended Data Fig. 8 | Correspondence between the SPN and AIBS WB taxonomies.** A confusion matrix between the SPN snRNA-seq taxonomy and AIBS WB scRNA-seq taxonomy (cluster level) is shown. The size of each dot corresponds to the number of overlapping cells/nuclei, and the color corresponds to the Jaccard similarity score between SPN 'type' and AIBS taxonomy 'cluster'. The matrix is filtered by Jaccard score > 0.1 or frequency (i.e., number of overlapping cells/nuclei) ≥ 5.

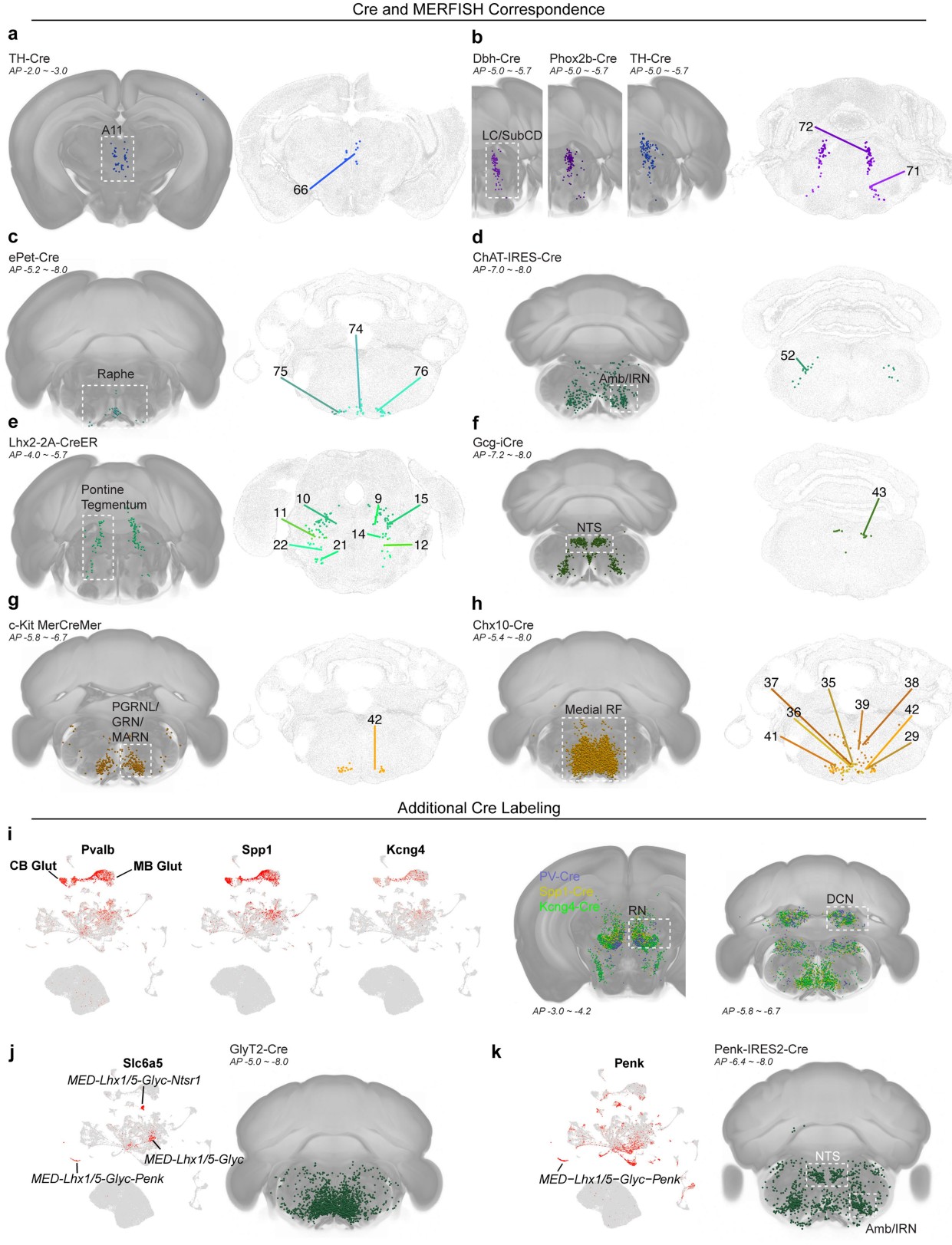

**Extended Data Fig. 9** | See next page for caption.

**Extended Data Fig. 9 | Correspondence between Cre-dependent retrograde labeling and MERFISH mapping. a-h**, Retrograde labeling was performed with AAVs expressing Cre-dependent GFP in various transgenic mouse lines and compared to MERFISH results. Left panels: representative coronal reconstructions of Cre-dependent retrogradely labelled nuclei various transgenic lines (TH-Cre, Dbh-Cre, Phox2b-Cre, ePet-Cre, ChAT-IRES-Cre, Lhx2-2A-CreER, Gcg-iCre, c-Kit MerCreMer, and Chx10-Cre). Approximate AP position and anatomical regions are annotated. Right panels: representative MERFISH sections showing the spatial location of the corresponding SPN type(s). **i-k**, UMAP plots show expression of marker genes in the SPN transcriptomic dataset, depicted alongside representative coronal reconstructions of Cre-dependent retrogradely labelled in additional transgenic lines *without* corresponding MERFISH mappings (targeted transgene yields too broad of a distribution to correspond to specific types in MERFISH results). **i**, PV-Cre, Spp1-Cre, and Kcng4-Cre label rubrospinal neurons and cerebellospinal neurons in Division 1, as well as reticulospinal neurons in Division 2. **j,k** GlyT2-Cre (**j**), and Penk-IRES2-Cre (**k**) lines label reticulospinal neuron populations throughout Division 2. AP, anterior-posterior position (relative to Bregma); RN, red nucleus; DCN, deep cerebellar nuclei; PGRNL, paragigantocellular reticular nucleus lateral part; GRN, gigantocellular reticular nucleus; MARN, magnocellular reticular nucleus; NTS, nucleus of the solitary tract; Amb, nucleus ambiguus; IRN, intermediate reticular nucleus; A11, cell group A11; LC, locus coeruleus; SubCD, subcoeruleus nucleus dorsal part (Paxinos nomenclature[54]).

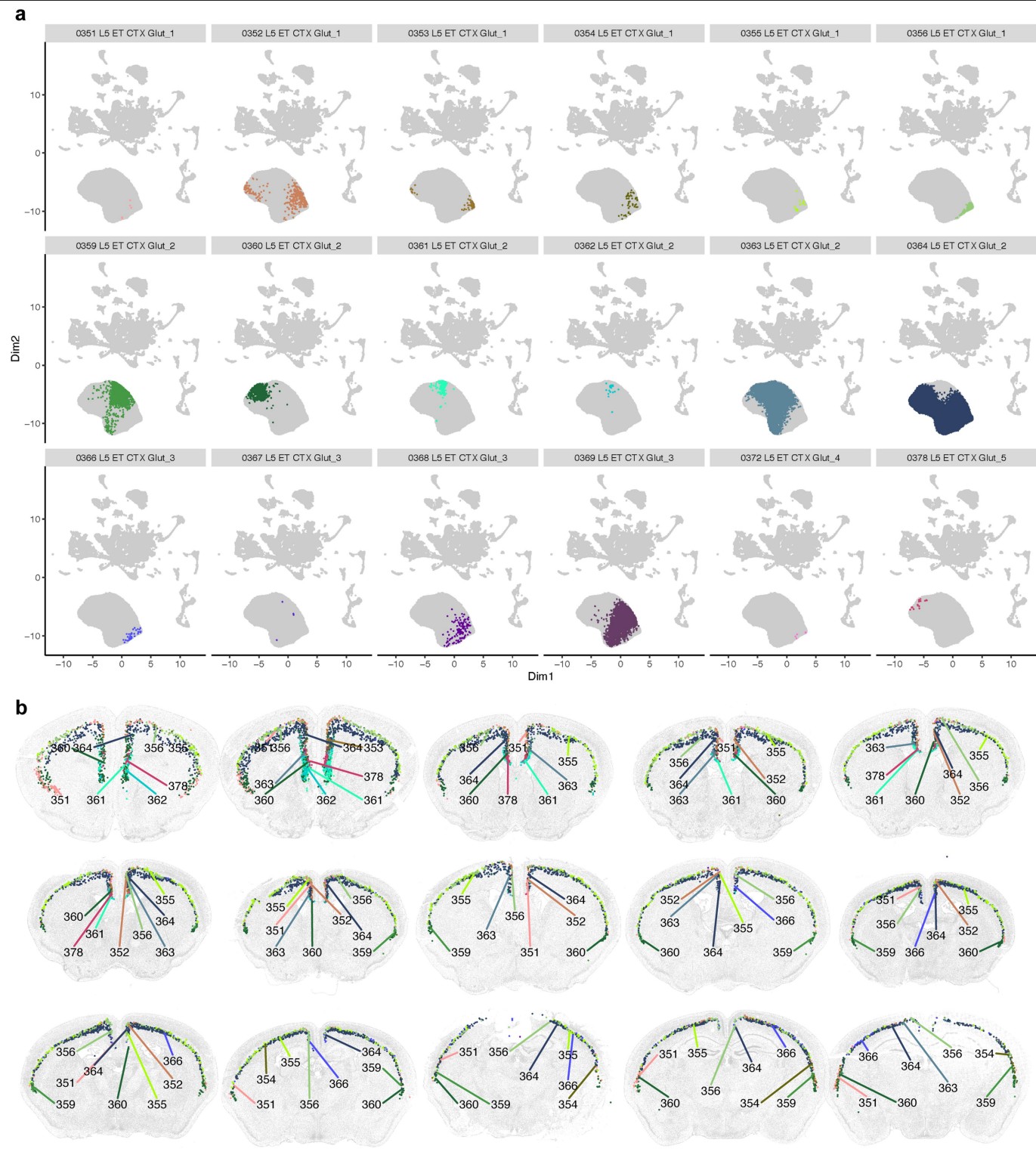

**Extended Data Fig. 10 | Anatomic and transcriptomic heterogeneity of corticospinal neurons.** Corticospinal neurons map to 18 clusters within the AIBS WB scRNA-seq taxonomy. **a**, UMAP representation of CSNs colored by clusters the nuclei map to in the AIBS WB taxonomy. **b**, Representative MERFISH sections showing the spatial location of the 18 AIBS WB clusters. Cluster number labels and colors depicted on UMAP plots correspond to clusters labelled on the MERFISH panels.

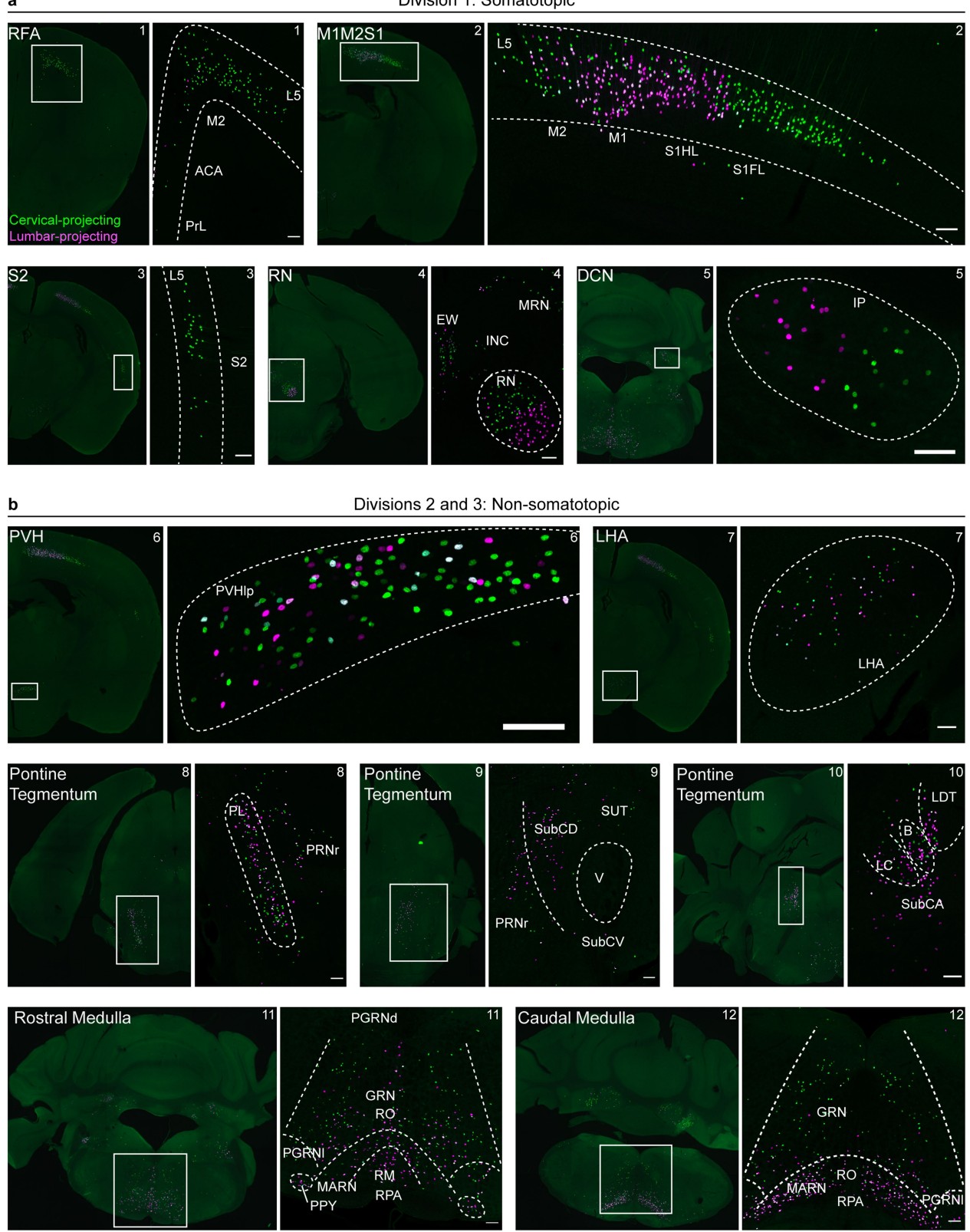

**Extended Data Fig. 11 |** See next page for caption.

**Extended Data Fig. 11 | Somatotopic segregation of Division 1, but not Division 2 or 3, spinal projecting neurons.** Confocal microscopy images of retrogradely labelled cervical- (green) and lumbar- (magenta) projecting SPNs throughout the brain. Populations in **a**, Division 1 exhibit somatotopic segregation of cervical- and lumbar- projecting SPNs, whereas those in **b**, Divisions 2 and 3 do not. Scale, 100 μm. Panels a-2, a-4, a-5 are the same as shown in Fig. 2f. Representative images shown are from N = 3 injected mice. RFA, rostral forelimb area; L5, layer 5; M2, secondary motor cortex; ACA, anterior cingulate area; PrL, prelimbic cortex (Paxinos nomenclature[54], not to be confused with PL); PL, paralemniscal nucleus (Paxinos); M1, primary motor cortex; S1HL, primary somatosensory cortex hindlimb region; S1FL, primary somatosensory cortex forelimb region; S2, secondary somatosensory cortex; RN, red nucleus; EW, Edinger-Westphal Nucleus; INC, interstitial nucleus of Cajal; MRN, midbrain reticular nucleus; DCN, deep cerebellar nuclei; IP, interposed nucleus; PVH, paraventricular hypothalamus; PVHlp, paraventricular hypothalamic nucleus lateral parvicellular part; LHA, lateral hypothalamic area; PRNr, pontine reticular nucleus rostral part; SubCD, subcoeruelus nucleus dorsal part (Paxinos); SubCV, subcoeruelus nucleus ventral part (Paxinos); SubCA, subcoeruelus nucleus alpha part (Paxinos); SUT, supratrigeminal nucleus; V, motor trigeminal nucleus; LC, locus coeruleus; B, Barrington's nucleus; LDT, laterodorsal tegmental nucleus; PGRNl, paragigantocellular reticular nucleus lateral part; PGRNd = paragigantocellular reticular nucleus dorsal part; PPY, parapyramidal nucleus; GRN, gigantocellular reticular nucleus; MARN, magnocellular reticular nucleus; RO, nucleus raphe obscurus; RM, nucleus raphe magnus; RPA, nucleus raphe pallidus.

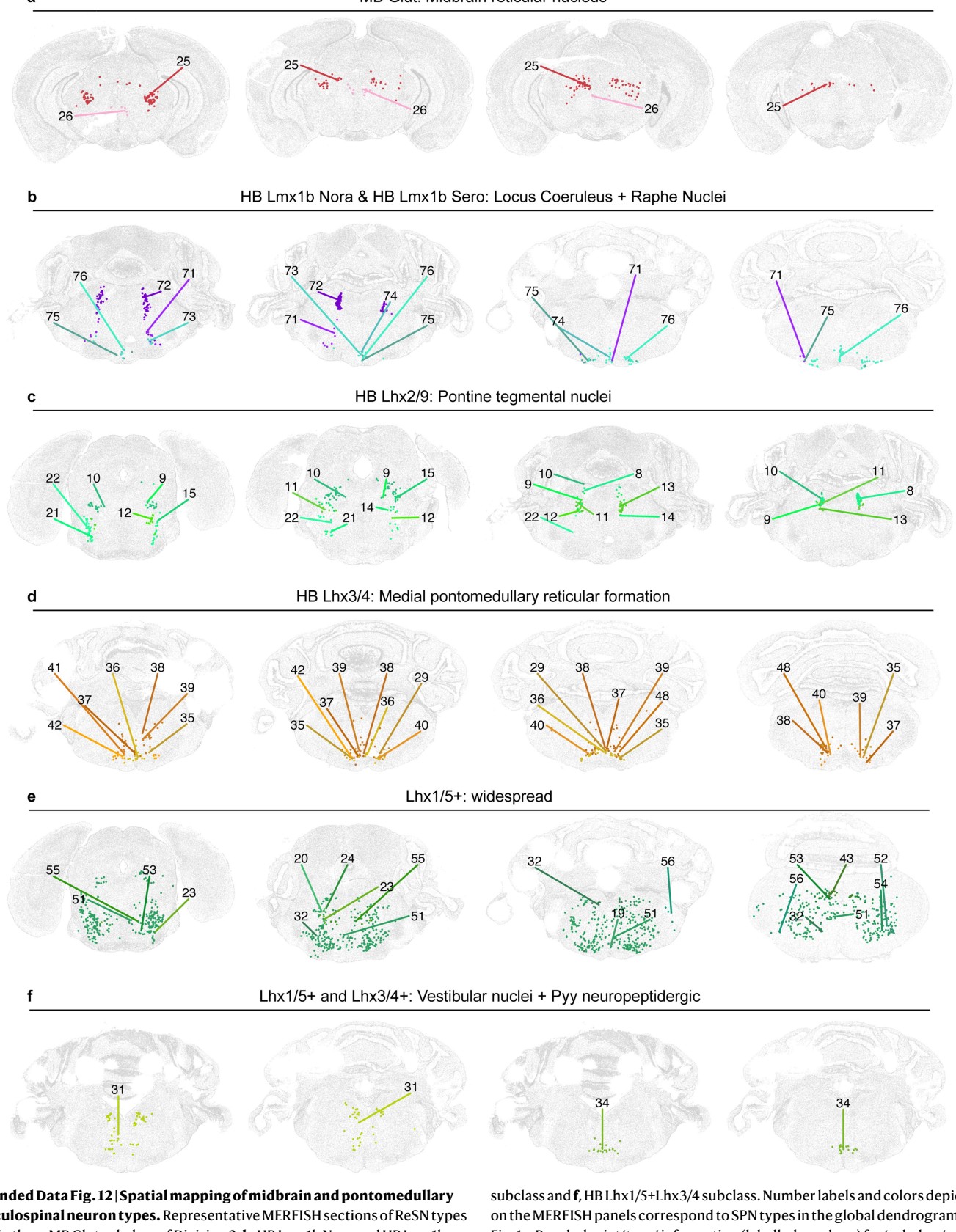

**Extended Data Fig. 12 | Spatial mapping of midbrain and pontomedullary reticulospinal neuron types.** Representative MERFISH sections of ReSN types within the **a**, MB Glut subclass of Division 2, **b**, HB Lmx1b Nora and HB Lmx1b Sero subclasses, **c**, HB Lhx2/9 subclass. **d**, HB Lhx3/4 subclass. **e**, HB Lhx1/5 subclass and **f**, HB Lhx1/5+Lhx3/4 subclass. Number labels and colors depicted on the MERFISH panels correspond to SPN types in the global dendrogram in Fig. 1c. Panels depict 'type' information (labelled numbers) for 'subclass' panels shown in Fig. 4c.

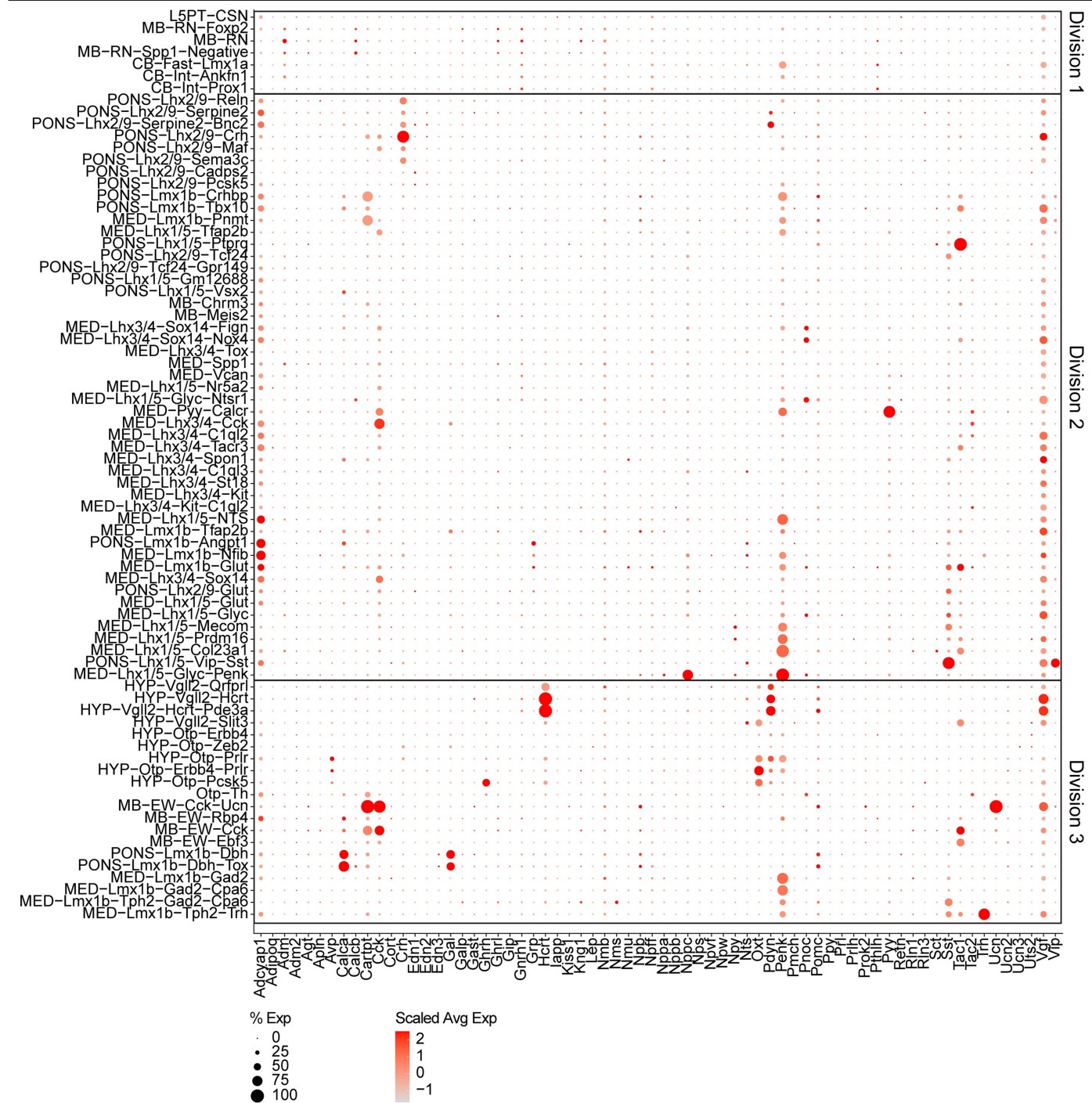

**Extended Data Fig. 13 | Neuropeptide expression across spinal projecting neuron types.** Dot plot showing expression of neuropeptides across the 76 SPN types. % Exp, percentage expressed; Scaled Avg Exp, scaled average expression.

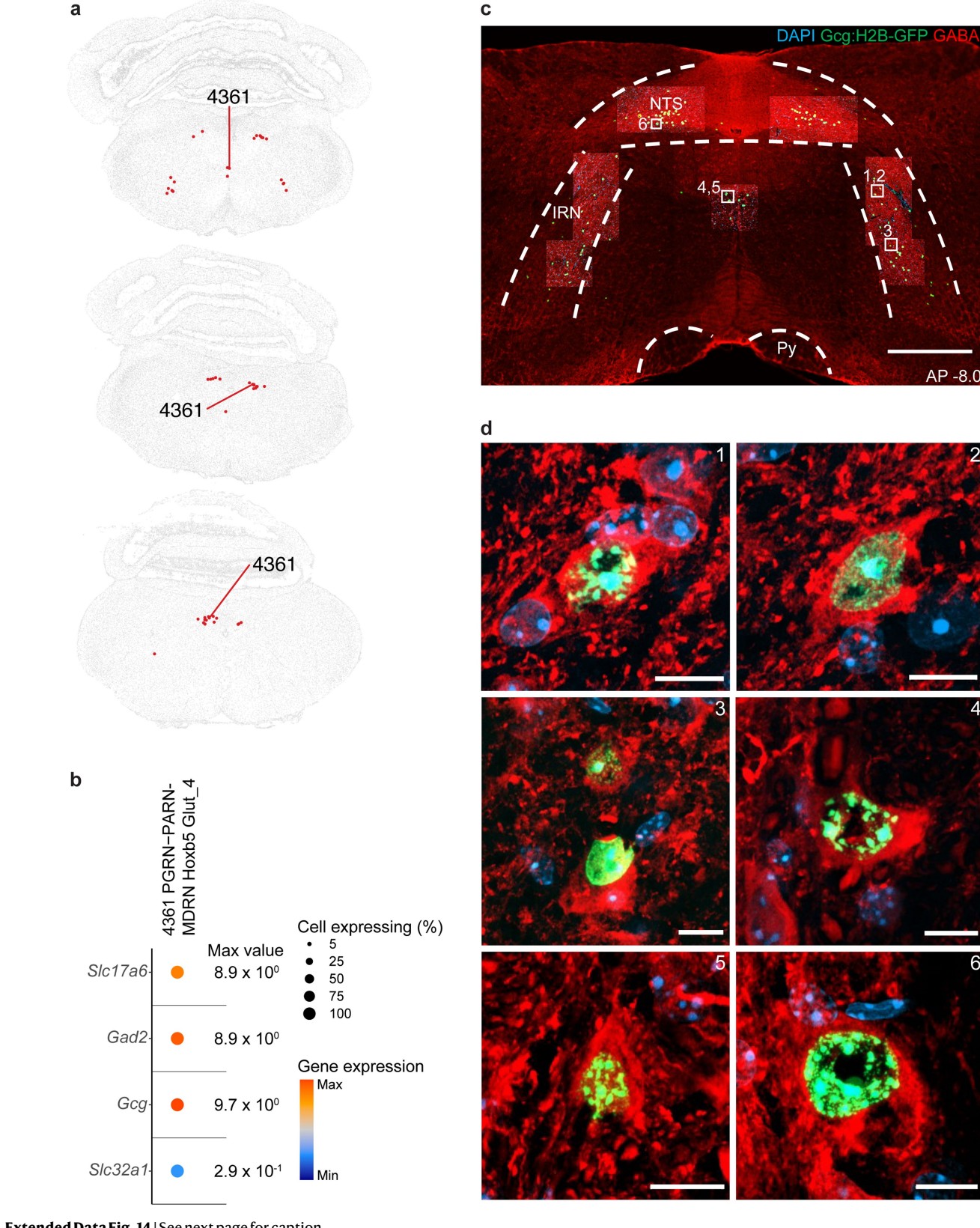

**a**

4361

4361

4361

**b**

4361 PGRN–PARN-
MDRN Hoxb5 Glut_4

Max value

*Slc17a6*    8.9 x $10^0$

*Gad2*    8.9 x $10^0$

*Gcg*    9.7 x $10^0$

*Slc32a1*    2.9 x $10^{-1}$

Cell expressing (%)
· 5
· 25
● 50
● 75
● 100

Gene expression
Max
Min

**c**

DAPI Gcg:H2B-GFP GABA

NTS

IRN

4,5

1,2

Py

AP -8.0

**d**

**Extended Data Fig. 14** | See next page for caption.

**Extended Data Fig. 14 | Neurons in the nucleus of the solitary tract (NTS) and intermediate reticular nucleus (IRN) co-express *Vglut2* and *Gad2*.**
**a**, MERFISH representations of cluster 4361, the primary cluster within the AIBS WB atlas which SPN type MED-Lhx1/5-NTS maps to. Cluster 4361 is located in the NTS and IRN and co-expresses *Slc17a6* (*Vglut2*) and *Gad2*, with low expression of *Slc32a1* (*Vgat*). **b**, Expression levels of *Slc17a6*, *Gad2*, *Gcg*, and *Slc32a1* in MERFISH dataset. **c-d**, GABA immunostaining of *Gcg*+ spinal projecting neurons in the NTS and IRN. *Gcg*+ spinal projecting neurons were labelled via retrograde labeling with AAVs expressing Cre-dependent H2B-GFP in a Gcg-Cre mouse line. **c**, Overlaid 10x and 20x confocal stack showing retrograde labelled Gcg:H2B-GFP signal in the NTS and IRN. **d**, 63x confocal stack showing overlapping signals from GABA staining and H2B-GFP. Representative images shown from N = 2 technical IHC replicates on tissue sections from the same retrogradely labelled brain sample. Number corresponds to the white inserts in (**c**). Red, GABA immunostaining; blue, DAPI; green, GFP; Scale: (**c**) 500 μm, (**d**) 10 μm. AP, anterior–posterior (relative to Bregma); Py, pyramidal tract.

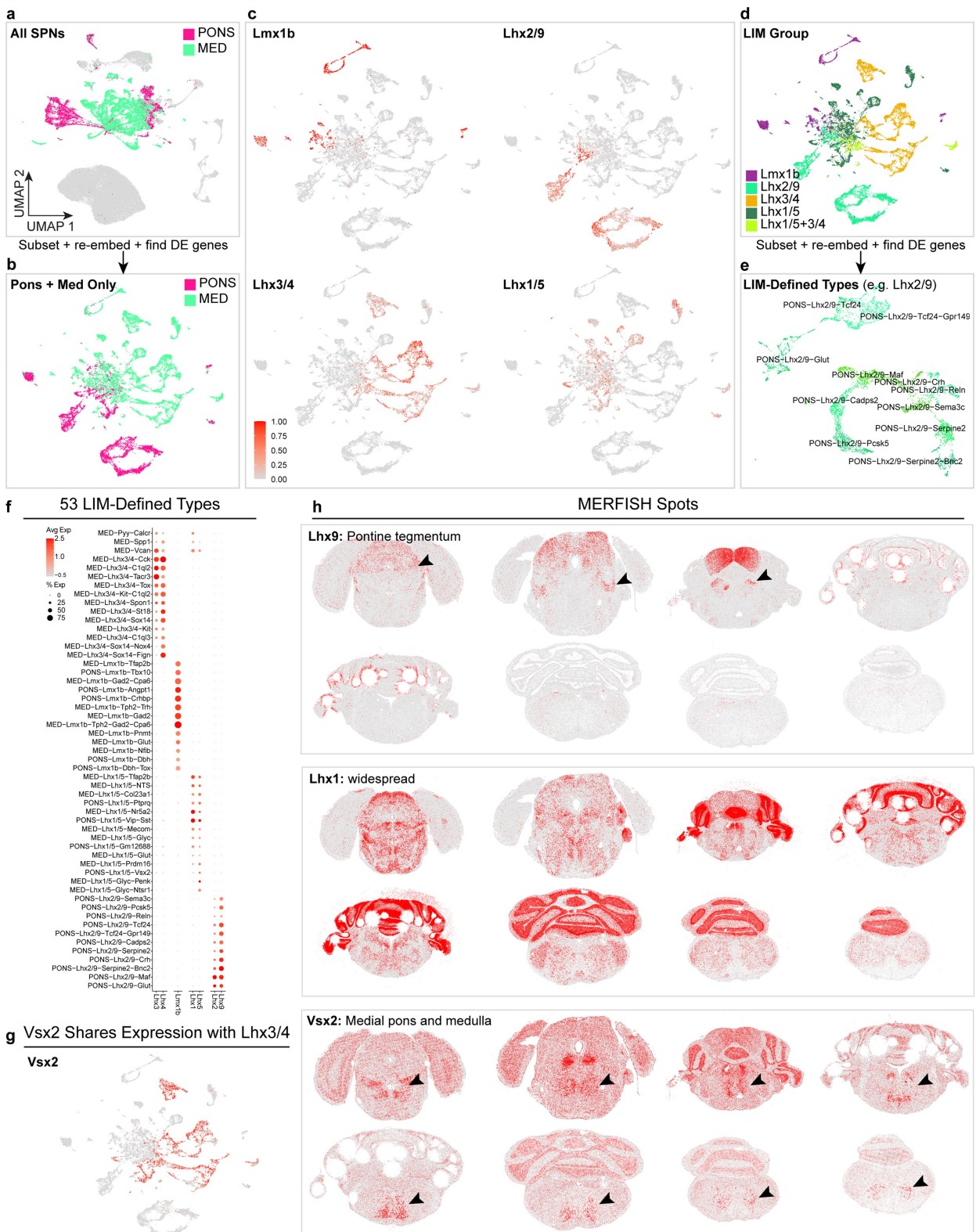

**Extended Data Fig. 15 | Multi-level clustering of pontomedullary reticulospinal neurons. a**, UMAP of all SPNs with nuclei from PONS- and MED-enriching dissections, colored as in Fig. 1. **b**, Nuclei from PONS- and MED-enriching dissections were subset and re-embedded (10x data, N = 22,100 nuclei). **c**, Expression of Seurat ModuleScores of *Lmx1b*, *Lhx2* and *Lhx9*, *Lhx3* and *Lhx4*, *Lhx1* and *Lhx5* show these genes are expressed in mutually exclusive clusters that, together, account for all ReSNs. **d**, UMAP of nuclei from PONS- and MED-enriching dissections, colored by LIM groups. **e**, Each LIM group

was subset and re-embedded to identify the final types in the taxonomy shown in Fig. 1c. Shown is the re-embedding of the Lhx2/9 LIM group. Panels **b**, **d**, and **e** are same as in Fig. 4a. **f**, Dot plot showing expression of *Lmx1b*, *Lhx2*, *Lhx9*, *Lhx3*, *Lhx4*, *Lhx1*, and *Lhx5* across the final 53 LIM-defined types. **g**, Expression of *Vsx2* in nuclei from PONS- and MED-enriching dissections. **h**, MERFISH spot plots of expression of *Lhx9*, *Lhx1*, and *Vsx2*. Arrows to emphasize concentrated regions with expression.

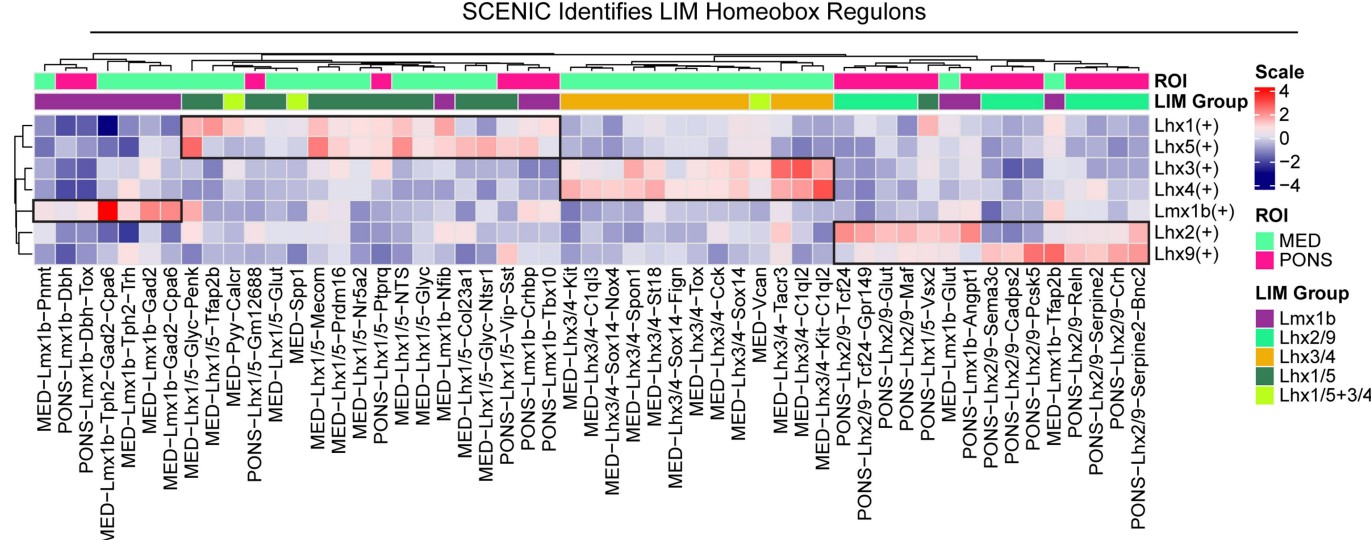

**Extended Data Fig. 16 | LIM Homeobox transcription factors make up regulatory modules determined by SCENIC.** Heatmap of scaled regulon activity determined by single-cell regulatory network inference and clustering (SCENIC). Regulons are transcription factors and their putative downstream targets. Rows represent regulons, with transcription factor listed on the right. Columns are each one of the 53 pontomedullary ReSN types. Black outlines indicate LIM-defined modules.

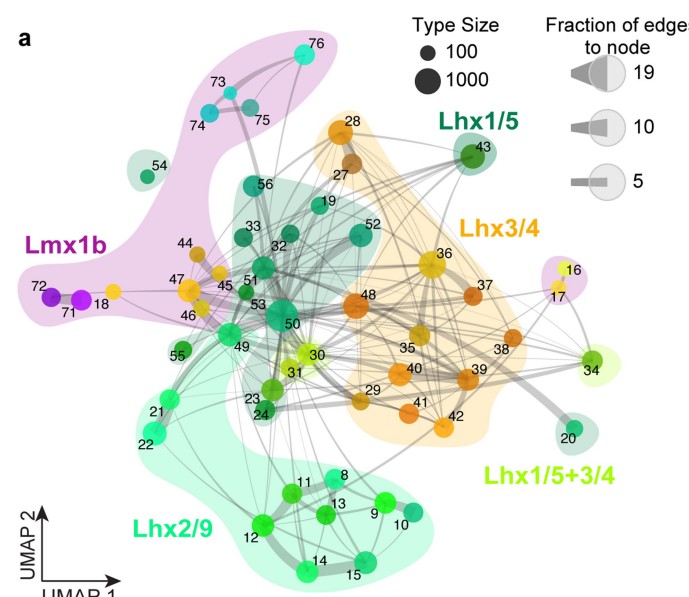

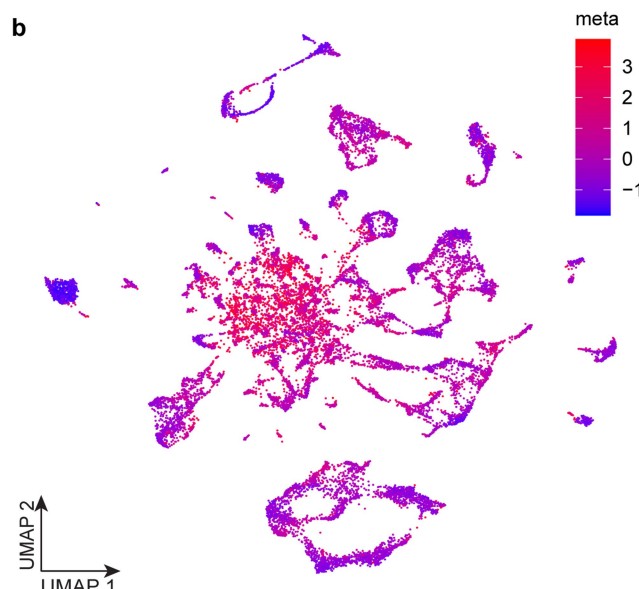

**Extended Data Fig. 17 | LIM-defined pontomedullary reticulospinal neurons have varying levels of complexity. a**, Constellation plot showing global relatedness across all pontomedullary ReSNs. Each transcriptomic type is represented by a node whose area represents the number of nuclei (log-scale). Nodes are positioned at the centre of the corresponding type in UMAP space in Fig. 4a. Relationships between nodes are indicated by edges. Shading behind plot indicates LIM group. Number labels and colors on nodes correspond to type number from Fig. 1c. **b**, Average distance to the K nearest neighbors (KNN) for each nucleus. Nuclei in highly homogenous clusters have much shorter distance to their KNN compared to nuclei in highly heterogenous clusters. This metric can be used to measure the local heterogeneity of each nucleus regardless of their cell type identities.

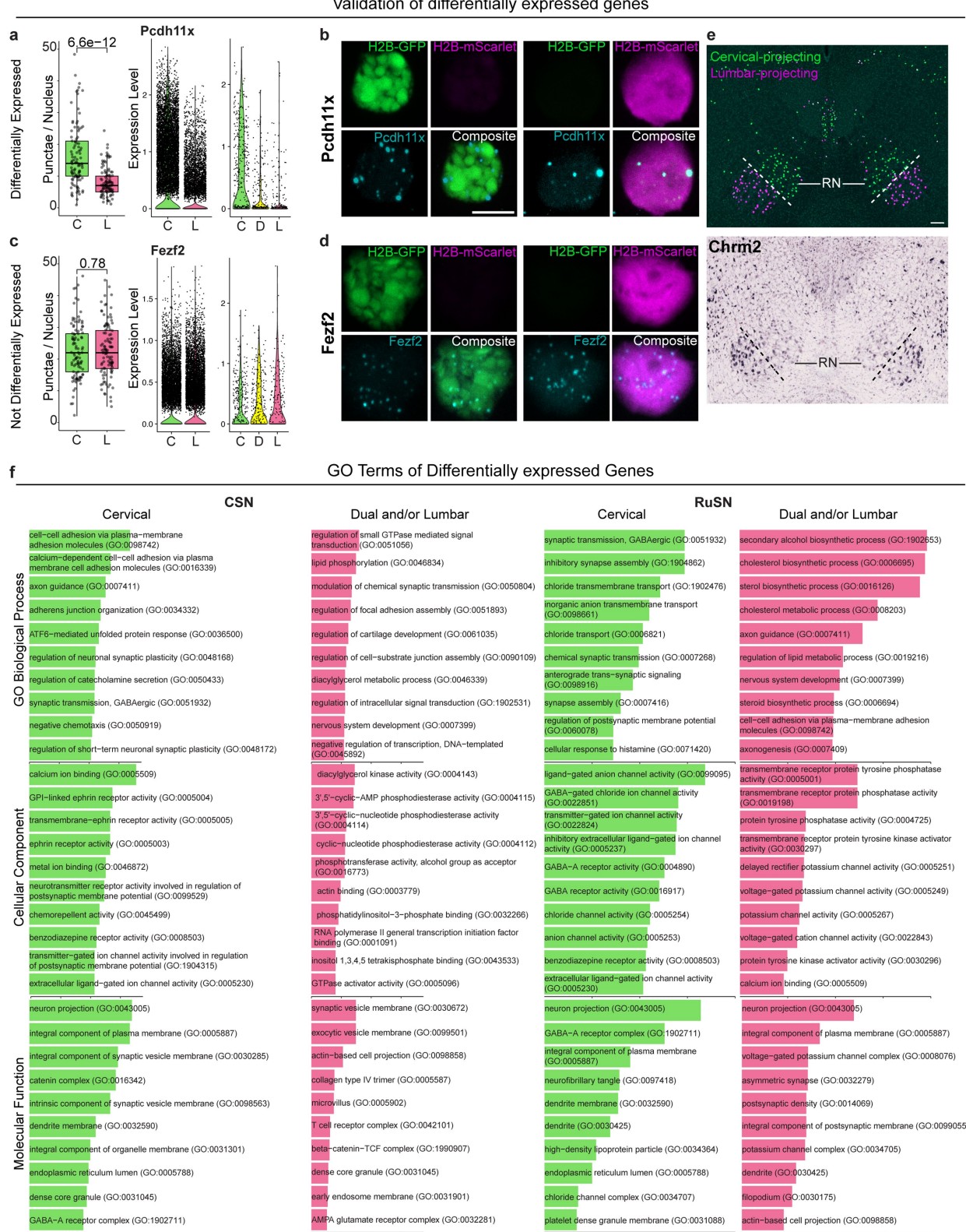

**Extended Data Fig. 18** | See next page for caption.

**Extended Data Fig. 18 | Validation and functional significance of differentially expressed genes between cervical and lumbar projecting neurons.**
**a-d**, Select differentially expressed (i.e., *Pcdh11x*) and control (i.e., *Fezf2*) genes were validated with single molecule fluorescence ISH on nuclei sorted from M1M2S1 dissections. The number of punctae per nucleus was quantified (**a, c**, left boxplot) shown with 10x (**a, c**, middle violin plot) and SSv4 (**a, c** right violin plot) snRNA-seq expression data. Numbers above boxplots indicate p-values (Wilcoxon test, two-sided). The centre line of the box and whisker plots depicts the median value (50th percentile) while the box contains the 25th to 75th percentiles; the whiskers correspond to the 5th and 95th percentiles. *Pcdh11x*: punctae quantified across N = 101 cervical and 102 lumbar nuclei.

*Fezf2*: punctae quantified across N = 101 cervical and 105 lumbar nuclei. **b, d**, representative images of hybridized genes *Pcdh11x* (**b**) and *Fezf2* (**c**) on sorted nuclei from M1M2S1-enriching dissections. Scale, 5 μm. **e**, Confocal image of retrogradely labelled rubrospinal neurons (top; scale, 100 μm), and in situ validation of *Chrm2* (a differentially expressed gene in lumbar-projecting RuSNs) using the Allen ISH Atlas (bottom; Allen Mouse Brain Atlas[63], mouse. brain-map.org). Confocal image as shown in Fig. 2f; representative image from N = 3 injected mice. **f**, Top 10 Gene Ontology (GO) terms for cervical- and lumbar-projecting CSNs and RuSNs. Differential expression was performed with Seurat using the "MAST" test; significant genes were defined as those with an FDR adjusted p-value of less than 0.05.

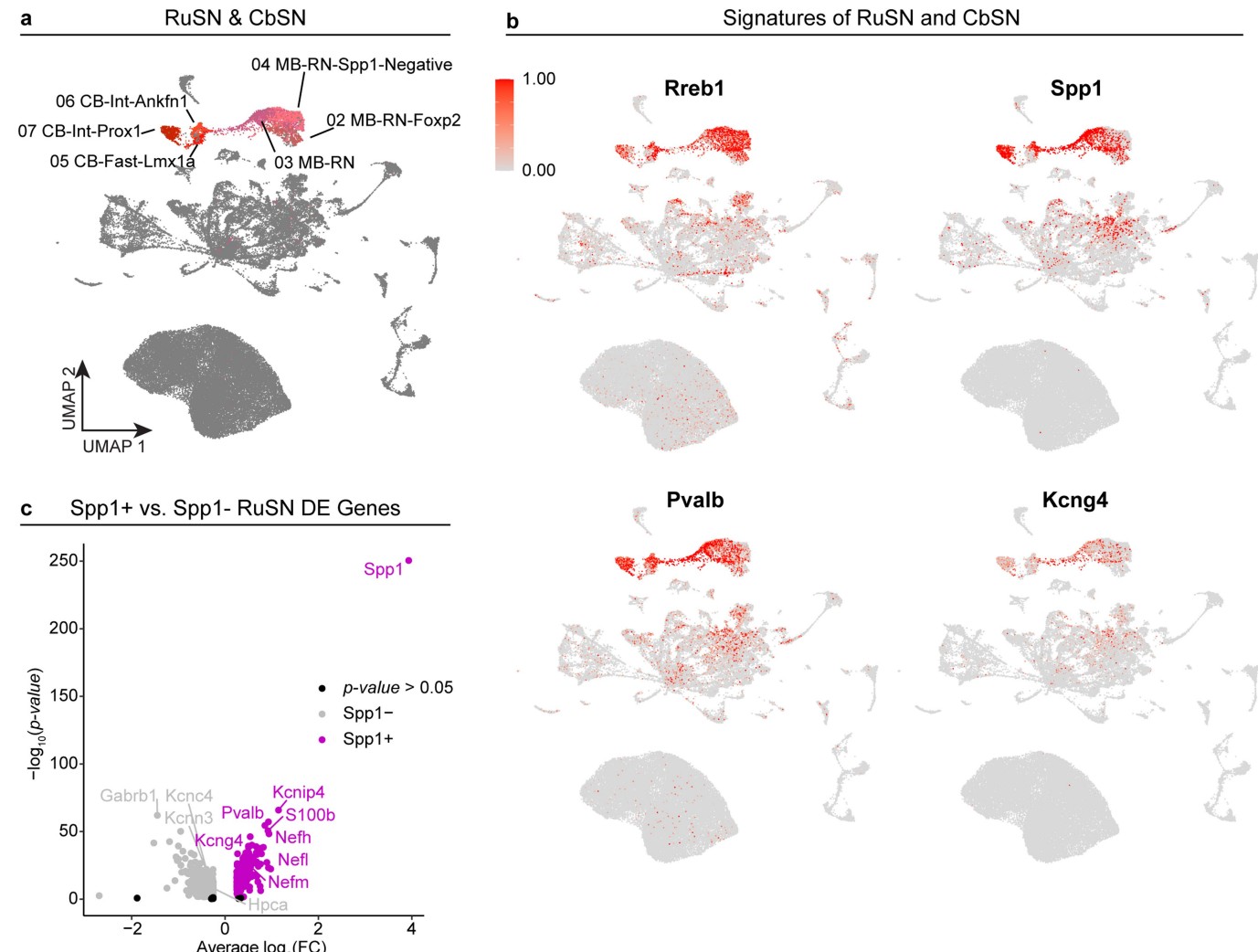

**Extended Data Fig. 19 | Activity- and size-related signatures of rubrospinal, cerebellospinal, and select reticulospinal neurons. a**, UMAP representation of all SPNs with RuSNs and CbSNs highlighted. **b**, Expression of RuSN and CbSN marker genes (*Rreb1, Spp1*) and activity-related genes (*Pvalb, Kcng4*). **c**, Volcano plot of differentially expressed genes between Spp1+ and Spp1- RuSNs (SSv4 dataset, N = 1,031 nuclei). Select genes relevant for cell size (*Spp1, Nefh, Nefm, Nefl, S100b*) and activity (*Pvalb, Kcng4, Kcnip4, Hpca, Kcnn3, Kcnc4, Gabrb1*) are annotated. Differential expression was performed with Seurat using the "MAST" test; significant genes were defined as those with an FDR adjusted p-value of less than 0.05.

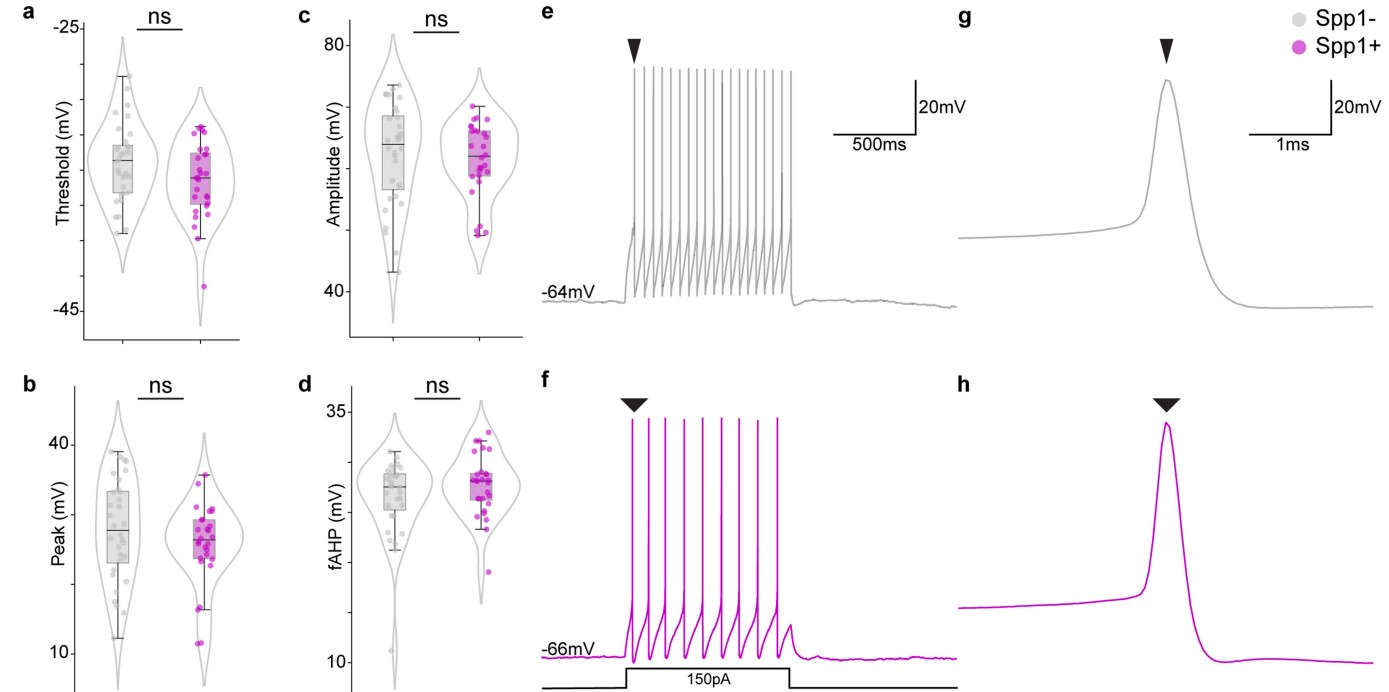

**Extended Data Fig. 20 | Additional electrical properties of Spp1 positive and negative rubrospinal neurons.** Whole-cell recordings of Spp1+ and Spp1- RuSNs showed no significant difference in action potential **a**, threshold, **b**, peak, **c**, amplitude, or **d**, fast afterhyperpolarization (fAHP). ns = not significant (p value > 0.05, Mann-Whitney, two-sided, 28 Spp1+ and 32 Spp1- cells). Number of cells and statistical tests are summarized in Supplementary Table 13.

The centre line of the box and whisker plots depicts the median value (50th percentile) while the box contains the 25th to 75th percentiles; the whiskers correspond to the 5th and 95th percentiles. Representative **(e, f)** traces and **(g, h)** action potentials of Spp1+ and Spp1- RuSNs for 150pA current injections. Arrow heads depict same action potential.

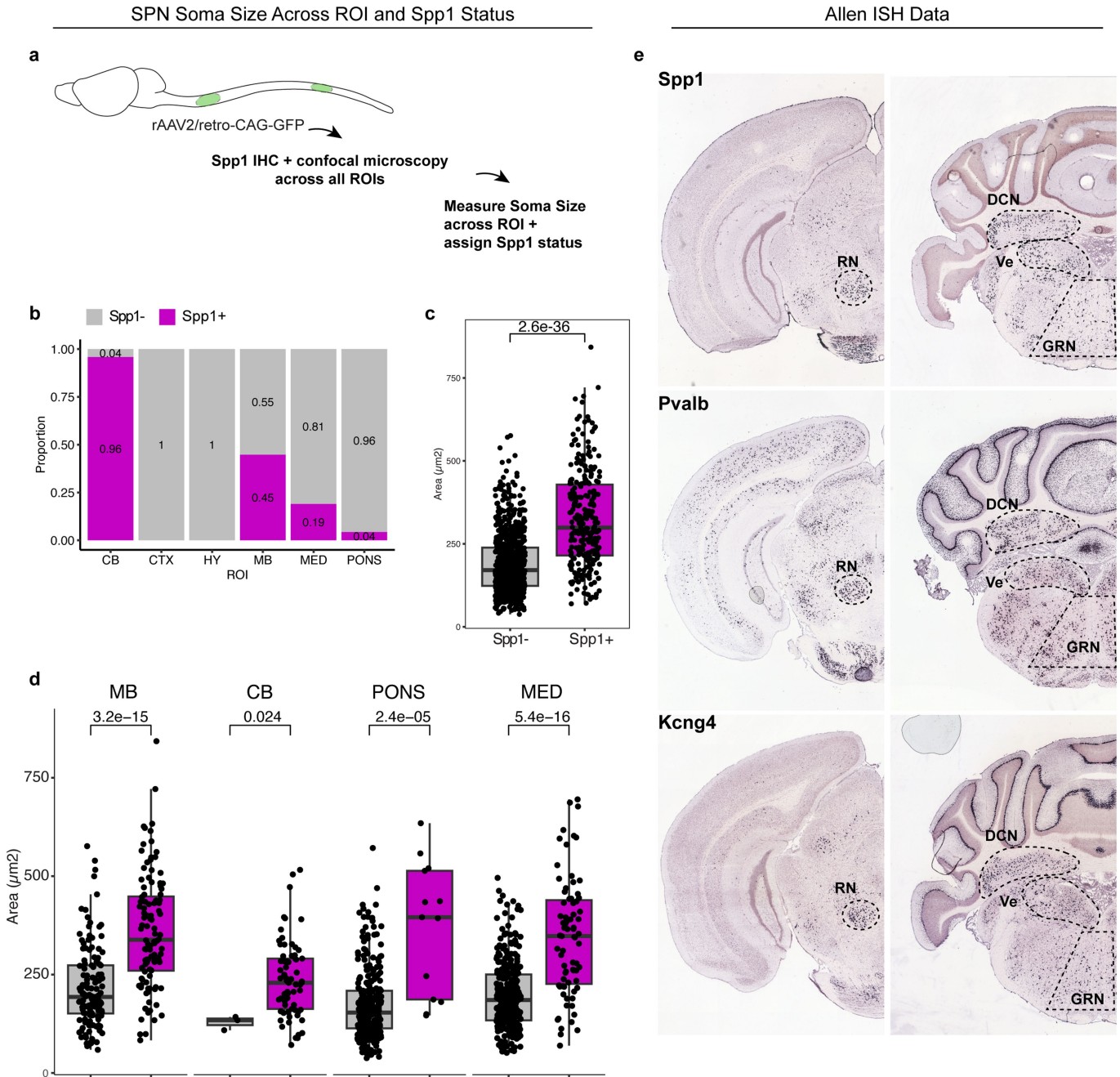

**Extended Data Fig. 21 | Soma size and Spp1 status across spinal projecting neuron populations. a**, Soma of SPNs were labelled with a retrograde AAV expressing GFP under a CAG promoter. Subsequently, IHC for Spp1 was performed, confocal images of each ROI were taken, and soma size / Spp1 status across each major ROI was measured. **b**, Proportion of Spp1+ SPNs across each ROI, as determined by IHC. **c**, Soma area of Spp1+ and Spp1- SPNs in ROIs that contained Spp1+ SPNs (i.e., CB, MB, MED, PONS regions quantified together; N = 260 Spp1+ and 756 Spp1- nuclei). **d**, Soma area of Spp1+ and Spp1- SPNs in ROIs that contained Spp1+ SPNs, separated by ROI (Spp1 + : N = 68 CB, 95 MB,

82 MED, 15 PONS; Spp1-: N = 4 CB, 152 MB, 312 MED, 288 PONS). Numbers above boxplots indicate p-values (Wilcoxon test, two-sided). The centre line of the box and whisker plots depicts the median value (50th percentile) while the box contains the 25th to 75th percentiles; the whiskers correspond to the 5th and 95th percentiles. **e**, Allen ISH Atlas data of *Spp1*, *Pvalb*, and *Kcng4* in RN, DCN, Ve, and GRN (Allen Mouse Brain Atlas[63], mouse.brain-map.org). RN, red nucleus; DCN, deep cerebellar nuclei; Ve, vestibular nucleus; GRN, gigantocellular reticular nucleus.

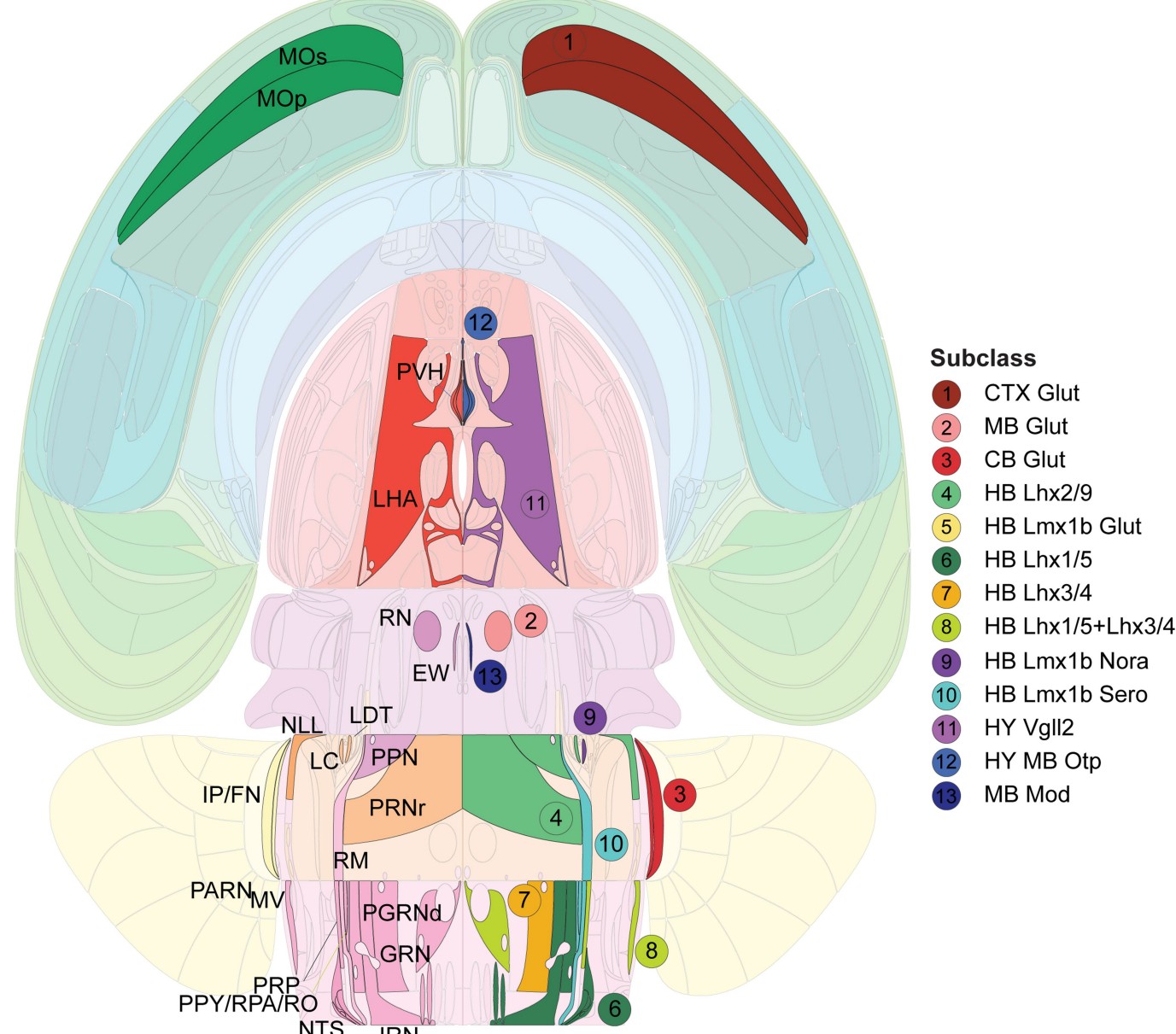

**Extended Data Fig. 22 | Summary of spinal projecting neuron subclass anatomical distribution.** Schematic whole-brain flat map summarizing anatomical distribution of SPN subclasses. In short, the SPN dataset was mapped to MERFISH data which was registered to Allen Brain Atlas (ABA) CCFv3[64]. These CCFv3 regions were annotated onto Swanson flatmap[65] and for each subclass in the SPN dataset the most dominant regions were colored in the flatmap. MOp, primary motor area; MOs, secondary motor area; RN, red nucleus; IP, interposed nucleus; FN, fastigial nucleus; PRNr pontine reticular nucleus rostral part; PPN, pedunculopontine nucleus; NLL, nucleus of the lateral lemniscus; LDT, laterodorsal tegmental nucleus; PARN, parvicellular reticular nucleus; IRN, intermediate reticular nucleus; NTS, nucleus of the solitary tract; MV, medial vestibular nucleus; PGRNd, paragigantocellular reticular nucleus dorsal part; GRN, gigantocellular reticular nucleus; PRP, nucleus prepositus; PPY, parapyramidal nucleus; LC, locus coeruleus; RM, nucleus raphe magnus; RPA, nucleus raphe pallidus; RO, nucleus raphe obscurus; PVH, paraventricular hypothalamic nucleus; LH, lateral hypothalamic area; EW, Edinger-Westphal nucleus.

# Reporting Summary

## Statistics

For all statistical analyses, confirm that the following items are present in the figure legend, table legend, main text, or Methods section.

| n/a | Confirmed | |
|---|---|---|
| ☐ | ☒ | The exact sample size (*n*) for each experimental group/condition, given as a discrete number and unit of measurement |
| ☐ | ☒ | A statement on whether measurements were taken from distinct samples or whether the same sample was measured repeatedly |
| ☐ | ☒ | The statistical test(s) used AND whether they are one- or two-sided *Only common tests should be described solely by name; describe more complex techniques in the Methods section.* |
| ☐ | ☒ | A description of all covariates tested |
| ☐ | ☒ | A description of any assumptions or corrections, such as tests of normality and adjustment for multiple comparisons |
| ☐ | ☒ | A full description of the statistical parameters including central tendency (e.g. means) or other basic estimates (e.g. regression coefficient) AND variation (e.g. standard deviation) or associated estimates of uncertainty (e.g. confidence intervals) |
| ☐ | ☒ | For null hypothesis testing, the test statistic (e.g. *F*, *t*, *r*) with confidence intervals, effect sizes, degrees of freedom and *P* value noted *Give P values as exact values whenever suitable.* |
| ☒ | ☐ | For Bayesian analysis, information on the choice of priors and Markov chain Monte Carlo settings |
| ☒ | ☐ | For hierarchical and complex designs, identification of the appropriate level for tests and full reporting of outcomes |
| ☒ | ☐ | Estimates of effect sizes (e.g. Cohen's *d*, Pearson's *r*), indicating how they were calculated |

*Our web collection on statistics for biologists contains articles on many of the points above.*

## Software and code

Policy information about availability of computer code

| Data collection | Software for mapping and analysis of transcriptomic datasets is fully described in the Methods. |
|---|---|
| | Confocal imaging data was acquired with Zen 3.3 and LAS X Stellaris. Slide scanner imaging data was acquired with VS-ASW. Serial two-photon tomography data was acquired with TissueVision's imaging software. Imaging data was analyzed using QuPath version 0.4.1 and NeuroInfo version 2023-1-1. |
| | Electrophysiology data was recorded with pClamp software (Version 11, Molecular Devices). |
| | Code availability: Code to reproduce analyses here is available at https://github.com/ZhigangHeLab/SPN_atlas. Additional code used in the manuscript is available at https://github.com/AllenInstitute/scrattch.bigcat and https://github.com/AllenInstitute/scrattch.mapping. |
| Data analysis | Software for mapping and analysis of transcriptomic datasets is fully described in the Methods. Software for analysis of imaging datasets (confocal, slide-scanner, and serial two-photon tomography) is fully described in the Methods. |

For manuscripts utilizing custom algorithms or software that are central to the research but not yet described in published literature, software must be made available to editors and reviewers. We strongly encourage code deposition in a community repository (e.g. GitHub). See the Nature Portfolio guidelines for submitting code & software for further information.

## Data

Policy information about <u>availability of data</u>

All manuscripts must include a <u>data availability statement</u>. This statement should provide the following information, where applicable:
- Accession codes, unique identifiers, or web links for publicly available datasets
- A description of any restrictions on data availability
- For clinical datasets or third party data, please ensure that the statement adheres to our <u>policy</u>

> The data are accessible through the Neuroscience Multi-omics (NeMO) Archive (https://assets.nemoarchive.org/dat-76h044v) and the Gene Expression Omnibus (GEO; accession number GSE247602). The AIBS WB atlas data are accessible through NeMO (https://assets.nemoarchive.org/dat-qg7n1b0).

## Research involving human participants, their data, or biological material

Policy information about studies with <u>human participants or human data</u>. See also policy information about <u>sex, gender (identity/presentation), and sexual orientation</u> and <u>race, ethnicity and racism</u>.

| | |
|---|---|
| Reporting on sex and gender | This study did not involve human participants. |
| Reporting on race, ethnicity, or other socially relevant groupings | This study did not involve human participants. |
| Population characteristics | This study did not involve human participants. |
| Recruitment | This study did not involve human participants. |
| Ethics oversight | This study did not involve human participants. |

Note that full information on the approval of the study protocol must also be provided in the manuscript.

# Field-specific reporting

Please select the one below that is the best fit for your research. If you are not sure, read the appropriate sections before making your selection.

☒ Life sciences          ☐ Behavioural & social sciences          ☐ Ecological, evolutionary & environmental sciences

For a reference copy of the document with all sections, see nature.com/documents/nr-reporting-summary-flat.pdf

# Life sciences study design

All studies must disclose on these points even when the disclosure is negative.

| | |
|---|---|
| Sample size | Sample size (number of animals) was determined by the experimental requirements for collection of sufficient tissue for each assay. For snRNA-seq data, nuclei suspensions were generated from 15 adult mice (8 female and 7 male) for 10x and 15 mice (6 female and 9 male) for SSv4. We did not observe differences between individual animals or batches. The number of nuclei collected was determined by limitations of each data modality.<br><br>For electrophysiology and imaging experiments, no statistical methods were used to predetermine sample size. |
| Data exclusions | For transcriptomic data, low quality nuclei were excluded based on criteria that are detailed in the Methods. Briefly, nuclei were filtered to retain only those with less than 5% mitochondrial counts and more than 2000 genes detected. These are standard thresholds used in previous studies. |
| Replication | No data point from electrophysiology data was excluded.<br><br>Findings from SSv4 and 10x platforms were compared across biological replicates. We did not observed any disagreement between replicates. |
| Randomization | Randomization is not applicable for our study since it does not involve a comparison between treatment and control groups. This study characterizes SPN cell types in untreated mice. snRNA-seq provides a random selection of nuclei from the original tissue source. |
| Blinding | Blinding was not needed as there were no treatment and control groups. |

# Reporting for specific materials, systems and methods

We require information from authors about some types of materials, experimental systems and methods used in many studies. Here, indicate whether each material, system or method listed is relevant to your study. If you are not sure if a list item applies to your research, read the appropriate section before selecting a response.

## Materials & experimental systems

| n/a | Involved in the study |
|---|---|
| ☐ | ☒ Antibodies |
| ☒ | ☐ Eukaryotic cell lines |
| ☒ | ☐ Palaeontology and archaeology |
| ☐ | ☒ Animals and other organisms |
| ☒ | ☐ Clinical data |
| ☒ | ☐ Dual use research of concern |
| ☒ | ☐ Plants |

## Methods

| n/a | Involved in the study |
|---|---|
| ☒ | ☐ ChIP-seq |
| ☐ | ☒ Flow cytometry |
| ☒ | ☐ MRI-based neuroimaging |

## Antibodies

| | |
|---|---|
| Antibodies used | Anti-Green Fluorescent Protein Antibody, AVES, SKU: GFP-1020<br>RFP Antibody Pre-adsorbed, Rockland, Item No. 600-401-379<br>Mouse Osteopontin/OPN Antibody, R&D Systems, Catalog#: AF808<br>Mouse anti-NeuN clone A60, Sigma Aldrich, MAB377<br>Donkey Anti-Chicken, Alexa 488 Conjugated, Jackson Immuno 703-545-155<br>Donkey anti-Mouse IgG (H+L) Highly Cross-Adsorbed Secondary Antibody, Alexa Fluor Plus 405, Thermo Fisher, A48257<br>Donkey anti-Rabbit IgG (H+L) Highly Cross-Adsorbed Secondary Antibody, Alexa Fluor Plus 594, Thermo Fisher, A32754<br>Donkey anti-Goat IgG (H+L) Cross-Adsorbed Secondary Antibody, Alexa Fluor 647, Thermo Fisher, A-21447<br>Abcam ab34771, Rabbit-anti-RFP<br>Millipore AP144P, Goat-anti-ChAT<br>Sigma A2052, Rabbit-anti-GABA |
| Validation | Links to datasheets from manufacturers detailing antibody validation are listed below:<br><br>AVES: https://cdn.shopify.com/s/files/1/0512/5793/4009/files/GFP_datasheet.pdf<br><br>Rockland: https://www.rockland.com/datasheet/?code=600-401-379<br><br>R&D Systems: https://resources.rndsystems.com/pdfs/datasheets/af808.pdf?v=20230413&_ga=2.75829141.1486737172.1681410426-2 76112423.1661781903<br><br>Sigma Aldrich: https://www.sigmaaldrich.com/US/en/product/mm/mab377<br><br>Jackson Immuno: https://www.jacksonimmuno.com/catalog/products/703-545-155<br><br>Donkey anti-Mouse: https://www.thermofisher.com/order/genome-database/dataSheetPdf?producttype=antibody&productsubtype=antibody_secondary&productId=A48257&version=288<br><br>Donkey anti-Rabbit: https://www.thermofisher.com/order/genome-database/dataSheetPdf?producttype=antibody&productsubtype=antibody_secondary&productId=A32754&version=288<br><br>Donkey anti-Goat: https://www.thermofisher.com/order/genome-database/dataSheetPdf?producttype=antibody&productsubtype=antibody_secondary&productId=A-2144 7&version=288<br><br>Abcam Rabbit-anti-RFP: https://www.abcam.com/products/primary-antibodies/biotin-rfp-antibody-ab34771.pdf<br><br>Millipore Goat-anti-ChAT: https://www.emdmillipore.com/US/en/product/Anti-Choline-Acetyltransferase-Antibody,MM_NF-AB144P<br><br>Sigma Rabbit-anti-GABA: https://www.sigmaaldrich.com/deepweb/assets/sigmaaldrich/product/documents/290/055/a2052dat.pdf |

## Animals and other research organisms

Policy information about studies involving animals; ARRIVE guidelines recommended for reporting animal research, and Sex and Gender in Research

| | |
|---|---|
| Laboratory animals | Mice were provided food and water ad libitum, housed on a 12-hour light-dark schedule (7am-7pm light period) with no more than 5 mice of the same sex per cage, and allowed to acclimate for 1 week after arrival. Ambient temperature and humidity of housing facility: 71ºF  +/- 3ºF; 35% - 70% +/- 5%<br><br>For sequencing experiments, 6 week old C57BL/6J mice were ordered from Jax, and housed in BCH's animal facility. For histology, 6 to 9 week old C57BL/6J and other Cre line (detailed in Methods section) were used for injection and sacrificed two to four weeks later |

| Wild animals | No wild animals were used in this study. |
|---|---|
| Reporting on sex | Nuclei suspensions were generated from 15 adult mice (8 female and 7 male) for 10x and 15 mice (6 female and 9 male) for SSv4. Sex based analyses were not conducted. |
| Field-collected samples | No field collected samples were used in this study.<br><br>Nuclei suspensions were generated from 15 adult mice (8 female and 7 male) for 10x and 15 mice (6 female and 9 male) for SSv4. |
| Ethics oversight | All experimental procedures were performed in compliance with animal protocols approved by the Institutional Animal Care and Use Committee at Boston Children's Hospital (Protocol #20-05-4165R). |

Note that full information on the approval of the study protocol must also be provided in the manuscript.

# Plants

| Seed stocks | N/A |
|---|---|
| Novel plant genotypes | N/A |
| Authentication | N/A |

# Flow Cytometry

## Plots

Confirm that:

☒ The axis labels state the marker and fluorochrome used (e.g. CD4-FITC).

☒ The axis scales are clearly visible. Include numbers along axes only for bottom left plot of group (a 'group' is an analysis of identical markers).

☒ All plots are contour plots with outliers or pseudocolor plots.

☒ A numerical value for number of cells or percentage (with statistics) is provided.

## Methodology

| Sample preparation | Generation of nuclei suspensions is described in detail in the Methods. |
|---|---|
| Instrument | For 10x, single SPN nuclei were sorted using a BD FACSARIA II with a 70 µm custom pressure nozzle (5O psi). For SSv4, single nuclei were sorted using a Sony SH800 Cell Sorter or MA900 Multi-Application Cell Sorter using a 100 µm chip. |
| Software | Sony Cell Sorter Software was used for the Sony SH800 and MA900 Cell sorters. BD FACSDiva Software was used for the BD FACSARIA II. |
| Cell population abundance | For a representative sample dissected from M1M2S1 cortex, GFP+ nuclei comprised 0.15% of all detected events, and GFP and/or mScarlet+ nuclei comprised 0.21% of all detected events, as shown in Extended Data Figure 3. |
| Gating strategy | We set FACS gating on forward scatter area - side scatter area plot (gate 1), forward scatter height - forward scatter area (gate 2), side scatter height - side scatter width (gate 3), DAPI area - forward scatter area (gate 4) and on fluorescent channels to include only GFP+ or mScarlet+ nuclei (gate 4), as shown in Extended Data Figure 3. |

☒ Tick this box to confirm that a figure exemplifying the gating strategy is provided in the Supplementary Information.

