## [Peer Review File · Nature]

Manuscript Title: A transcriptomic taxonomy of mouse brain-wide spinal projecting neurons

Reviewer Comments & Author Rebuttals

Reviewer Reports on the Initial Version:

Referees' comments:

Referee #1:

Remarks to the Author:

This manuscript by Winter et al. described the effort to conduct single nucleus RNA sequencing (snRNA-seq) on 65,002 retrogradely labelled spinal projecting neurons (SPNs), including 61,484 neurons with 10x v3.1 (targeting 120K reads) and 3,518 neurons with SMART-seq v4 (SSv4, targeting 500K reads). The authors identified 3 divisions, 13 subclasses and 76 neuronal types. Fluorescent activated nucleus sorting (FANS) allows for cervical and lumbar projecting neurons to be profiled separately (e.g., dual projecting neurons with SMART-seq). Integrating this data with whole brain scRNA-seq and MERFISH from a companion paper by the Allen Institute for Brain Science allowed them to delineate three broad components of the SPNs: transcriptomically homogeneous fast-acting excitatory point-to-point system with somatotopic arrangement; heterogeneous fast-acting but broadly distributed system from the reticular formation; modulatory system with slow acting neurotransmitters and/or neuropeptides. Among the Division 1 neurons, molecular markers highly correlate with electrophysiological signatures, e.g., Spp1+ rubrospinal neurons (RuSNs) share features (e.g., fast conducting) with alpha RGCs in the retina. Among the reticulospinal neurons (ReSNs) in Divisions 2 and 3, a LIM homeobox transcription factor code precisely defines 5 spatially segregated and functionally distinct groups, including new classes of reticulospinal neurons not labeled with the Chx10-Cre mouse line commonly used in the literature. To our knowledge, this is the first study to systematically describe brain-wide spinal projecting neurons integrating molecular, anatomical and some electrophysiological measurements, providing a useful foundation for understanding the organizational principles of SPNs. It is especially interesting that developmental genes (e.g., LIM, Eph) continue to be expressed in the adult CNS in a way that is consistent with them underlying the structural organization of the CNS. The study also has important implications in therapeutic development for a number of neurological conditions that affect the communication between the brain and the spinal cord. Thus, this study will have a long-lasting impact on both basic and translational neuroscience.

The following comments are meant to improve the manuscript.

1) Retrograde labeling focuses on the cervical and lumbar spinal cord but does not cover the rest of the cord. This is reasonable for the first study using this approach. The authors may wish to discuss the caveats of missing other parts of the spinal cord (e.g., thoracic) in their conclusions regarding both the broad organizational principles and the detailed clustering results. Along this topic, it will be helpful to show some spinal cord sections (e.g., longitudinal) to illustrate the spread of the virus from the cervical and lumbar injection sites.

2) The authors carefully considered transsynaptic labeling of AAV. However, the 10% XFP cutoff appears to be arbitrary, which can perhaps be best appreciated in Ext. Fig. 3g where some neuronal types hover around 10% and there is at least one known exception (indicated by an *, EW neurons). For instance, what is the level of confidence for HYP_Neuron_SO, MB_Neuron_SO, and Hindbrain_Neuron_SO to be categorized as secondary order neurons? The authors may wish to discuss how future effort may refine the distinction between first and second order neurons (either experimentally or bioinformatically).

3) It is striking that corticospinal neurons (CSNs) exhibit little molecular heterogeneity even though they arise from different brain regions and represent a substantial proportion of the neurons sequenced (29K out of >60K), whereas most other neuronal types are represented only by hundreds of neurons but exhibit much higher molecular heterogeneity (Ext. Fig. 4c). Are there differences in CSNs from RFA, M1M2S1, and S2 in any way (other than their anatomical locations)? What are the possible biological and evolutionary implications of this?

4) Anatomic distribution of different neuronal types is perhaps most impressively illustrated in Ext. Fig. 9 for ReNSs. It is amazing that these heterogeneous neuronal types can be defined so precisely at both the molecular and anatomical levels. It would help to clarify on the numbers of genes used in the MERFISH plots. Also, any discussion on how likely such combinatorial code extends to other neuronal types in the adult CNS would be interesting.

5) Some extended figures such as Ext. Figs. 4b, 9, 11e are so significant / interesting (illustrating iterative clustering, LIM code, etc.) that it is worth considering moving them to main figures.

6) It is impressive that the authors used 13 Cre knockin lines to validate the anatomical distribution of CPNs in the brain (Fig. 4d, Ext. Fig. 7). It would be helpful to discuss a bit more on the relationship between Cre-dependent retrograde tracing data and MERFISH data (both globally and on specific examples).

7) The concept of "cable lines" can be elaborated more, e.g., in the last paragraph of Discussion.

8) It will be interesting to include a brief discussion on the 10x v3.1 versus SSV4 approaches in the context of the entire study, taking into consideration that the cell collection methods used also differ between the two approaches.

Minor:

- Sup Table 6: some gene names are dates by mistake (e.g., 1-Mar, March 1.1, etc.)

Referee #2:

Remarks to the Author:

The manuscript by Winter et al is a technical tour de force and provides an atlas that I presume many people will rely upon for years. Neurons from the brain that descend upon the spinal cord (SPNs) have eluded this type of systematic, transcriptomic, and neuroanatomical characterization. SPNs indeed convert "thoughts" into "actions" as the authors mention, and now the field has a way to determine if/how these molecularly defined SPNs contribute to unique motor, somatosensory, and autonomic functions. In general, the study is well done, rigorous, and robust. Despite my overall enthusiasm however, there are several critiques that I list below:

Critiques:

1. In general, the goal of this study is atlas-based, defining SPN cell types that descend to cervical or lumbar spinal cord. It remains to be determined the functional consequence of manipulating any of these defined SPNs. While the authors clearly cannot delineate every cell type they have identified, a few exemplary cell types with easy to measure behaviors that are modulated by SPNs would greatly strengthen this work. For example, descending modulation of pain from brain to spinal cord is well studied. Could activating or inhibiting one of their SPN cell types modulation pain behavior? This is simply one of many such examples. Adding results like this with a handful of cre, flp, or intersectional lines could show the benefits of this approach, beyond just a mere "parts list" handbook, to showing that

this approach can uncover new principles, or even therapeutic targets.

2. Why the focus on the entire study with AAV2 retro? The authors themselves mention some of the limitations with this strategy. I was surprised not to see any anterograde tracing to at least validate some of the SPN populations. This reciprocal approach could also display where exactly in the spinal cord these SPN populations project to, which could help in deciphering their functions.

3. The depth of the spinal cord injections was 0.60 mm and 1.00 mm. Presumably these deep injections should mainly target the ventral horn. Why the focus here? What about targeting the dorsal horn to uncover SPNs for somatosensation? Along these lines, I could not find a single spinal cord histology image, so we cannot gauge the degree of spread of the initial viral injection (only cartoons are shown for the spinal cord). This is a problem – the authors must show evidence of the injection site.

4. With the rapid development of tissue clearing and imaging approaches, it seems that it is now possible to visualize the entire SPN projection, with brain and spinal cord attached. If the authors could show this, for at least 1 or a handful of SPN populations, this would greatly bolster the study – especially to demonstrate that we are really looking at point-to-point monosynaptic projections.

5. For the spinal cord injections, the authors should also show some cervical and lumbar DRGs to be certain that their injections are not also labeling DRG neurons. Lines like TH-Cre label many DRG and autonomic neurons and thus the route to get to the brain could be different than what the authors envision if some DRGs or sympathetic ganglia are labeled. Checking DRG expression is even more of an issue though for A-Beta mechanoreceptor DRG neurons, where a single cell sends one projection to the spinal cord and another to the brainstem. If these cells happened to be labeled by the AAV injections that the authors would mistakenly assume the brainstem during were SPNs.

6. The authors write in the first few sentences of the paper, “Without commands from the brain, however, the spinal cord cannot initiate and execute any meaningful behavior^{2–4}.” This statement is not true. The authors clearly have a focus and bias in this paper (which is fine), but this statement ignores the importance of ascending sensory-spinal cord-brain circuits, and local spinal cord reflexes that are capable of executing behavior – even in the absence of most of the cortex and other parts of the brain.

Referee #3:

Remarks to the Author:

The study by Winter, Jacobi, Su et al uses retrograde labeling in combination with single nuclei RNA sequencing to identify and transcriptionally characterize spinal projecting neurons. The study design is simple, yet elegant, the analysis is carefully done, the manuscript is well written, and the data is overall clearly presented. I believe the study represents a leap forward in understanding molecularly distinct spinal projecting neurons and will be an incredibly valuable resource for the community moving forward. I have a few comments that I believe would strengthen the manuscript:

- One thing that I believe is missing and could be very helpful is an overall schematic with the findings, combining a map of the brain with regions and subclasses. I appreciate this might be a challenging thing to make considering the complexity and heterogeneity of the data
- From what I can tell from the methods, it looks like no doublet analysis was performed. What was the target nuclei number for 10x? Are the authors not concerned that the presence of doublets could mask the transcriptional signatures of some of the smaller clusters?
- Please mention what age animals were injected with AAV retro in the main text
- While the histogram in Figure 1c is a great way to visualize the transcriptomic taxonomy of SPNs, I wonder if there is a way of including the relative proportions of the three main divisions as well. Right now, without looking at the text it is unclear that eg. CTX MB CB Glut constitute 53% of all the surveyed SPNs. There is some space underneath the dendrogram, and I wonder if showing the division color labels as a barplot with the proportions of each out of all SPNs is an option?
- Similarly, a relative distribution of types would be useful in the extended data
- The authors mention the presence of dual neurotransmitter machinery in MED-Lhx1/5-NTS. Since these

cells only appear to show GAD2 and not SLC32A1, it would be useful to show protein expression of GAD/GABA to substantiate the claim.

- It is unclear why Figures 5b, c do not show the lumbar only category? It would be useful and clearer to include this category (especially since other panels do have all three categories)

- The colors in Figure 6a are hard to see (both for the color-coded UMAP and featureplots). Maybe the pt.size can be increased to help visualization?

- Overall the authors do a good job of putting the study into context and referencing relevant literature. However, other studies using AAV-retro to label projecting neurons should also be mentioned

- While this is an incredibly useful resource, it would be helpful to have an eye on human counterparts for these cell types. I understand that this might be out of the scope for the current manuscript, but since a similar study cannot be generated in humans, the lack of a human comparison seems like a missed opportunity. Data for all the different regions surveyed is not available for human, but mapping of cortical projecting neuron subtypes to human cortical cells would be an important addition.

Author Rebuttals to Initial Comments:

Italic responses refer to referee requests which we believe to be beyond the scope of the current study.

Referees' comments:

Referee #1 (Remarks to the Author):

This manuscript by Winter et al. described the effort to conduct single nucleus RNA sequencing (snRNA-seq) on 65,002 retrogradely labelled spinal projecting neurons (SPNs), including 61,484 neurons with 10x v3.1 (targeting 120K reads) and 3,518 neurons with SMART-seq v4 (SSv4, targeting 500K reads). The authors identified 3 divisions, 13 subclasses and 76 neuronal types. Fluorescent activated nucleus sorting (FANS) allows for cervical and lumbar projecting neurons to be profiled separately (e.g., dual projecting neurons with SMART-seq). Integrating this data with whole brain scRNA-seq and MERFISH from a companion paper by the Allen Institute for Brain Science allowed them to delineate three broad components of the SPNs: transcriptomically homogeneous fast-acting excitatory point-to-point system with somatotopic arrangement; heterogeneous fast-acting but broadly distributed system from the reticular formation; modulatory system with slow acting neurotransmitters and/or neuropeptides. Among the Division 1 neurons, molecular markers highly correlate with electrophysiological signatures, e.g., Spp1+ rubrospinal neurons (RuSNs) share features (e.g., fast conducting) with alpha RGCs in the retina. Among the reticulospinal neurons (ReSNs) in Divisions 2 and 3, a LIM homeobox transcription factor code precisely defines 5 spatially segregated and functionally distinct groups, including new classes of reticulospinal neurons not labeled with the Chx10-Cre mouse line commonly used in the literature. To our knowledge, this is the first study to systematically describe brain-wide spinal projecting neurons integrating molecular, anatomical and some electrophysiological measurements, providing a useful foundation for understanding the organizational principles of SPNs. It is especially interesting that developmental genes (e.g., LIM, Eph) continue to be expressed in the adult CNS in a way that is consistent with them underlying the structural organization of the CNS. The study also has important implications in therapeutic development for a number of neurological conditions that affect the communication between the brain and the spinal cord. Thus, this study will have a long-lasting impact on both basic and translational neuroscience.

We appreciate the positive comments of this reviewer.

The following comments are meant to improve the manuscript.

- 1) Retrograde labeling focuses on the cervical and lumbar spinal cord but does not cover the rest of the cord. This is reasonable for the first study using this approach. The authors may wish to discuss the caveats of missing other parts of the spinal cord (e.g., thoracic) in their conclusions regarding both the broad organizational principles and the detailed clustering results. Along this topic, it will be helpful to show some spinal cord sections (e.g., longitudinal) to illustrate the spread of the virus from the cervical and lumbar injection sites.**

We included a discussion of the caveats of the injection scheme in the "Technical limitations and solutions" section of the Discussion.

Additionally, we included representative confocal microscopy images of coronal and longitudinal (i.e., dorsal-ventral horizontal) sections of the spinal injection sites in new Extended Data Fig. 1. These images show the distribution of injected virus throughout spinal gray matter (i.e., coronal

images) as well as which spinal segments are targeted along the length of the cord (i.e., horizontal images). In addition to showing injection sites of the AAV2/retro vector used to label SPNs, we included images of injection sites using a similar vector, AAV2/2, that does not have retrograde labeling capacity. This allows more precise visualization of transduced cells at the injection site as it does not label intraspinal projection neurons (i.e., propriospinal neurons) that obscure the precise injection level.

- 2) The authors carefully considered transsynaptic labeling of AAV. However, the 10% XFP cutoff appears to be arbitrary, which can perhaps be best appreciated in Ext. Fig. 3g where some neuronal types hover around 10% and there is at least one known exception (indicated by an *, EW neurons). For instance, what is the level of confidence for HYP_Neuron_SO, MB_Neuron_SO, and Hindbrain_Neuron_SO to be categorized as secondary order neurons? The authors may wish to discuss how future effort may refine the distinction between first and second order neurons (either experimentally or bioinformatically).**

We appreciate the positive comments of this reviewer about this effort to carefully consider the caveats of AAV2/retro and agree with the uncertainty of the empirically determined 10% threshold. Future efforts, for example using retrograde dye which do not spread across the cells, may refine the distinction between first- and second-order neurons. Ultimately anterograde tracing should verify these predictions. In this regard, we performed new experiments to validate the MB-EW-Cck-Ucn type as spinal projecting neurons. Specifically, we injected AAV2/9-CAG-FLEX-ChR2-TdTomato into the midbrain Edinger Westphal (EW) nucleus in a Ucn-Cre mouse line. As Ucn is a specific marker of this SPN population in this anatomical region, this injection scheme selectively labels the MB-EW-Cck-Ucn type. We then confirmed the presence and assessed the distribution of labeled axonal fibers in the cervical, thoracic, lumbar, and sacral spinal cord. In addition to presenting these results in new Extended Data Fig. 5, we added a brief discussion of these limitations and plans in the “Technical limitations and solutions” section of the Discussion.

- 3) It is striking that corticospinal neurons (CSNs) exhibit little molecular heterogeneity even though they arise from different brain regions and represent a substantial proportion of the neurons sequenced (29K out of >60K), whereas most other neuronal types are represented only by hundreds of neurons but exhibit much higher molecular heterogeneity (Ext. Fig. 4c). Are there differences in CSNs from RFA, M1M2S1, and S2 in any way (other than their anatomical locations)? What are the possible biological and evolutionary implications of this?**

Although CSNs exhibit relatively little transcriptomic heterogeneity compared to subcortical SPNs, we find that CSNs within the L5PT-CSN type map to several clusters in the AIBS WB scRNAseq taxonomy corresponding to different cortical regions (Yao et al 2023), indicating that CSNs arising from different cortical regions are transcriptionally distinct. Results are summarized in Supplementary Tables 3-5 and new Extended Data Fig. 10. Further, differential expression analysis among CSNs from the three region-enriching dissections (i.e., M1M2S1, RFA, S2) reveal DE genes which are summarized in new Supplementary Table 6.

We are also intrigued by the biological and evolutionary implications of such homogeneity of abundant CSNs. In a recent study by Cheng et al., (Cell 185, 311, 2022), the authors show that vision selectively drives the specification of glutamatergic cell types in upper layers (L) (L2/3/4), while deeper-layer glutamatergic neurons are established prior to eye opening in the mouse visual cortex. Thus, functional specifications of the neurons in Division 1 and perhaps other deep layer

glutamatergic neurons might be mainly determined by their synaptic partners, including both pre-synaptic inputs and post-synaptic targets. We added a brief discussion of this point in the “A three-component organization of SPN projections” section of the Discussion.

- 4) Anatomic distribution of different neuronal types is perhaps most impressively illustrated in Ext. Fig. 9 for ReNSs. It is amazing that these heterogeneous neuronal types can be defined so precisely at both the molecular and anatomical levels. It would help to clarify on the numbers of genes used in the MERFISH plots. Also, any discussion on how likely such combinatorial code extends to other neuronal types in the adult CNS would be interesting.**

500 genes were carefully selected for having the greatest distinguishing capability among clusters in MERFISH experiments based on the whole brain scRNAseq dataset presented in the companion paper Yao et al 2023. We have clarified this point in the “Integration of spinal projecting neuron and whole brain taxonomies” section.

Further, Yao et al 2023 provides a detailed analysis on transcription factor determinants of cell type classification across the whole mouse brain. Specifically, they identified a combinatorial transcription factor code that defines cell types across all parts of the brain. Our results extend these findings specifically to identify a transcription factor code for pontomedullary reticulospinal neuron types, which are better resolved in our study. We included a brief discussion of the whole brain transcription factor combinatorial code identified by Yao et al 2023 and refer the reader to these companion results in the “Molecular insights into reticular formation SPNs” section of the Discussion.

- 5) Some extended figures such as Ext. Figs. 4b, 9, 11e are so significant / interesting (illustrating iterative clustering, LIM code, etc.) that it is worth considering moving them to main figures.**

We appreciate the reviewer’s enthusiasm for the data illustrated in the extended figures. We have re-arranged the figures in their current form to be inclusive of the range of interesting analyses while prioritizing the main findings of the paper. Per this reviewer’s suggestion, we updated main Figure 4 to include some aspects of Ext. Figures 12 and 15 (previously numbered Ext. Data. Fig 9 and 11, respectively). Though we agree that Ext. Fig. 6b (previously Ext. Fig 4b) is illustrative and aids the reader’s understanding of our clustering methodology, we did not move it into main Figure 1 due to the concern that it would make an already busy figure even busier.

- 6) It is impressive that the authors used 13 Cre knockin lines to validate the anatomical distribution of CPNs in the brain (Fig. 4d, Ext. Fig. 7). It would be helpful to discuss a bit more on the relationship between Cre-dependent retrograde tracing data and MERFISH data (both globally and on specific examples).**

We provided an updated Extended Data Fig. 9 (previously Ext. Fig. 7) which now shows the correspondence between Cre-dependent tracing data and MERFISH results for 8 specific examples. We also added a brief discussion of the relationship between the Cre-dependent retrograde tracing and the MERFISH datasets in the second paragraph of section “Integration of spinal projecting neuron and whole brain taxonomies”. In general, we found high anatomical correspondence between the Cre-dependent retrograde tracing and the MERFISH results. However, as Cre-dependent labeling only targets a single gene and MERFISH results reflect expression of multiple genes, we find that the Cre-dependent retrograde tracing results tend to

be less specific, particularly for SPN types that are defined by multiple genes. Further, in principle, Cre line labeling should reflect the history of gene expression in a particular cell (i.e., past and present), but MERFISH results only indicate current expression patterns.

7) The concept of “cable lines” can be elaborated more, e.g., in the last paragraph of Discussion.

We adjusted the language used to describe these electrophysiologically distinct pathways.

8) It will be interesting to include a brief discussion on the 10x v3.1 versus SSv4 approaches in the context of the entire study, taking into consideration that the cell collection methods used also differ between the two approaches.

We added a brief acknowledgement of the utility of complementary 10x and SSv4 approaches in paragraph 1 of the Discussion. Briefly, 10x enabled the development of a high-throughput, comprehensive atlas of SPNs, whereas SSv4 enabled targeted assessment of specific biological features (i.e., spinal projection target and assessment of electrophysiological differences among closely related types). Of note, nuclei were labeled, isolated, and collected via the same protocol for both 10x and SSv4, except 10x used pooled sorting and SSv4 used indexed plate-based sorting (see methods).

Minor:

- **Sup Table 6: some gene names are dates by mistake (e.g., 1-Mar, March 1.1, etc.)**

We thank the reviewer for the careful attention - we have fixed this error (it was caused by Excel formatting; now saved as .csv files).

Referee #2 (Remarks to the Author):

The manuscript by Winter et al is a technical tour de force and provides an atlas that I presume many people will rely upon for years. Neurons from the brain that descend upon the spinal cord (SPNs) have eluded this type of systematic, transcriptomic, and neuroanatomical characterization. SPNs indeed convert “thoughts” into “actions” as the authors mention, and now the field has a way to determine if/how these molecularly defined SPNs contribute to unique motor, somatosensory, and autonomic functions. In general, the study is well done, rigorous, and robust. Despite my overall enthusiasm however, there are several critiques that I list below:

We appreciate the encouraging comments of this reviewer. Critiques:

- 1. In general, the goal of this study is atlas-based, defining SPN cell types that descend to cervical or lumbar spinal cord. It remains to be determined the functional consequence of manipulating any of these defined SPNs. While the authors clearly cannot delineate every cell type they have identified, a few exemplary cell types with easy to measure behaviors that are modulated by SPNs would greatly strengthen this work. For example, descending modulation of pain from brain to spinal cord is well studied. Could activating or inhibiting one of their SPN cell types modulation pain behavior? This is simply one of many such examples. Adding results like this with a handful of cre, flp, or intersectional lines could show the benefits of this approach,**

beyond just a mere “parts list” handbook, to showing that this approach can uncover new principles, or even therapeutic targets.

We agree with this reviewer that the focus of the present manuscript was to contribute a robust, atlas-based characterization of SPN cell types on a brain-wide scale. We believe our work is significant in its current form as it provides the foundation upon which the suggested functional studies could be built. We sincerely appreciate the reviewer’s insightful suggestion to extend our study by determining the functional consequences of manipulating identified SPN cell types, such as investigating their potential role in modulating pain behavior. Indeed, this and similar lines of investigation could significantly enhance the implications of our work, extending its utility beyond a “parts list” and potentially unveiling new principles or therapeutic targets. While we acknowledge the immense scientific value of these proposed investigations, we respectfully note that they represent substantial stand-alone research endeavors and extend significantly beyond the scope of our current manuscript. Given the complexity of the circuits, manipulating one cell type at a time may not generate straightforward behavioral outcomes, and careful circuit dissections may be necessary to generate meaningful insights. Furthermore, these recommended experiments would require comprehensive behavioral studies that exceed the given 4-6 months revision timeline (our experience with this type of experiment indicates this will likely take greater than 1 year to generate reliable, meaningful data).

- 2. Why the focus on the entire study with AAV2 retro? The authors themselves mention some of the limitations with this strategy. I was surprised not to see any anterograde tracing to at least validate some of the SPN populations. This reciprocal approach could also display where exactly in the spinal cord these SPN populations project to, which could help in deciphering their functions.**

Despite some limitations of AAV2/retro (e.g., a small proportion of “second order” labeling), its superior transduction efficiency and high expression levels render it as a tool for high throughput study like this. In addition to our discussion on the limitations of AAV2/retro in the “Technical limitations and solutions” section, we have now included anterograde labeling to validate some example populations, as described above in response to referee #1 comment #2. We anterogradely labeled two populations and assessed termination patterns in the spinal cord. Namely: MB-EW-Cck-Ucn (Ucn-Cre line, Edinger Westphal injection; see above), and MB-RN and MB-RN-Foxp2 (Spp1-Cre line, red nucleus injection). These results are depicted in new Extended Data Fig. 5.

- 3. The depth of the spinal cord injections was 0.60 mm and 1.00 mm. Presumably these deep injections should mainly target the ventral horn. Why the focus here? What about targeting the dorsal horn to uncover SPNs for somatosensation? Along these lines, I could not find a single spinal cord histology image, so we cannot gauge the degree of spread of the initial viral injection (only cartoons are shown for the spinal cord). This is a problem – the authors must show evidence of the injection site.**

As described in response to referee #1 comment #1, we included representative confocal microscopy images of coronal and longitudinal sections of the spinal injection sites in new Extended Data Fig. 1. These images show that our injection scheme does indeed target both ventral and dorsal horns. This might be due to the diffusion of injected viral vectors along the injection sites (both depths) and needle tracks. We added a sentence emphasizing coverage of the dorsal horn, ventral horn, and intermediate zone in the first paragraph of section “Generation of an anatomically informed transcriptomic atlas of spinal projection neurons”. Additionally, although our injection scheme targets the dorsal horn, ventral horn, and intermediate zone, all

areas of the spinal gray matter may not be equally targeted at each injection level. We added a sentence acknowledging this potential caveat in the “Technical limitations and solutions” section.

- 4. With the rapid development of tissue clearing and imaging approaches, it seems that it is now possible to visualize the entire SPN projection, with brain and spinal cord attached. If the authors could show this, for at least 1 or a handful of SPN populations, this would greatly bolster the study – especially to demonstrate that we are really looking at point-to-point monosynaptic projections.**

We are grateful for the reviewer’s suggestion to strengthen our study using tissue clearing and imaging approaches for visualizing entire SPN projections, thereby demonstrating point-to-point monosynaptic projections. However, we respectfully contend that this suggested experiment, while technically feasible, might not significantly bolster the impact of our study. Further, while our collaborators at the Allen Institute have the necessary setup to conduct these experiments, their experience with this approach (including labeling, brain clearing, imaging, and data analysis) suggests this request could take several months to complete even one round of experiment, extending beyond the 4-6 months revision timeline.

Our research employs AAV2/retro, a methodology widely recognized for its ability to determine monosynaptic projections (Tervo et al 2016, Tasic et al 2018, Zhang et al 2021, Wang et al 2018, Wang et al 2022, and others). We are confident that our data, obtained through this method, offer robust evidence of the SPN cell types and their monosynaptic projections. The results presented in the current manuscript are therefore decisive in the context of our objectives. The proposed tissue clearing and imaging technique would primarily serve to visualize these projections in a different manner, without necessarily providing novel or significantly enhanced information. It is also worth mentioning the challenges of visualizing the continuity of individual axons, which adds another layer of complexity to this request.

Given these considerations, we respectfully propose that this recommendation is not essential for the validation of our current results and extends beyond the scope of our current study. We believe our manuscript, as it stands, contributes a robust, atlas-based characterization of SPN cell types on a brain-wide scale, and that any further experiments in cell-type-specific projection patterns with tissue clearing and imaging could be more appropriately explored in dedicated future studies.

- 5. For the spinal cord injections, the authors should also show some cervical and lumbar DRGs to be certain that their injections are not also labeling DRG neurons. Lines like TH-Cre label many DRG and autonomic neurons and thus the route to get to the brain could be different than what the authors envision if some DRGs or sympathetic ganglia are labeled. Checking DRG expression is even more of an issue though for A-Beta mechanoreceptor DRG neurons, where a single cell sends one projection to the spinal cord and another to the brainstem. If these cells happened to be labeled by the AAV injections that the authors would mistakenly assume the brainstem during were SPNs.**

We appreciate the reviewer’s concerns regarding potential confounds from dorsal root ganglion (DRG) neuron labeling. However, we believe there may be a misunderstanding regarding the implications of our use of AAV2/retro injections in the spinal cord gray matter and the biological realities of DRG neuron connectivity.

In our study, we inject AAV2/retro expressing nuclear GFP/mScarlet into the spinal cord gray matter. This method labels the nucleus of any neuron that projects axons directly to the site of injection. As such, the labeled neurons can be propriospinal neurons (cell bodies within the spinal cord), spinal projecting neurons (cell bodies in the brain/brainstem), and sensory DRG neurons (cell bodies in the DRG outside of the spinal cord). However, since the expressed proteins are nuclear targeted GFP or mScarlet, such fluorescent proteins will not appear in their processes, including ascending projection to the brainstem.

- 6. The authors write in the first few sentences of the paper, “Without commands from the brain, however, the spinal cord cannot initiate and execute any meaningful behavior^{2–4}.” This statement is not true. The authors clearly have a focus and bias in this paper (which is fine), but this statement ignores the importance of ascending sensory-spinal cord-brain circuits, and local spinal cord reflexes that are capable of executing behavior – even in the absence of most of the cortex and other parts of the brain.**

We agree that our initial introduction unintentionally underrepresented the role of ascending sensory-spinal cord-brain circuits and local spinal cord reflexes in executing behavior, and the influence of descending pathways on somatosensation. Acknowledging this oversight, we revised the first few sentences of our manuscript.

Referee #3 (Remarks to the Author):

The study by Winter, Jacobi, Su et al uses retrograde labeling in combination with single nuclei RNA sequencing to identify and transcriptionally characterize spinal projecting neurons. The study design is simple, yet elegant, the analysis is carefully done, the manuscript is well written, and the data is overall clearly presented. I believe the study represents a leap forward in understanding molecularly distinct primal projecting neurons and will be an incredibly valuable resource for the community moving forward. I have a few comments that I believe would strengthen the manuscript:

We appreciate the comments provided by this reviewer to strengthen the manuscript.

- 1. One thing that I believe is missing and could be very helpful is an overall schematic with the findings, combining a map of the brain with regions and subclasses. I appreciate this might be a challenging thing to make considering the complexity and heterogeneity of the data**

We have included a summary schematic of the anatomical locations of each subclass as new Extended Data Fig. 22.

- 2. From what I can tell from the methods, it looks like no doublet analysis was performed. What was the target nuclei number for 10x? Are the authors not concerned that the presence of doublets could mask the transcriptional signatures of some of the smaller clusters?**

We removed putative doublets by manually inspecting clusters (e.g., clusters co-expressing neuron-glia markers, or layer-5 and non-layer 5 markers) rather than relying on an automated doublet detection algorithm. This methodology is in line with other published studies (Kozareva et al 2020 Nature, Kamath et al 2021 Nature Neuroscience, Yadav et al 2023 Neuron, etc...). We have clarified this in the methods section.

The decision to not use a Doublet algorithm was based on the following reasons: 1) No need to run for our SSv4 data because of the sorting method (1 nucleus / well), 2) All 10x lanes were under-loaded, with low predicted multiplet rates (see new Supplementary Table 15), and 3) we were concerned about limitations of doublet detection algorithms, including ability to detect homotypic (vs. heterotypic) doublets. Since CSNs appear to be much more homogeneous transcriptionally than subcortical SPNs, we predict that these doublet detection algorithms would perform worse on this population.

3. Please mention what age animals were injected with AAV retro in the main text

We added age of injection to the main text in the section “Generation of an anatomically informed transcriptomic atlas of spinal projecting neurons”.

4. While the histogram in Figure 1c is a great way to visualize the transcriptomic taxonomy of SPNs, I wonder if there is a way of including the relative proportions of the three main divisions as well. Right now, without looking at the text it is unclear that eg. CTX MB CB Glut constitute 53% of all the surveyed SPNs. There is some space underneath the dendrogram, and I wonder if showing the division color labels as a barplot with the proportions of each out of all SPNs is an option?

We added a bar plot with the percentages of SPNs within each of the three main divisions under the dendrogram in Figure 1. We also added percentages to the UMAP labels in panels d (ROI) and e (Division).

5. Similarly, a relative distribution of types would be useful in the extended data

Extended Data Fig. 6c depicts the number of nuclei of each type (the 2nd column from the right).

6. The authors mention the presence of dual neurotransmitter machinery in MED-Lhx1/5-NTS. Since these cells only appear to show GAD2 and not SLC32A1, it would be useful to show protein expression of GAD/GABA to substantiate the claim.

We included MERFISH data further substantiating co-expression of Slc17a6 and Gad2 with low Slc32a1 expression (Extended Data Fig. 14a,b). MED-Lhx1/5-NTS maps to clusters 4361, 4285, 4316 in the AIBS WB atlas, which similarly show co-expression of Slc17a6 and Gad2 with low expression of Slc32a1.

Additionally, as the reviewer suggested, we performed immunohistochemistry for GABA on retrogradely labeled cells in the nucleus tractus solitarius. To specifically label the MED-Lhx1/5-NTS type, we performed retrograde labeling with a Cre-dependent virus (AAV2/retro-CAG-FLEX-H2B-GFP) in a Gcg-Cre line. IHC for GABA supports presence of functional Gad2 protein in Gcg+ nuclei labeled with Cre-dependent H2B-GFP in a Gcg-Cre transgenic line (Extended Data Fig. 14c,d).

7. It is unclear why Figures 5b, c do not show the lumbar only category? It would be useful and clearer to include this category (especially since other panels do have all three categories)

As shown in the schematic for Figure 5a, the 10x dataset was generated from separately collected cervical-projecting (GFP+) versus dual- (GFP+/mScarlet+) and lumbar- (mScarlet+)

projecting SPNs nuclei across the whole brain; *sort yield limitations precluded separating dual- from lumbar-projecting SPNs in the 10x dataset*. The SSV4 dataset used indexed plate-based sorting to separate cervical- (GFP+), lumbar- (mScarlet+), and dual- (GFP+/mScarlet+) projecting SPNs (Fig. 5a). We have clarified this point in paragraph 1 of “Molecular specification of spinal projection target” section.

8. The colors in Figure 6a are hard to see (both for the color-coded UMAP and featureplots). Maybe the pt.size can be increased to help visualization?

We updated the point size to improve visualization. We also added a MERFISH plot to panel b to demonstrate corresponding spatial representation of clusters 02-04 in panel a.

9. Overall the authors do a good job of putting the study into context and referencing relevant literature. However, other studies using AAV-retro to label projecting neurons should also be mentioned

We initially referenced Tervo et al 2016, Tasic et al 2018, Wang et al 2018, Golan et al 2023, Wang et al 2022, and Beine et al 2022. We have added the following references: Economo 2018, Kim et al 2019, and Yang 2023.

10. While this is an incredibly useful resource, it would be helpful to have an eye on human counterparts for these cell types. I understand that this might be out of the scope for the current manuscript, but since a similar study cannot be generated in humans, the lack of a human comparison seems like a missed opportunity. Data for all the different regions surveyed is not available for human, but mapping of cortical projecting neuron subtypes to human cortical cells would be an important addition.

We sincerely value the reviewer’s insightful suggestion to compare our findings with human counterparts. Indeed, this is a fascinating direction for further investigation and one that we plan to explore in future studies. In this revision, we have added a few sentences highlighting the conserved general organization of SPNs across the species and primate-specific Betz cells, in a hope to stimulate future studies to compare the molecular and functional differences of SPNs in primates and rodents.

Other updates

- 1. Updated mapping and integration to AIBS WB dataset (Yao et al 2023).** Our collaborators at AIBS have revised their mouse WB cell type taxonomy during the revision process of their companion manuscript (Yao et al 2023). In particular, they refined the clustering of a few midbrain/hindbrain neuronal subclasses that appeared messy and also supplemented their scRNA-seq dataset with a small number of snRNA-seq data (1,687 nuclei) to fill some missing cell types (e.g., midbrain red nucleus SPNs). The revised mouse WB taxonomy now includes 34 classes, 338 subclasses, 1,201 supertypes and 5,322 clusters overall. They also remapped their MERFISH data into the revised taxonomy, which resulted in improved mapping. In the revisions of our manuscript, we have updated mapping and integration of SPNs to the whole brain scRNAseq and MERFISH datasets, based on updates Yao et al made during their revision process. These changes are reflected in Supplementary Tables 3-5, Extended Data Fig. 8, and all figures that show MERFISH sections. These changes do not alter

the overall results and conclusions of our study; they only finetune and improve the consistency between our study and the Yao et al study.

- 2. Retrograde labeling in Chx10-Cre mouse line.** We performed retrograde labeling with the Cre-dependent virus in a Chx10-Cre mouse line to improve visualization of the HB Lhx3/4 subclass anatomical representation in Figure 4 panel d-4.
- 3. Added representative electrophysiology traces.** We added representative traces and action potentials of Spp1+ and Spp1- RuSNs to Extended Data Fig. 20.
- 4. Minor text edits throughout to improve clarity.**
- 5. Minor aesthetic changes to figures throughout to improve visualization.**

Reviewer Reports on the First Revision:

Referees' comments:

Referee #1 (Remarks to the Author):

The authors have adequately addressed the concerns of all reviewers, with welcome additions and revisions. This is a beautifully executed study that will provide a valuable resource for the wide neuroscience research community and I would recommend its publication in Nature without further delay.

Notable revisions in response to my concerns include:

- 1) New Ext. Data Fig. 1 shows spinal cord injection sites of both retro AAVs and regular AAVs, illustrating the extent of the viral spread.
- 2) New Ext. Data Fig. 5 shows two examples of anterograde tracing to validate direct axon projections into the spinal cord (instead of through secondary order neurons).
- 3) The newly added section "Technical limitations and solutions" is excellent.
- 4) New Ext. Data Fig. 10 shows that CSNs arising from different cortical regions are transcriptomically distinct.
- 5) Updated discussion in "Molecular insights into reticular formation SPNs" is useful.
- 6) Updated Fig. 4 includes some key aspects of two previous Ext. Data Figures.
- 7) Updated Ext. Data Fig. 9 shows correspondence between Cre line results and MERFISH datasets, with clarifying discussion in Results.

For the points raised by Reviewer 2 on functional validation of selected new neuronal types with behavioral assessment and 3D imaging of cleared tissues, these would represent excellent future direction following the study, but I agree with the authors that they are beyond the scope of the current study due to technical challenges and the lengthy nature of these studies (if they are to be carefully done to generate results that can stand the test of time). No single paper can solve all the problems and caveats. The current manuscript already represents an unusually large amount of work / data with new significant insights into the biology of spinal projecting neurons.

Referee #2 (Remarks to the Author):

The authors have carefully and thoughtfully addressed my comments and concerns. I fully support publication now and look forward to seeing this work in press. Nice job!

Referee #3 (Remarks to the Author):

The authors addressed all comments satisfactorily